# Felis Catus Optimization (FCO): A novel nature-inspired metaheuristic algorithm

**Mohammad Salehi**[1]*, **Raouf Khayami**[2], **Mirpouya Mirmozaffari**[3]

**1** Shiraz University of Technology, Shiraz, Iran, **2** Shiraz University of Technology, Shiraz, Iran, **3** Department of Industrial Engineering, Dalhousie University, Halifax, Canada

* Mo.salehi@sutech.ac.ir

## Abstract

This study introduces Felis Catus Optimization (FCO), a novel nature-inspired metaheuristic algorithm modeled on the ecological and adaptive behavioral dynamics of urban domestic cats. FCO divides its population into explorer (male) and exploiter (female) agents to maintain a dynamic equilibrium between global search and local refinement. Male agents perform asynchronous triplet movements governed by adaptive exploration scaling, while female agents execute Gaussian-based local exploitation and cooperative litter burst. A rejuvenation-and-noise ecological cycle replaces explicit renewal events, sustaining diversity and preventing stagnation through random reallocation and mild environmental perturbation. These mechanisms collectively achieve continuous exploration using direct position-update rules. Extensive experiments on CEC 2005 and CEC 2017 benchmarks confirmed FCO's competitive behavior ranking among top optimizers and outperforming seven algorithms significantly under Holm's post-hoc procedure ($p < 0.05$). The critical-difference (CD) analysis positioned FCO in the central, statistically equivalent cluster, validating its robust convergence pattern. Applications to three real-world engineering design problems demonstrated consistent near-optimal performance and low result variance. Overall, FCO exhibits stable convergence, reliable population renewal, and strong resilience against premature stagnation, establishing it as a scalable and dependable optimizer for continuous and constrained engineering problems.

## 1. Introduction

Optimization is a fundamental problem in various scientific and engineering disciplines, seeking to find the best solution from a set of feasible solutions. The complexity of many real-world problems, characterized by non-linearity, high dimensionality, non-convexity, and the presence of multiple local optima, often renders traditional mathematical optimization techniques inadequate. These classical methods, such as gradient descent or linear programming, can get trapped in local minima or fail

---

**Data availability statement:** All relevant data, including the source code and output results used in this study, are publicly available within the paper and its Supporting information files, as well as in the Zenodo repository (https://doi.org/10.5281/zenodo.18393892).

**Funding:** The author(s) received no specific funding for this work.

**Competing interests:** NO authors have competing interests Enter: The authors have declared that no competing interests exist.

to converge in the presence of complex search spaces [1,2]. In response to these limitations, metaheuristic algorithms have emerged as powerful and versatile tools for tackling intricate optimization challenges. These algorithms, drawing inspiration from natural phenomena, biological evolution, and physical processes, provide robust and adaptive frameworks for exploring vast search spaces and identifying near-optimal solutions [3]. The behavior of animals, particularly their social interactions, behavioral metaphor (used here purely as a behavioral metaphor rather than a formal model), and survival strategies, offers a rich source of inspiration for designing new optimization algorithms. Understanding these natural systems can lead to algorithms that possess inherent mechanisms for balancing exploration and exploitation, maintaining population diversity, and adapting to dynamic environments. Algorithms such as Particle Swarm Optimization (PSO) [4], Grey Wolf Optimizer (GWO) [5], and Whale Optimization Algorithm (WOA) [6] have demonstrated significant success, yet many continue to grapple with fundamental challenges, including premature convergence, loss of population diversity, and a suboptimal balance between global exploration and local exploitation [7]. The inherent difficulty in striking an effective balance between exploring diverse scales of the search space and exploiting promising areas for refinement often limits the performance of existing metaheuristics. While exploration mechanisms are crucial for avoiding entrapment in local optima, efficient exploitation is vital for achieving high solution accuracy and convergence speed [8]. Addressing this critical trade-off is paramount for developing algorithms that are both broadly applicable and finely tuned. This paper introduces the Felis Catus Optimization (FCO) algorithm, a novel nature-inspired metaheuristic algorithm designed to address the challenges of complex optimization problems. FCO is conceptually derived from the observed behavioral and social dynamics of urban domestic cats (Felis catus). These animals exhibit a fascinating interplay of independent exploration and coordinated exploitation within their behavioral ranges, driven by social hierarchies and survival needs. FCO dynamically partitions its population into distinct 'explorer' and 'exploiter' agents. Explorers utilize adaptive random walks across **continuous search space** for broad and balanced search coverage. Exploiters, conversely, engage in Gaussian-perturbed local searches based on fitness comparison, akin to a cat meticulously investigating a specific promising area for prey. This dynamic segregation and distinct strategic focus allow FCO to maintain population diversity while enabling deep local refinement, embodying a sophisticated, biologically grounded approach. The core contribution of this work lies in the development and rigorous evaluation of FCO. Specifically, we present:

- Novel Algorithm Design: A novel optimization framework inspired by feline behavior, integrating adaptive exploration and exploitation strategies.

- Dual-Agent Strategy: A unique population partitioning mechanism that dynamically assigns roles to explorer (male) and exploiter (female) agents.

- Comprehensive Evaluation: Extensive performance evaluation across 17 CEC 2005 and 28 CEC 2017 benchmark functions, demonstrating FCO's statistical advantage in convergence speed and solution accuracy as confirmed by p value

against established algorithms like PSO [4], GWO [5], Manta Ray Foraging Optimization (MRFO), Equilibrium Optimizer (EO), Sine Cosine Algorithm (SCA), Sand Cat Optimizer (SCO), Elite Based Optimizer with Covariance Matrix Adaptive Reconstruction (EBO CMAR), Covariance Matrix Adaptation Evolution Strategy (CMA ES), Runge Kutta Optimization Algorithm (RUN), Cheeta Optimization Algorithm (ChOA), Differential Evolution (DE), L-SHADE Algorithm (L-SHADE), Coyote Optimization Algorithm (COA), Reptile Search Algorithm (RSA) [9], Quantum-Inspired Optimization (QIO), Salp Swarm Algorithm (SSA) [10], and Whale Optimization Algorithm (WOA) [6]. Successful application and validation of FCO on three complex engineering design problems (welded beam, compression spring, and pressure vessel optimization), highlighting its ability to address nonlinear constraints and real-world feasibility requirements, as indicated by the test cases.

- Comparative Analysis: A key innovation lies in the algorithm's adaptive global exploration mechanism, which enhances the diversity and coverage of the search space. This approach enables controlled yet flexible movements, allowing agents to systematically explore both local and global scales, thereby improving the algorithm's ability to locate optimal solutions in challenging environments. Complementing this, a tournament-based renewal strategy facilitates dynamic exchanges between subpopulations, preserving diversity and mitigating the risk of premature convergence. This mechanism ensures robust search dynamics, maintaining the algorithm's adaptability throughout the optimization process. Table 1 provides a structural comparison between the proposed Felis Catus Optimization (FCO) algorithm and a representative set of widely used metaheuristics, covering particle swarm based, predator prey inspired, and physics based designs. Seven key features (summarized in Table 1) were chosen to capture structural and mechanistic diversity: (1) explicit population partitioning into specialized explorer and exploiter subpopulations, (2) adaptive control of search parameters, (3) Adaptive parameter adjustment, (4) a dedicated migration (population-renewal) mechanism (Rejuvenation + Noise Injection + Female switching among candidate males (fitness-based)), (5) lifecycle modeling via aging and elimination, and (6) search strategies driven by direct social interactions.

**Table 1. Structural comparison of the proposed Felis Catus Optimization (FCO) algorithm with 17 competitor metaheuristic methods across seven design criteria. Checkmarks (✓) indicate the presence of a feature in the algorithm, while crosses (✗) denote its absence.**

| Algorithm | Population division (explorer/exploiter) | adaptive control of search parameters | Adaptive parameter adjustment | Migration mechanism | Aging/ elimination dynamics | Social/ interaction-based inspiration |
|---|---|---|---|---|---|---|
| GWO | ✗ | ✗ | ✗ | ✗ | ✗ | ✓ |
| WOA | ✗ | ✗ | ✗ | ✗ | ✗ | ✓ |
| PSO | ✗ | ✗ | ✓ | ✗ | ✗ | ✓ |
| DE | ✗ | ✗ | ✓ | ✗ | ✗ | ✗ |
| L-SHADE | ✗ | ✗ | ✓ | ✗ | ✗ | ✗ |
| RUN | ✗ | ✗ | ✓ | ✗ | ✗ | ✗ |
| CMA-ES | ✗ | ✗ | ✓ | ✗ | ✗ | ✗ |
| CMAR | ✗ | ✗ | ✓ | ✗ | ✗ | ✗ |
| EO | ✗ | ✗ | ✓ | ✗ | ✗ | ✗ |
| MRFO | ✗ | ✗ | ✓ | ✗ | ✗ | ✓ |
| SSA | ✗ | ✗ | ✗ | ✗ | ✗ | ✓ |
| RSA | ✗ | ✗ | ✗ | ✗ | ✗ | ✓ |
| SCA | ✗ | ✗ | ✓ | ✗ | ✗ | ✓ |
| SCO | ✗ | ✗ | ✗ | ✗ | ✓ | ✓ |
| ChOA | ✗ | ✗ | ✓ | ✗ | ✗ | ✓ |
| COA | ✗ | ✗ | ✗ | ✗ | ✗ | ✓ |
| QIO | ✗ | ✗ | ✓ | ✗ | ✗ | ✓ |
| FCO (Proposed) | ✓ | ✓ | ✓ | ✓ | ✓ | ✓ |

As shown in Table 1, most metaheuristic algorithms implement only a subset of these features, often focusing narrowly on either swarm dynamics or stochastic movement rules. In contrast, FCO uniquely incorporates all seven, resulting in a modular hybrid framework inspired by biological behaviors. This integration enables sustained population diversity, dynamic exploration–exploitation balancing, and context-adaptive search trajectories, which collectively underpin its competitive performance across benchmark and real-world optimization tasks. In the broader taxonomy of optimization methods, algorithms are defined as hybrid strategies that combine population-based global search with individual-level local refinement, often inspired by cultural evolution concepts. By integrating a distinct local search phase (female agents' Gaussian exploitation) into a structured global search process based on adaptive global exploration (male agents), the proposed FCO operationalizes this framework at both strategic and mechanistic levels. The implementation of FCO in MATLAB is designed with modularity at its core, featuring distinct functions for benchmarking, boundary management, and movement control. This flexible architecture not only supports seamless integration with real world optimization challenges but also enables hybridization with other algorithmic models. The modular design facilitates customization and scalability, making FCO well suited for applications such as constrained engineering design and dynamic optimization tasks. Furthermore, this framework lays a strong foundation for future enhancements, fostering interdisciplinary research and the adaptation of FCO to a wide range of complex problems.

These contributions collectively establish FCO as an innovative and versatile metaheuristic, with significant potential for addressing complex optimization challenges in engineering, data science, and related fields, while offering a foundation for future algorithmic advancements.

The structure of this paper is as follows: Section 2 reviews related work, encompassing heuristic methods and metaheuristic algorithms, including single solution-based metaheuristics, population based evolutionary algorithms, swarm intelligence-based algorithms, and nature and behavior inspired metaheuristics. Section 3 introduces FCO, detailing its biological inspiration derived from the social and adaptive behavioral dynamics of urban cats, followed by the mathematical formulation of the algorithm. This section further elaborates on key components, including population initialization, adaptive behavioral allocation, exploration behavior (male cats), exploitation behavior (female cats), competitive selection dynamics, aging and elimination, environmental rejuvenation, and a summary of the algorithm's flow. Section 4 presents the results and discussion of the proposed FCO algorithm. This section first evaluates the performance of FCO on benchmark test functions through a comprehensive experimental setup, including optimization results and statistical analyses based on the Friedman test and the Wilcoxon signed-rank test. Subsequently, real-world engineering design problems namely the tension/compression spring design, welded beam design, and pressure vessel design are investigated to demonstrate the applicability and robustness of FCO in constrained optimization scenarios. Section 5 concludes the paper by summarizing the main findings and outlining directions for future research.

## 2. Related work

Metaheuristic algorithms have emerged as vital tools for tackling complex optimization problems, particularly when classical methods are ineffective due to nonlinearity, high dimensionality, or the absence of gradient information [7]. Drawing inspiration from natural phenomena, these algorithms are designed to effectively balance exploration and exploitation within vast, multimodal search spaces [11]. This study classifies optimization approaches into two primary categories: (i) heuristic methods and (ii) metaheuristic algorithms, with each category further subdivided based on their search strategies and sources of inspiration. This structured classification enhances the conceptual understanding of their mechanisms and practical applicability, providing a robust framework for evaluating innovative algorithms like FCO, which leverages adaptive, nature-inspired behaviors to address diverse optimization challenges in engineering, data science, and beyond.

## 2.1 Heuristic methods

Heuristic methods are problem specific optimization techniques that leverage domain knowledge or logical rules to deliver satisfactory solutions within a reasonable timeframe. These approaches are typically deterministic, efficient for well-defined problems, and well suited for small to medium sized instances. The Lagrangian Multiplier Method, a classical mathematical optimization technique, introduces multipliers to manage constraints [12], making it particularly effective for constrained continuous optimization problems. Its analytical tractability is a key strength, though it may falter in non-convex landscapes [13]. The Simplex Method, developed by George Dantzig, addresses linear programming problems through vertex-to-vertex traversal of the feasible scale [14]. Renowned for its efficiency in convex, linear problems, it is widely applied in operations research. Branch and Bound (B&B), a tree-based search method, systematically explores sub problems in combinatorial and integer optimization, pruning branches that cannot surpass the current best solution [15]. While it guarantees global optima for discrete problems, its computational cost can be prohibitive. The APPROX Algorithm, a heuristic approximation method, balances solution quality and computational efficiency for complex combinatorial problems, particularly in NP hard settings [16]. The DAVID Method, inspired by divide and conquer principles, decomposes problems into smaller sub problems, solves them independently, and combines results [16], offering fast convergence and simplicity. These heuristic methods provide a foundation for understanding the evolution of optimization strategies, paving the way for advanced metaheuristics like FCO, which build on their principles to address more complex, multimodal optimization challenges with greater flexibility and robustness.

## 2.2 Metaheuristic algorithms

Metaheuristic algorithms are indispensable for addressing complex optimization problems, particularly when classical methods are hindered by nonlinearity, high dimensionality, or the absence of gradient information [17]. Drawing inspiration from natural phenomena, these algorithms are engineered to effectively balance exploration and exploitation within expansive, multimodal search spaces [18]. This study classifies metaheuristic algorithms into four primary categories: (i) single solution-based methods, (ii) population based evolutionary methods, (iii) swarm intelligence-based algorithms, and (iv) nature and behavior inspired strategies. This structured classification elucidates their conceptual mechanisms and practical applicability, providing a robust framework for contextualizing innovative approaches like FCO, which leverages adaptive, nature-inspired behaviors to tackle diverse optimization challenges with enhanced efficiency and robustness across engineering, data science, and related domains.

**2.2.1 Single solution based algorithms.** Single solution metaheuristics iteratively refine a single candidate solution, making them particularly effective for low dimensional or smoother search spaces. These methods incorporate mechanisms to escape local optima, enhancing their robustness. Simulated Annealing (SA), inspired by the thermodynamic process of annealing in solids [19], employs a temperature parameter to control the acceptance probability of suboptimal solutions, enabling early exploration to avoid local optima. While these algorithms are relatively simple and require minimal parameter tuning, they are often surpassed by population-based methods, such as FCO, in high dimensional or rugged landscapes. The simplicity of single solution metaheuristics provides valuable insights into local search dynamics, complementing the adaptive and population-based strategies of FCO, which excel in tackling complex, multimodal optimization challenges across diverse domains.

**2.2.2 Population based evolutionary algorithms.** These methods maintain and evolve a population of candidate solutions through iterative processes, leveraging operators such as crossover, mutation, and selection, inspired by Darwinian evolution. Genetic Algorithms (GAs) emulate natural selection by applying genetic operators to a population of individuals [20], enabling progressive improvement of solutions across generations. Covariance Matrix Adaptation Evolution Strategy (CMA-ES) is a stochastic evolutionary algorithm that adapts the covariance matrix of a multivariate Gaussian distribution to guide population sampling. By learning correlations between decision variables, CMA-ES efficiently reshapes the search distribution toward promising regions, showing strong robustness in high-dimensional

continuous optimization. It belongs to the class of population-based evolutionary algorithms [21]. Elite-Based Optimizer with Covariance Matrix Adaptive Reconstruction (EBO-CMAR) is a population-based evolutionary algorithm that reconstructs the covariance structure of elite individuals to generate new candidate solutions via a Gaussian sampling process. By combining statistical learning with directional variation, it maintains a balance between exploration and exploitation and performs effectively on high-dimensional continuous problems [22]. L-SHADE (Linear Population Size Reduction SHADE) is an adaptive differential evolution algorithm that employs historical memories of control parameters (F, CRF, CRF, CR) and a linear population size reduction strategy. This mechanism progressively shifts the search from exploration to exploitation while preserving diversity, ensuring superior robustness in large-scale optimization. L-SHADE belongs to the class of population-based evolutionary algorithms [23]. Runge–Kutta Optimization Algorithm (RUN) employs the Runge–Kutta numerical integration framework to guide population updates. By forming intermediate states from random differential vectors and best-mean information, RUN achieves accurate motion control and balanced exploration–exploitation. It belongs to the class of population-based evolutionary algorithms [24]. Differential Evolution (DE) generates new candidates via vector-based recombination of existing solutions, demonstrating exceptional performance in continuous optimization tasks [4]. Lévy flight distribution is a random walk mechanism based on heavy-tailed probability distributions, where most steps are small but occasional very long jumps occur. This property allows an algorithm to efficiently explore the search space, avoid premature convergence, and escape local optima, making it a powerful tool for balancing exploration and exploitation in metaheuristic optimization [25]. Population based evolutionary methods provide robust exploration capabilities, making them well suited for high dimensional and multimodal optimization problems. Nevertheless, achieving an optimal balance between exploration and exploitation remains a critical challenge. By contrast, innovative algorithms like FCO enhance this balance through adaptive population dynamics and nature-inspired mechanisms, offering a versatile approach to complex optimization challenges in fields such as engineering, data science, and operations research.

**2.2.3 Swarm intelligence based algorithms.** Swarm intelligence (SI) constitutes a biologically inspired subset of population-based metaheuristic algorithms, modeling the collective behavior of decentralized, self-organized systems [26]. These algorithms emulate the intelligent behaviors of social animals, such as ants, birds, fish, and wolves, to address complex optimization problems. Key characteristics of SI algorithms include each agent operates autonomously, guided by local information and interactions (Distributed control), complex global patterns emerge from simple, localized rules (emergent behavior) and the system dynamically adjusts to changes in the environment or search landscape (Dynamic adaptation).

Prominent SI algorithms include Particle Swarm Optimization (PSO), which mimics the social dynamics of bird flocks or fish schools, with each particle adjusting its trajectory based on individual and collective experiences [4], Ant Colony Optimization (ACO), which replicates the pheromone based path selection of ants to identify optimal solutions [27], Grey Wolf Optimizer (GWO), which models the leadership hierarchy and cooperative hunting strategies of grey wolves [5] and Gravitational Search Algorithm (GSA) is a metaheuristic algorithm inspired by Newton's law of gravity. In GSA, particles act as material objects with mass proportional to the quality of the response and attract each other. The motion of particles is calculated under the influence of gravity and acceleration [28].This algorithm has a high ability to explore complex and multidimensional spaces and avoid getting stuck in local optima. These algorithms excel in navigating multimodal search spaces, yet their performance can be limited by parameter sensitivity and convergence issues. White Shark Optimization (WSO) is a nature-inspired metaheuristic algorithm modeled on the hunting behavior of white sharks. It balances exploration and exploitation through adaptive strategies based on prey detection, pursuit, and attack mechanisms [29]. African Vultures Optimization Algorithm (AVOA) mimics the scavenging and cooperative hunting behavior of African vultures. It dynamically switches between exploratory and exploitative phases based on fitness and group coordination. Adaptive mechanism helps avoid local optima and accelerates convergence [30]. Reptile Search Algorithm (RSA) is inspired by the foraging patterns of reptiles, integrating active and passive search modes. Movements follow wave like patterns and are

guided by environmental sensing. Well suited for complex multimodal and multi objective optimization problems [9]. The Multi-Verse Optimization (MVO) algorithm is a nature-inspired metaheuristic developed based on the multiverse theory in physics. It simulates the interaction among different universes through three main operators: white holes, black holes, and wormholes.

MVO demonstrates high global search ability, fast convergence, and robustness in handling nonlinear, multimodal, and constrained optimization problems. Moreover, its simple structure and parameter settings make it highly adaptable for hybridization with other optimization algorithms in real-world engineering applications [31]. Equilibrium Optimizer (EO) is a physics-based swarm intelligence algorithm inspired by dynamic mass balance models. Each particle updates its position toward an equilibrium state using adaptive control parameters derived from exponential and random coefficients, maintaining a dynamic trade-off between exploration and exploitation. EO belongs to the class of swarm intelligence-based algorithms [32]. Manta Ray Foraging Optimization (MRFO) is a swarm intelligence algorithm inspired by the chain, cyclone, and somersault foraging behaviors of manta rays. Its adaptive control coefficient β balances exploration and exploitation by gradually decreasing over iterations. MRFO belongs to the class of swarm intelligence-based algorithms [33]. Sine Cosine Algorithm (SCA) is a simple swarm-intelligence method that updates candidate positions through sine- and cosine-driven oscillatory motions toward the best solution. The adaptive control parameter $r_1$ continuously reduces the amplitude of movement to refine convergence [34]. Sand Cat Optimizer (SCO) is a swarm-intelligence algorithm emulating the hunting and adaptive sensing behavior of desert sand cats. It alternates between a wide-range exploration phase using exponentially decreasing sensing radius and a focused exploitation phase around the best-detected prey [35]. Tunicate Swarm Algorithm (TSA) models the swarm dynamics of tunicates, marine invertebrates that move and adapt collectively. It combines group migration with individual exploratory behavior. Robust in high-dimensional spaces with fast convergence rates [36]. Whale Optimization Algorithm (WOA) is based on the bubble-net feeding behavior of humpback whales. It utilizes spiral updating and random search strategies to simulate prey encircling. Simple yet powerful, widely used for both continuous and discrete optimization tasks [6]. The Salp Swarm Algorithm (SSA) is a metaheuristic inspired by the chain-like movement of salps in the ocean, where the leader salp moves toward the food source and the followers update their positions accordingly. SSA is characterized by its simplicity, fast convergence, well balanced exploration and exploitation mechanism, and strong capability in solving complex nonlinear optimization problems [10]. In contrast, the proposed FCO algorithm leverages adaptive, nature-inspired mechanisms that dynamically balance exploration and exploitation, providing a robust and versatile framework for solving complex optimization problems across engineering, data-science, and control-system domains.

**2.2.4 Nature and behavior inspired meta heuristics.** This class of algorithms is inspired by the distinct behaviors and survival strategies observed in biological species. Unlike swarm-intelligence-based algorithms that depend on global collective interactions, these nature-inspired metaheuristics emphasize individual or localized behavioral dynamics. For instance, the Bat Algorithm (BA) emulates the echolocation capability of bats, allowing dynamic switching between global and local search phases to enhance solution discovery [37]. The Cuckoo Search (CS) algorithm is inspired by the brood parasitism behavior of cuckoos and incorporates Lévy flight patterns, effectively balancing diversification and intensification across the search space [38]. These algorithms offer task specific search strategies and have demonstrated strong performance across a range of applications, including engineering design, production scheduling, and neural network optimization. The Red Deer Algorithm (RDA), introduced by F. K. Abdul Razaque in 2020, models the mating behavior of red deer. It achieves a balance between exploration and exploitation through competitive interactions among dominant males and structured harem formation, enabling robust global search performance [39]. Quantum Invasive Optimizer (QIO) is a hybrid metaheuristic that fuses quantum-based dispersion and biologically inspired invasive motion. Each agent probabilistically switches between quantum exploration and invasive exploitation, offering strong diversity preservation and directional learning. QIO belongs to the class of nature- and behavior-inspired metaheuristics [40]. Coyote Optimization Algorithm (COA) models the social adaptation and cooperative behavior observed in coyote packs.

Each pack maintains internal learning through interactions with its mean and best members and exchanges knowledge across packs via social operators, ensuring sustained diversity and strongonvergence. It belongs to the class of nature- and behavior-inspired metaheuristics [41].

Recent research has focused on enhancing these biologically inspired techniques through hybridization (e.g., PSO-GA), fuzzy-logic integration (e.g., Fuzzy GWO), and application-specific customization. Among such developments, the FCO algorithm is a novel approach inspired by the foraging and behavioral metaphor of urban cats. It employs an adaptive behavioralframework with a clear delineation between explorer and exploiter agents, designed to minimize search noise and improve convergence reliability in complex, multimodal landscapes. Advanced mechanisms such as age-based renewal, adaptive role reassignment, and tournament-based selection further enhance its performance compared to conventional SI-based algorithms. Advanced mechanisms such as age-based renewal, adaptive role reassignment, and tournament-based selection further enhance its performance compared to conventional SI-based algorithms. As the field of nature-inspired optimization continues to evolve, algorithms like FCO highlight the growing emphasis on hybrid adaptability, structural diversity, and biologically faithful modeling. These trends point to a future where optimization strategies become increasingly aligned with the nuanced dynamics of real-world systems, offering greater efficiency and robustness across diverse problem domains. Many algorithms based on feline behavior have been proposed so far, such as: Cat Swarm Optimization (CSO) [42], Cheetah Optimization Algorithm (ChOA) [43], and Siberian Tiger Optimization Algorithm (STO) [44]. The Cat Swarm Optimization (CSO) algorithm [42], introduced by Chu and Tsai (2006), is one of the early bio-inspired metaheuristics based on feline behavior. Although similar in name to the FCO, uses completely different strategies. CSO is based on the hunting of cats. It models the natural tendency of cats to switch between two distinct modes: Seeking Mode (exploration) and Tracing Mode (exploitation). In contrast, the proposed FCO algorithm is inspired by the social and behavioral metaphor of urban cats and introduces a structured yet adaptive population division that fundamentally distinguishes it from the original CSO formulation. The population is explicitly separated into two cooperative subgroups explorers (males) and exploiters (females) each performing complementary search roles. Explorers execute broad stochastic exploration across adaptive behavioral ranges, dynamically adjusting their movement intensity according to fitness and age, while exploiters conduct fine-grained local refinement through Gaussian-based litter-burst mechanisms. Unlike CSO, in which parameters remain static throughout iterations, FCO continuously updates its exploration–exploitation ratio based on feedback from population performance and competitive selection dynamics. This adaptive adjustment, combined with the gradual dominance of elite cats that expand their influence range over time, enables FCO to maintain diversity, avoid premature convergence, and sustain high-quality search performance. Moreover, CSO does not incorporate age- or behavior-based memory, nor any explicit renewal mechanism. In contrast, FCO enhances search-space coverage through adaptive global exploration and structured movement control, whereas CSO relies on a PSO-like velocity update toward the global best. Therefore, FCO is fundamentally distinct from CSO in its population division, exploration and exploitation mechanisms, elitism, adaptive role control, renewal dynamics, and lifelong memory modeling. The Cheetah Optimization Algorithm(ChOA) [43], introduced by Akbari et al (2022), COA simulates cheetahs identifying prey and using high speed attack strategies with three phases such as targeting, chasing, and attacking. Cheetahs are randomly generated and move towards vulnerable spots with the high speed that is characteristic of this feline, then exploit and hunt the prey by changing position around it.

The Siberian Tiger Optimization Algorithm (STO) [44], introduced by Trojovsky et al (2023). STO has two main phases: Hunting and Bear fighting. In the hunting phase, the tigers first find a suitable position for hunting (Exploration) and then, approaching the target, hunt (Exploitation). In the bear fighting phase, they perform a global search operation (Attack on Bear) and try to save themselves from being trapped in local optimum points. Unlike the STO and COA, FCO is inspired by the social life of cats, different males compete with each other to attract females. Also, each female cat is associated with a specific activity range male cat, and each male has a certain number of females. Over time, males lose their effective range by weakening and eventually dying, and rival male cats take over part of the influence range of a number

   

of their female cats, depending on their strength. Also, when females become weak, they decide to reassign to powerful stronger male cats. FCO is based on structured activity ranges within the search space for males, and the exploitation and exploration operations continue dynamically over time until the end of the iterations. As can be seen in other algorithms inspired by the behavior of wild cats, the hunting behavior of that animal is usually the main goal, while in the proposed algorithm, the social life, interactions, and life cycle of the urban cat are the implementation goals. Beyond the functional classification described above, metaheuristic algorithms can also be categorized according to the primary source of their inspiration whether biological, physical, behavioral, or hybrid in nature. This complementary taxonomy provides a broader conceptual understanding of each algorithm's metaphorical origin, highlighting how different natural or physical analogies lead to distinct search dynamics and parameter-control strategies. Table 2 summarizes this dual-layer classification, grouping the sixteen compared algorithms with respect to both their function-based category (as discussed in Section 2.2) and their inspiration type.

In summary, the comparative review presented in this section demonstrates the remarkable diversity of metaheuristic design philosophies, ranging from purely mathematical adaptation to deeply biological and hybrid inspirations. While most existing algorithms capture isolated aspects of exploration or exploitation, few explicitly model structured social hierarchies or behavioral dynamics that promote sustained diversity over time. This conceptual gap motivates the development of the Felis Catus Optimization (FCO) algorithm a novel framework inspired by the spatial and social organization of urban domestic cats. FCO translates the naturally balanced interaction between independent exploration and cooperative exploitation into a computational model, aiming to establish a self-adaptive and reproducible metaheuristic capable of maintaining search diversity, accelerating convergence, and enhancing solution accuracy in complex high-dimensional problems.

## 3. Felis Catus Optimization (FCO)

The proposed Felis Catus Optimization (FCO) algorithm draws its conceptual foundation from the complex ecological and social organization of urban domestic cats (*Felis catus*). Unlike pack-oriented species, these felines inhabit semi-structured colonies where individuals balance solitary behavior with loosely defined behavioral ranges. Each cat alternates between roaming to discover new resources and returning to familiar safe areas, continuously adjusting activity according to age, dominance status, and environmental stimuli. This equilibrium between independent exploration and localized exploitation provides the biological blueprint for the algorithm's dual-role framework, dividing the population into complementary explorer (male) and exploiter (female) agents. Explorers perform adaptive global exploration that emulates male cats' adaptive roaming and competitive defense, while exploiters conduct noise-driven local refinement analogous to female cats' precise foraging within familiar localities. Beyond this dual-role structure, FCO integrates an ecological life cycle encompassing aging, adaptive sex-ratio adjustment, litter-burst reproduction, and diversity restoration through environmental rejuvenation rather than explicit renewal across discrete scales. These mechanisms collectively maintain

**Table 2. Summary taxonomy of metaheuristic algorithms by primary inspiration and mechanistic domain. Sixteen algorithms are categorized by their dominant source of inspiration: animal/behavioral, physical-process, evolutionary/mathematical, or hybrid and by their underlying search mechanism. This classification illustrates how metaphoric origin guides exploration–exploitation balance and population dynamics.**

| Primary Inspiration Category | Representative Algorithms | Dominant Mechanistic Traits |
|---|---|---|
| Animal-Inspired (Biological Behavior) | FCO, COA, GWO, PSO, WOA, SCO, SSA, RSA | Collective or spatial behaviors, predator–prey dynamics, and adaptation through social or spatial interaction. |
| Physical-Process-Inspired | EO, SCA, RUN | Laws of motion, oscillation, or equilibrium modeling controlling position updates. |
| Evolutionary/ Genetic/ Mathematical-Adaptation | CMA-ES, EBO-CMAR, L-SHADE | Population evolution via mutation, selection, and covariance adaptation; mathematically governed stochastic sampling. |
| Hybrid or Complex Nature/ Quantum-Based Systems | QIO, COA | Combined biological and physical metaphors; adaptive parameter control and quantum/migration mechanisms. |

population diversity, reinforce elite solutions, and reproduce natural ecological renewal processes, thereby enabling FCO to achieve a resilient balance between exploration and exploitation in high-dimensional and multimodal optimization. One of the distinctive features of urban cats is their persistent and thorough behavioral exploration. Male cats continuously monitor their entire areas and do not disregard areas simply because they lacked food, mating opportunities, or other stimuli in previous visits [45]. As soon as an attraction appears in these previously uninteresting areas of the search space, cats instinctively return, demonstrating a highly adaptive search strategy. This behavior is so strategically significant that it has even inspired what is known in marketing as the cat strategy. In this context, marketers argue that just because an area or customer base has not responded previously does not mean future visits will yield the same result new opportunities may arise. This adaptive and opportunity driven behavior serves as a central inspiration for FCO. By modeling this principle, FCO enables its agents to dynamically reassess and revisit fewer promising scale of the search space when new potential emerges, enhancing its capacity to avoid premature convergence and maintain a balance between exploration and exploitation in complex optimization scenarios.

FCO is inspired by the complex ecological and social behaviors of urban domestic cats, particularly the adaptive behavioral dynamics and reproductive strategies of both male and female individuals in densely populated environments. This semi structured yet adaptive social system provides a rich and biologically grounded metaphor for balancing exploration and exploitation, two essential components of metaheuristic search strategies [45].

In natural urban ecosystems, each dominant male cat typically maintains broader roaming ranges, the size and quality of which are directly linked to his strength, health, and hierarchical status [46]. These territories function not only as foraging grids but also as centers of reproductive influence. A dominant male may have control over a group of four to ten female cats within his domain, with the number of females reflecting his physical and social dominance. The females, while generally loyal to a single dominant male, display limited mobility and tend to remain within localized areas near food sources or shelters. This behavior highlights an intensive, localized search strategy that closely parallels the concept of exploitation in optimization processes [45]. In FCO, this behavioral structure is reflected in the clear separation between global (male) and local (female) search agents, supporting a robust balance between exploration and exploitation.

As shown in Fig 1, the four illustrated zones serve only as a biological metaphor to depict how dominant males and candidate females interact in natural urban cat communities. These conceptual sketches help visualize the ecological inspiration behind FCO but do not represent literal spatial divisions in the algorithmic model. In Fig 1A, each conceptual scale contains a male cat (explorer). Based on its power, which is a combination of the fitness function and age, this male cat has been able to attract a number of female cats (exploiters). Male cats are shown in black and female cats in yellow.

As time passes in Fig 1B, the male cat in conceptual scale 2 (Z2) ages, its power also decreases and this cat dies. When the male cat disappears, female cats are attracted to the range of influenceof rival male cats. Out of the 5 female cats in conceptual scale 2 (Z2), three cats are attracted to conceptual scale 4 (Z4). Also, the conceptual scale 3 cat is constantly increasing its range of influenceand attracting more females, so the 2 remaining cats in conceptual scale 2 (Z2) are attracted to it.

But as you can see, the conceptual scale 1 (Z1) cat is also weakening, and although it still maintains its territory, it has lost one of its females, and this female has also been attracted to the rival cat in conceptual scale 3 (Z3). In the next iteration, the main competition will be between conceptual scales 3 and 4 (Z3,Z4), unless the conceptual scale 1 cat can suddenly (mutationally) compete and attract the females of the rivals with its new power.

Male cats, in contrast, exhibit broad and adaptive roaming behavior, continuously scanning their surroundings and adjusting their movements in response to competing males. This behavior provides a natural metaphor for global exploration mechanisms in optimization. As competing males grow stronger, they may challenge the dominant male, leading to dynamic competition and a restructuring of influence and mating opportunities. This adaptive and competitive process inspires the tournament-based selection and role-adjustment strategies employed in FCO. Notably, weaker or aging males gradually lose influence and access to mates, ultimately being displaced or reinitialized through role reassignment. Such

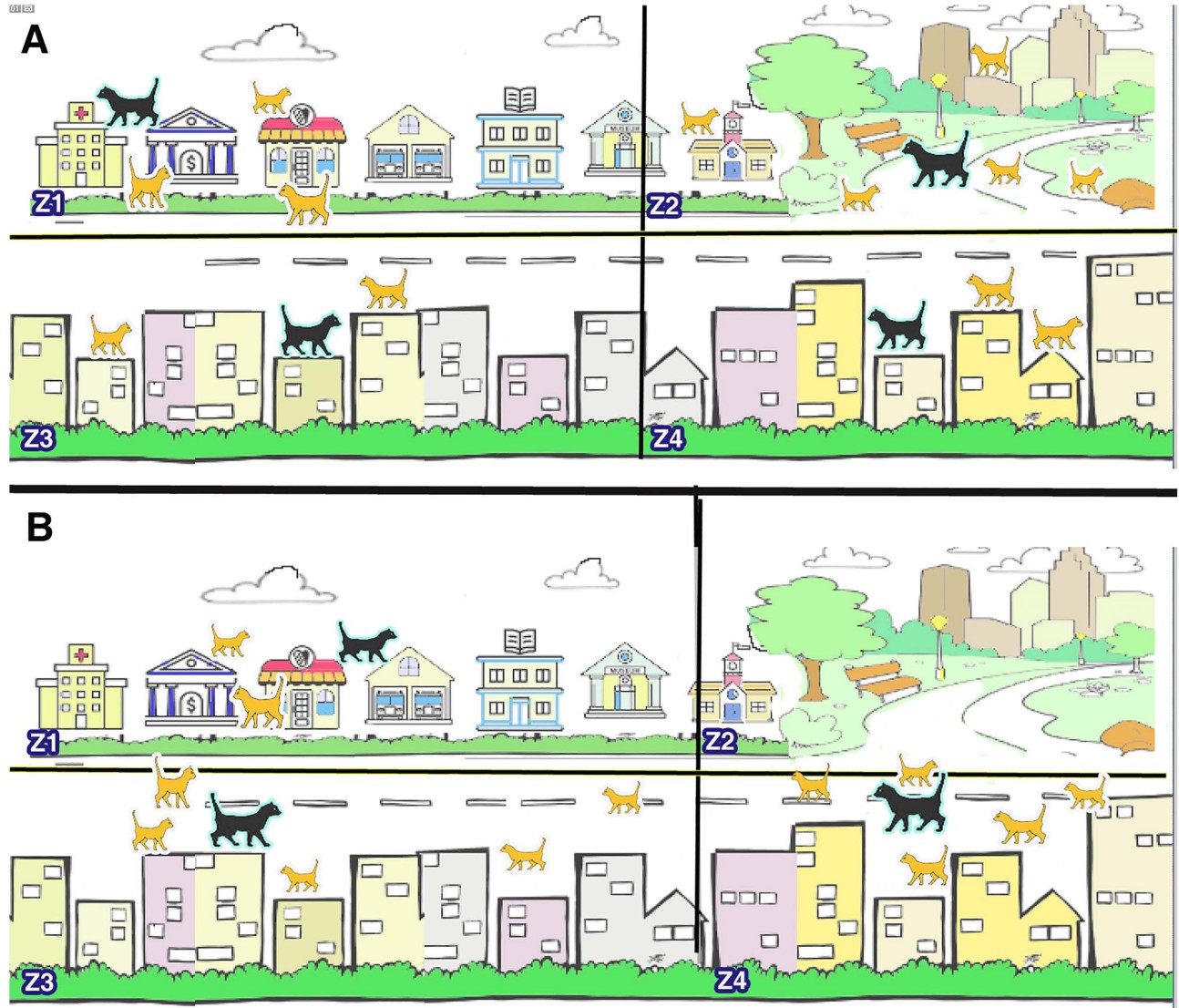

**Fig 1. Positions of explorers and exploits in continuous space.**

mechanisms preserve population diversity, prevent premature convergence, and promote sustained exploration, thereby improving the algorithm's ability to converge on high-quality global optima.

FCO operationalizes this ecological model by dividing the population into two distinct agent roles: explorer males and exploiter females. Male agents perform broad, global exploration by dynamically traversing large areas of the search space. Their movement is governed by fitness-based competition (analogous to physical strength) and is reinforced through competitive selection dynamics and dynamic dominance reassessment. In parallel, female agents carry out fine grained local search around high potential scales representing food sources while adaptively adjusting their association based on the dominance of candidate males. The interaction between these two roles, driven by mechanisms such as reproductive success (solution refinement) and behavioral defense (solution elitism), enables FCO to maintain a stable yet flexible search process. This structure supports both exploration and exploitation in a coordinated manner, making FCO particularly effective for solving complex, multimodal optimization problems.

## 3.1 Mathematical formulation of the FCO algorithm

FCO simulates the behavioral, social, and survival behaviors of urban male and female cats within an optimization framework. The algorithm operates on a multi-dimensional continuous search space, incorporating structured rules for exploration, exploitation, mating control, and population dynamics. The following section presents a formal, modular mathematical representation of the algorithm, outlining its key components and mechanisms.

### 3.1.1 Population initialization.

Let the search space be defined as an n dimensional real vector space, $R^n$. Within this space, a population of $N$ agents, referred to as cats, is initialized. This population is partitioned into male and female subgroups according to predefined ratios or adaptive criteria. Formally, the population is represented as:

$$P = \{C_1, C_2 \ldots C_N\}, \text{ where } C_i = (x_i, sex_i, age_i, f_i) \tag{1}$$

Here, each cat $C_i$ is characterized by:

- $x_i \in R^n$: The position vector of the cat within the search space, representing a candidate solution to the optimization problem.

- $sex_i \in \{male, female\}$: The biological sex of the cat, which determines its behavioral role in the algorithm (exploration or exploitation).

- $age_i \in N$: The discrete age of the cat, used to model lifecycle dynamics such as maturity, aging, and replacement.

- $f_i = f(x_i)$: The fitness value of the cat, evaluated by the objective function, : $R^n \rightarrow R$, which quantifies the quality of the candidate solution.

The initialization ensures a diverse starting population by sampling all position vectors uniformly within the specified bounds of the optimization problem, followed by computing their corresponding fitness values.

### 3.1.2 Continuous search space representation.

In FCO, the spatial configuration of the population is defined directly in the continuous n-dimensional search space bounded by lower (lb) and upper (ub) limits. Each agent is represented by a coordinate vector:

$$G = \left\{ g_i \mid i \in \{1, \ldots, N\} \right\} \tag{2}$$

Where $g_i \in R^n$ denotes the position vector for agent i in the search space, and N is the total population size. This formulation treats all agents as operating within a unified continuous domain without discrete partitioning.

Influence limits between male and female agents are dynamically determined from their fitness values using:

$$k_i = \left\lfloor \left( \frac{f_{max} - f_i}{f_{max} - f_{min}} \right) \cdot k_{max} \right\rfloor \tag{3}$$

Where:

- $f_i$ is the fitness value of male $i$,

- $f_{max}$ and $f_{min}$ are the maximum and minimum fitness values in the current population,

- $k_{max} \in [4,10]$ is the maximum allowable number of females a single male can dominate,

- $\lfloor . \rfloor$ Ensures an integer value.

This mechanism ensures that fitter males are rewarded with broader influence (larger harems), aligning with the natural mating behavior observed in feline species. It also serves to reinforce selective pressure within the algorithm by

linking reproductive control to solution quality, thereby promoting the propagation of high-quality solutions throughout the population.

   **3.1.3 Exploration behavior (male cats).** In FCO, male cats drive asynchronous global exploration across wide areas of the continuous search space, dynamically adjusting their adaptive search radius according to age and fitness. Their movement is guided by three primary influences: attraction toward the globally best male (directed motion), exploration step-size scaling driven by the dynamic radius $R_i^t$, and triplet interaction among candidate males to sustain asynchronous exploration. The position update rule for a male cat $i$ at iteration $t$ is defined as:

$$x_i^{t+1}=x_i^t+\alpha_i^t\left(x_{best}^t-x_i^t\right)+\beta_i^t R_i^t N(0, I)+\delta_i^t(x_{i-1}^t-x_{i+1}^t)x_i^{t+1}=x_i^t+\alpha.\left(x_{best}-x_i^t\right)+\beta.N(0, I) \tag{4}$$

Where:

- $\alpha, \beta, \delta \in R^+$: Adaptive coefficients controlling directional attraction, behavioral expansion, and triplet influence respectively. $x_{best} \in R^n$: The position of the best performing cat in the current population,

- $x_i^t \in R^+$: The current position of male cat $i$,

- $N(0, I)$ : A standard multivariate Gaussian noise vector with zero mean and identity covariance matrix, simulating local randomness.

- In practice, the δ-term's effect is realized implicitly through selecting among the nearest males, which captures the triplet influence without an explicit δ-vector.

This update rule encodes the asynchronous Triplet-Move mechanism of FCO, where males shift among neighboring scales with adaptive step sizes. The dynamic radius $R_i^t$ expands influence ranges proportionally to strength, while stochastic perturbations maintain diversity. The joint control of $\alpha_i^t$, $\beta_i^t$ and $\delta_i^t$ ensures balanced global search.

   **3.1.4 Exploitation behavior (female cats).** In the proposed nature-inspired metaheuristic algorithm, female cats perform adaptive local exploitation driven by mating and litter-burst mechanisms, concentrating around promising areasor locally dominant males rather than fixed centers. This behavior is mathematically expressed as:

$$x_j^{t+1}=x_j^t+\gamma.N\left(I\sigma^2, 0\right)+\eta(L_j^t-x_{best}^t) \tag{5}$$

Where:

- $\gamma$ is the exploitation scaling factor that modulates the intensity of the local search,

- $\sigma$ is the standard deviation governing the spread of the Gaussian distribution and thereby controlling the extent of local exploration,

- $N(I\sigma^2, 0)$ Denotes a multivariate normal distribution with zero mean and a covariance matrix $I\sigma^2$, ensuring isotropic local search behavior.

- η is the adaptive litter-burst coefficient determining newborn influence strength.

- $L_j^t$ is the mean position of offspring generated around female j through litter-burst reproduction.

Fig 2 shows the exploration and exploitation dynamics of the FCO algorithm: Left, male cats perform global exploration toward the best solution with stochasticity, Right, female cats conduct local exploitation via Gaussian sampling around current positions. So:

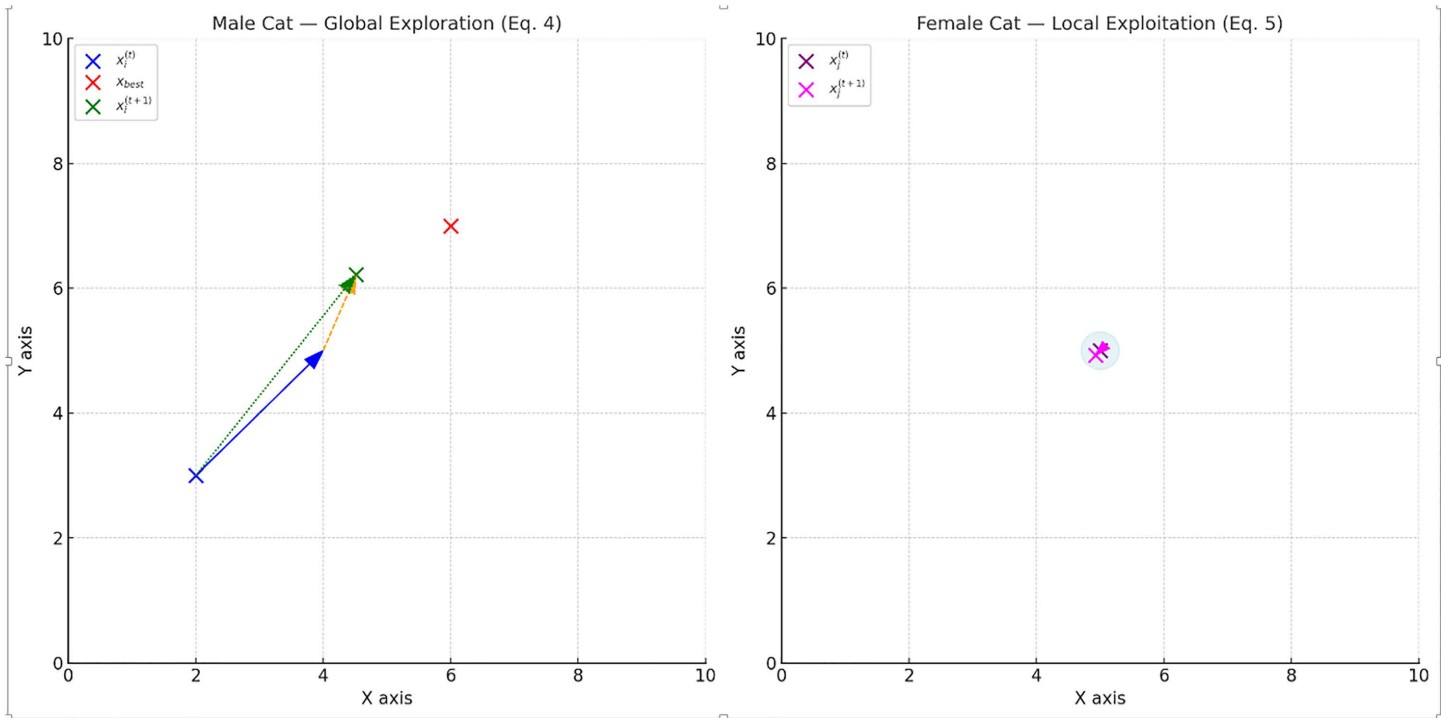

**Fig 2. Visualization of exploration and exploitation mechanisms in FCO algorithm.**

- **Left: Male Cat Global Exploration (Eq. 4).**

This diagram illustrates the global exploration behavior of male agents. Each male cat moves toward the globally best solution, guided by an attraction term scaled by α, combined with a stochastic component based on Gaussian noise scaled by β. This mechanism allows male cats to explore distant areas of the search space while maintaining diversity.

- **Right: Female Cat Local Exploitation (Eq. 5)**

This diagram depicts the adaptive local exploitation behavior of female agents in FCO. Each female generates a cluster of offspring (litter-burst) around its current position using Gaussian sampling, where the spread is modulated by σ and the influence toward the global best is tuned by η. This mechanism allows fine-grained exploitation of high-fitness scales while ensuring continuous population renewal and diversity across the search space..

This formulation models the Mating & Litter-Burst process of FCO, enabling females to generate offspring around promising around promising areas, locally replace weaker individuals, and sustain adaptive exploitation. By tuning γ, σ, and η, the algorithm maintains a balanced intensification–diversification cycle, yielding robust search efficiency and high reproducibility.

**3.1.5 Competitive selection dynamics.** In the proposed nature-inspired metaheuristic algorithm, a fitness-based competitive selection mechanism governs male dominance dynamics within the population. Specifically, if a male, denoted as $C_m$, shows greater fitness than the current dominant male according to the evaluated performance criteria, $M_i$, it initiates a challenge for control. The outcome of this challenge is determined as follows:

- Let $f_m$ represent the fitness of the challenger male $C_m$, and $f_{M_i}$ denote the fitness of the current dominant male $M_i$.

- If $f_m < f_{M_i}$ indicating that $C_m$ has better fitness (assuming a minimization problem where lower fitness values are preferable), the challenger male $C_m$ supplants $M_i$.

- Upon the replacement of the dominant male, the females previously associated with the defeated male are reassigned among candidate males in proportion to their dominance scores. A small environmental noise $\varepsilon \sim \mathbf{N}(\mathbf{0}, \sigma_R^2)$ is also injected to slightly perturb local behavioral ranges, promoting adaptive local-range adjustment.

This competitive mechanism emulates natural processes observed in certain social and biological systems, where dominance hierarchies are established through fitness based challenges. By incorporating such dynamics, the algorithm promotes the exploration of diverse solutions while maintaining a balance between exploitation and diversity in the search process. The redistribution of females following a successful challenge further enhances the algorithm's adaptability, as it facilitates the reallocation of resources (i.e., female individuals) to potentially more promising scales of the solution space. This approach not only mirrors ecological and evolutionary principles but also contributes to the algorithm's robustness and efficiency in tackling complex optimization problems.

### 3.1.6 Aging and elimination.
In the proposed nature-inspired metaheuristic algorithm, the age of each cat, denoted as $age_i$:

$$age_i \rightarrow age_i + 1 \tag{6}$$

However, a cat $C_i$ is eliminated from the population under specific conditions:

- If $age_i > age_{maxi} > age_{max}$, indicating that the cat has exceeded the maximum allowable age.

Upon elimination, the cat $C_i$ is replaced by a new cat, initialized with random attributes according to the algorithm's initialization procedure.

This aging mechanism simulates natural lifecycle dynamics, ensuring population turnover and preventing stagnation in the search process. By removing cats that exceed the maximum age, the algorithm maintains diversity and introduces fresh solutions into the population. The replacement with randomly initialized cats further promotes exploration of the solution space, enhancing the algorithm's ability to escape local optima and adapt to complex optimization landscapes.

### 3.1.7 Environmental rejuvenation and noise-driven reallocation.
To maintain population diversity and prevent stagnation, FCO employs two complementary mechanisms: Environmental Noise Injection (ENI) and Random Rejuvenation (RJ). In the revised continuous version of the algorithm, the discrete renewal behavior used in earlier prototypes is replaced by a smooth, population-wide rejuvenation process that operates directly in the continuous search space and no longer depends on predefined spatial scales or cell-to-cell transfers.

**(a) Environmental Noise Injection (ENI)**

When the population diversity Div(t) falls below a small threshold, a mild random perturbation is applied to the position of every agent:

$$X_i \leftarrow X_i + \varepsilon, \ \textbf{where } \varepsilon \textbf{ is a small random fluctuation}. \tag{7}$$

This noise helps the population escape locally stagnant configurations and encourages exploration of candidate areas in the continuous search space without disrupting the overall convergence pattern.

**(b) Random Rejuvenation (RJ)**

If an agent exceeds the maximum allowable age ($age_i > age_{max}$), it is rejuvenated by replacing its position with a newly generated one sampled uniformly within the search bounds:

$$X_i \leftarrow \textbf{lb} + \textbf{rand}\left(\mathbf{1}, \textbf{dim}\right) \cdot (\textbf{ub} - \textbf{lb}) \tag{8}$$

This mechanism introduces fresh solutions while maintaining the algorithm's exploratory capacity, acting as a lightweight population renewal operator. Together, ENI and RJ preserve diversity, prevent premature stagnation, and maintain a balanced exploratory–exploitative behavior throughout the optimization process

This rejuvenation reintroduces exploratory momentum while preserving valuable knowledge gained from prior iterations, ensuring continuous population renewal. The process ensures both global exploration (driven by male agents moving toward $X_{best}^t$) and local exploitation (by female agents within ENI-perturbed scales), achieving an adaptive homeostatic balance between stability and innovation in the evolving search landscape (See Fig 3).

**3.1.8 Algorithm flow summary.** The pseudo code of the FCO algorithm is presented in Fig 4.

Overall, the proposed **Felis Catus Optimization (FCO v2)** framework establishes a fully continuous and reproducible metaheuristic model, characterized by adaptive adaptive behavioral dynamics, environmental noise injection, and a rejuvenation-based population renewal process. These mechanisms collectively ensure sustained diversity, stable

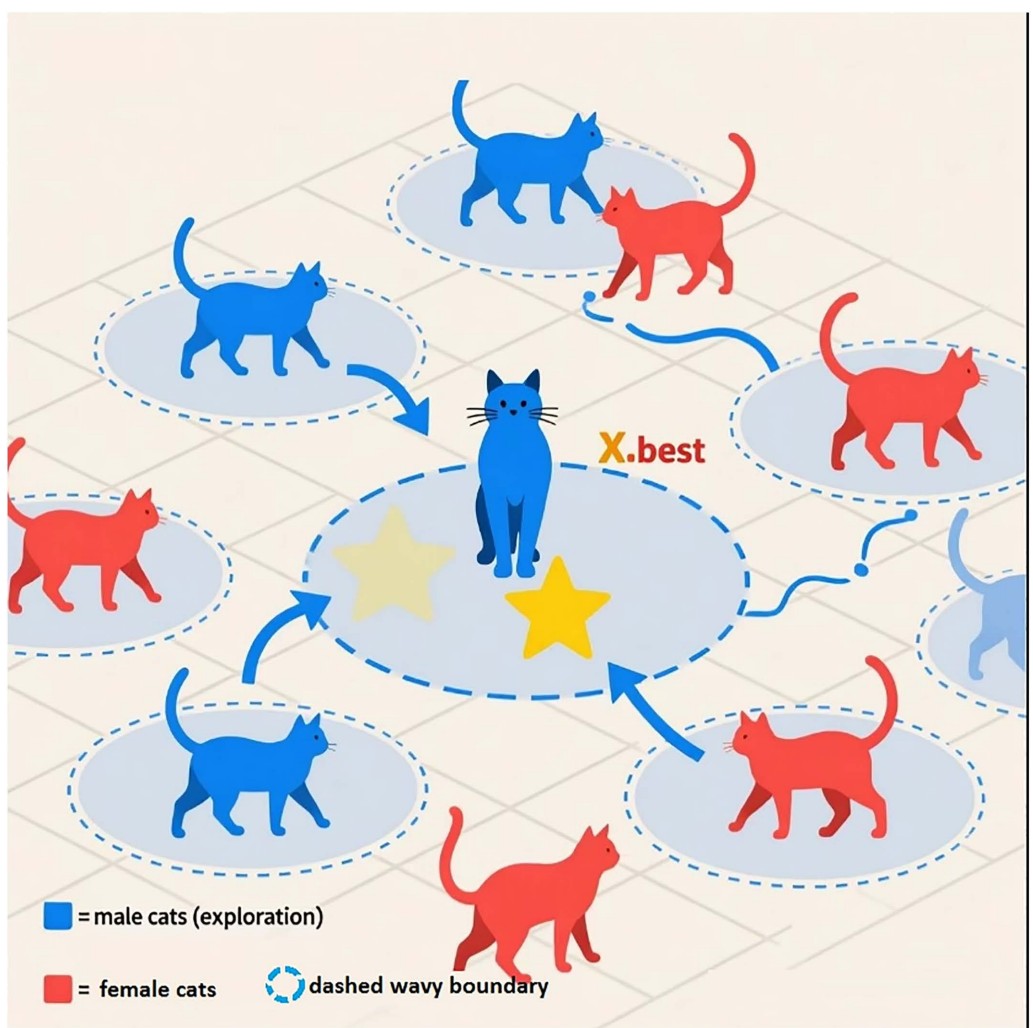

**Fig 3. Schematic representation of noise-driven rejuvenation and reallocation in FCO.** Environmental Noise Injection (ENI) introduces mild random perturbations to agent positions to maintain population diversity, while the Random Rejuvenation (RJ) mechanism replaces aging agents with newly initialized ones within the search bounds, serving as a continuous alternative to discrete renewal events.

| **Algorithm** The pseudo code of the **Felis Catus Optimization** (FCO) |
| --- |

Input: Population size $P$, Maximum iterations $Max_{itr}$, Age threshold $age_{Max}$
Output: Best solution found

1. Initialize:

   Initialize population $P$ with random male and female cats by ratio ρ_m
   Set initial ages for all cats to 0
   Evaluate and store initial fitness values

2. For iteration = 1 to $Max_{itr}$
   // Male behavior phase
   For each male $m$ in $P$
      Perform exploration:
   •      Update position using either a nearest-male direction or the current best male
   • Apply adaptive step-size scaling (α)
   • Apply small random displacement with R (dynamic search radius)
   • Evaluate new fitness
   • If improved → accept and reset age
   Else → increase age
   Update dynamic search radius R based on age

   End For

   // Female behavior phase
   For each female $f$ in $P$
      Perform local exploitation:
   •      Move toward the best candidate female with step λ_FS
   • Apply Gaussian perturbation
   • Evaluate new fitness
   • If improved → accept and reset age
   Else → increase age

   End For

   // Population management
   Apply Rejuvenation (RJ):
      For each cat c in population
         •      Increase age of c
         •      If age > maxAge or search radius becomes too small
         → Rejuvenate c randomly within bounds
            Reset age to zero
         End If
      End For
   Apply Environmental Noise Injection (ENI):
   •   If population diversity becomes low
         ○   Add mild random noise to all positions
         ○   Restore exploration and prevent stagnation
         End If

   // Update best solution
      If new position improves fitness
         Update global best record
      End If

   3. Return best solution
   End

**Fig 4. Pseudo code of the FCO algorithm.**

convergence behavior, and robustness under double-precision implementation. The complete MATLAB source code and supplementary benchmarking scripts of FCO are publicly archived in the Zenodo repository for verification and replication purposes Zenodo. DOI: https://doi.org/10.5281/zenodo.18393892

## 3.2 Computational complexity and runtime analysis

The computational complexity of the proposed Felis Catus Optimization (FCO) is determined by analyzing the dominant operations within one full iteration across all agents.

Let N be the population size, D be the dimensionality of the problem, and T the total number of iterations.

### 1. Initialization Stage:

Random generation and fitness evaluation for all N agents require $O(N \times D)$ operations.

### 2. Exploration and Exploitation Phases:

During each iteration, the position update of male and female agents involves arithmetic, Gaussian sampling, and fitness evaluation. The overall per-iteration cost is dominated by the objective-function evaluations, leading to a complexity of $O(N \times D)$ per iteration.

### 3. Population Management (ENI and RJ mechanisms):

The Environmental Noise Injection (ENI) introduces $O(N)$ perturbation updates, while the Rejuvenation (RJ) mechanism selectively re-creates a small fraction p of the population (typically $p < 0.05$), thus adding $O(p \times N \times D) \approx O(N \times D)$ in the asymptotic sense.

Accordingly, the total computational complexity of FCO is expressed as:

$$O_{FCO} = O(T \times N \times D) \tag{9}$$

Table 3 summarizes the average function-wise runtime (in seconds) of 18 metaheuristic algorithms on the CEC 2005 and CEC 2017 benchmark suites. For each benchmark, the runtime of every function was measured as the mean of 30 independent runs and then averaged across all benchmark functions (17 for CEC 2005 and 30 for CEC 2017). All experiments were performed in MATLAB R2019a (64-bit) on Windows 7 (Intel Core i5-3230M, 2.60 GHz, 4 GB RAM) with identical search budgets (population size = 80, iterations = 1000).

Despite exhibiting a higher runtime on CEC 2017 (155.87 s), the FCO algorithm demonstrates competitive computational efficiency on CEC 2005 (12.52 s, ranked 4th among 18 methods). Since FCO's computational complexity remains linear ($O(T \times N \times D)$), the greater processing time observed in CEC 2017 is not caused by intrinsic algorithmic overheads but instead by MATLAB-level runtime factors particularly Just-In-Time (JIT) compilation warm-up, parallel-pool initialization (parpool), and memory reallocation during the first iterations of large-scale problems.

Statistical analyses on CEC 2017 (Tables 13 and 14) show that FCO achieved a mean Friedman rank of 5.50 across 30 functions and outperformed seven algorithms (SSA, SCA, WOA, CMAR, COA, EO, and QIO) with statistically significant differences under the Holm post-hoc test ($p < 0.05$). This evidences that the extended runtime cost is compensated by substantially better convergence stability and accuracy. Consequently, FCO achieves an equitable trade-off between computational load and optimization quality.

Its slightly higher runtime principally due to fixed environmental overhead yields demonstrably superior optimization consistency and robustness, consolidating FCO's position among the top-performing metaheuristics in the CEC 2017 benchmark suite.

**Table 3. Average runtime (seconds) of 18 metaheuristic algorithms on CEC 2005 and CEC 2017.**

| Alg | CEC 2005 | CEC 2007 |
|---|---|---|
| FCO (Proposed) | **12.522** | **155.87** |
| ChOA | 2.262 | 25.32 |
| CMA-ES | 55.708 | 23.95 |
| CMAR | 124.325 | 48.16 |
| COA | 82.594 | 39.14 |
| DE | 76.891 | 29.71 |
| EO | 69.557 | 25.97 |
| GWO | 69.374 | 29.59 |
| L_SHADE | 43.075 | 17.62 |
| MRFO | 75.44 | 30.17 |
| PSO | 67.914 | 26.69 |
| QIO | 95.754 | 43.77 |
| RSA | 63.722 | 24.69 |
| RUN | 88.997 | 42.21 |
| SCA | 60.088 | 22.02 |
| SCO | 64.082 | 24.17 |
| SSA | 60.266 | 21.51 |
| WOA | 58.849 | 20.57 |

## 4. Results and discussion

To comprehensively evaluate the robustness and adaptability of the proposed FCO algorithm, two well-established benchmark suites CEC 2005 and CEC 2017 were employed. The CEC 2005 set contains a group of classical single-, multimodal, and composition functions that continue to serve as a reference standard in metaheuristic performance benchmarking. In contrast, the CEC 2017 suite introduces higher dimensionality, rotation, shifting, and hybrid composition functions, representing a more challenging and modern evaluation environment. Using both collections ensures cross-generational consistency and allows examination of FCO's stability as benchmark complexity evolves. This dual comparison further highlights the algorithm's generalization ability: while several established methods (e.g., WOA, CMA-ES, DE) show rank deterioration from 2005 to 2017, FCO maintains a stable mid-top position, confirming its resilience to changing landscape geometry and dimensional scaling. Therefore, the inclusion of both test suites provides a rigorous and context-rich validation of FCO's robustness and cross-benchmark reliability.

### 4.1 Benchmark test functions

In this section CEC 2005 Benchmark Test Functions and CEC 2017 Benchmark Test Functions are evaluated,

**4.1.1 CEC 2005 benchmark test functions.** Table 4 (Unimodal benchmark functions) presents the mathematical definitions, dimensionality, search boundaries, and global optima for seven standard unimodal benchmark functions (F1–F7). These functions are characterized by a single global minimum and no local optima, making them ideal for evaluating the exploitation ability of optimization algorithms. Because the search space is smooth and convex, an effective optimizer should efficiently converge to the global minimum. These functions serve as foundational tools for testing convergence speed and precision. Each entry in the table includes the function's formula and relevant parameters for reproducibility.

Table 5 (Multimodal benchmark functions) includes five multimodal benchmark functions (F8–F12) that are commonly used to assess the exploration capabilities of optimization algorithms. These functions feature numerous local minima scattered throughout the search space, creating challenges for avoiding premature convergence. Evaluating performance

**Table 4. Unimodal benchmark functions.**

| Function | | Formula | Dim | Range | $f_{min}$ |
|---|---|---|---|---|---|
| F1 | Sphere | $f_1(x) = \sum_{i=1}^{n} x_i^2$ | 30 | [−100,100] | 0 |
| F2 | Schewfel 2.22 | $f_2(x) = \sum_{i=1}^{n} |x_i| + \prod_{i=1}^{n} |x_i|$ | 30 | [−10, 10] | 0 |
| F3 | Schewfel 1.12 | $f_3(x) = \sum_{i=1}^{n} \left(\sum_{j-1}^{i} Z_j\right)^2$ | 30 | [−100,100] | 0 |
| F4 | Schwefel 2.21 | $f_4(x) = \max\{|x_i|, 1 \le i \le n\}$ | 30 | [−100,100] | 0 |
| F5 | Rosenbrock | $f_5(x) = \sum_{i=1}^{n-1} [100(x_{i+1} - x_{ij}^2)^2 + (x_i - 1)^2]$ | 30 | [−30, 30] | 0 |
| F6 | Step | $f_6(x) = \sum_{i=1}^{n} ([x_i + 0.5])^2$ | 30 | [−100,100] | 0 |
| F7 | Quratic | $f_7(x) = \sum_{i=1}^{n} i x_i^4 + random[0.1)$ | 30 | [−1.28,1.28] | 0 |

**Table 5. Multimodal benchmark functions.**

| Function | | Formula | Dim | Range | $f_{min}$ |
|---|---|---|---|---|---|
| F8 | Rastrigin | $f_8(x) = \sum_{i=1}^{n} [x_i^2 - 10\cos(2\pi x_i) + 10]$ | 30 | [-5.12,5.12] | 0 |
| F9 | Ackley | $f_9(x) = -20\exp\left(-0.2\sqrt{\frac{1}{n}\sum_{i=1}^{n} x_i^2}\right) - \exp\left(\frac{1}{n}\sum_{i=1}^{n}\cos(2\pi x_i)\right) + 20 + e$ | 30 | [−32, 32] | 0 |
| F10 | Greiwank | $f_{10}(x) = \frac{1}{4000}\sum_{i=1}^{n} x_i^2 - \prod_{i=1}^{n}\cos\left(\frac{x_i}{\sqrt{i}}\right) + 1$ | 30 | [−600,600] | 0 |
| F11 | Penalized function 1 | $f_{11}(x) = \frac{\pi}{n}\left\{10\sin(\pi y_1) + \sum_{i=1}^{n-1}(y_i - 1)^2[1 + 10\sin^2(\pi y_{i+1})] + (y_n - 1)^2\right\} + \sum_{i=1}^{n} u(x_i, 10, 100, 4)$ $y_i = 1 + \frac{x_i+1}{4}$ $U(x_i, a, k, m) = \begin{cases} k(x_i - a)^m & x_i > a \\ 0 & -a < x_i < a \\ k(-x_i - a)^m & x_i < -a \end{cases}$ | 30 | [−50, 50] | 0 |
| F12 | Penalized function 2 | $f_{12}(x) = 0.1\left\{\sin^2(3\pi x_i) + \sum_{i=1}^{n}(x_i - 1)^2[1 + \sin^2(3\pi x_i + 1)] + (x_n - 1)^2[1 + \sin^2(2\pi x_n)]\right\} + \sum_{i=1}^{n} u(x_i, 5, 100, 4)$ | 30 | [−50, 50] | 0 |

on these functions helps determine the algorithm's ability to escape local traps and locate the global optimum. Each function varies in modality and topological complexity, making them suitable for diverse test scenarios. The table provides detailed formulations and ranges to facilitate direct comparison and implementation.

Table 6 (Fixed-dimension multimodal benchmark functions) lists benchmark functions F13–F17, which are multimodal but defined over fixed dimensions. These functions are frequently used to evaluate an algorithm's performance on problems with known dimensional constraints and complex fitness landscapes. Their fixed nature allows for consistent benchmarking, especially in terms of accuracy, convergence behavior, and stability. These functions often simulate real-world optimization challenges in engineering or system design. Detailed function descriptions, along with their bounds and known optima, are included for precise evaluation.

Fig 5 presents the surface plots of the unimodal benchmark functions F1–F7, offering visual insight into their mathematical landscapes. These functions are used to assess the exploitation strength of optimization algorithms, as they each contain a single global minimum. The plots illustrate varying degrees of convexity, smoothness, and complexity. While some surfaces, like F1, are perfectly smooth and symmetric, others such as F5 and F7 display more irregular or noisy features, making convergence slightly more challenging. Overall, these visualizations help to intuitively understand the structural behavior of the functions and how an algorithm might navigate their respective topographies.

Fig 6 illustrates the two-dimensional surface plots of the multimodal benchmark functions F8–F12, highlighting the presence of multiple local optima distributed across the search space. These complex landscapes are intentionally designed to evaluate the global search capability of optimization algorithms by challenging them to escape from local traps and identify

**Table 6. Fixed dimension multimodal benchmark functions.**

| Function | | Formula | Dim | Range | $f_{min}$ |
|---|---|---|---|---|---|
| F13 | Shekel's foxholes | $f_{13}(x) = \left( \frac{1}{500} + \sum_{j=1}^{25} \frac{1}{j + \sum_{i=1}^{2} (x_i - a_{ij})^6} \right)^{-1}$ | 4 | [−65, 65] | 1 |
| F14 | Gear Train Problem | $f_{14}(x) = \sum_{i=1}^{n} \left[ a_i - \frac{x_1(b_i^2 + b_i x_2)}{b_i^2 + b_i x_3 + x_4} \right]^2$ | 4 | [−5, 5] | 0.00030 |
| F15 | Six hump Camel | $f_{15}(x) = 4x_1^2 - 2.1x_1^4 + \frac{1}{3}x_1^6 + x_1 x_2 - 4x_2^2 + 4x_2^4$ | 2 | [−5, 5] | 1.0316 |
| F16 | Branin | $f_{16}(x) = \left( x_2 - \frac{5.1}{4\pi^2} x_1^2 + \frac{5}{\pi} x_1 - 6 \right)^2 + 10 \left( 1 - \frac{1}{8\pi} \right) \cos x_1 + 10$ | 2 | [−5, 5] | 0.398 |
| F17 | Goldstein Price | $f_{17}(x) = \left[ 1 + (x_1 + x_2 + 1)^2 (19 - 14x_1 + 3x_1^2 - 14x_2 + 6x_1 x_2 + 3x_2^2) \right]$ $\times \left[ 30 + (2x_1 - 3x_2)^2 \times (18 - 32x_1 + 12x_1^2 + 48x_2 - 36x_1 x_2 + 27x_2^2) \right]$ | 2 | [−2, 2] | 3 |

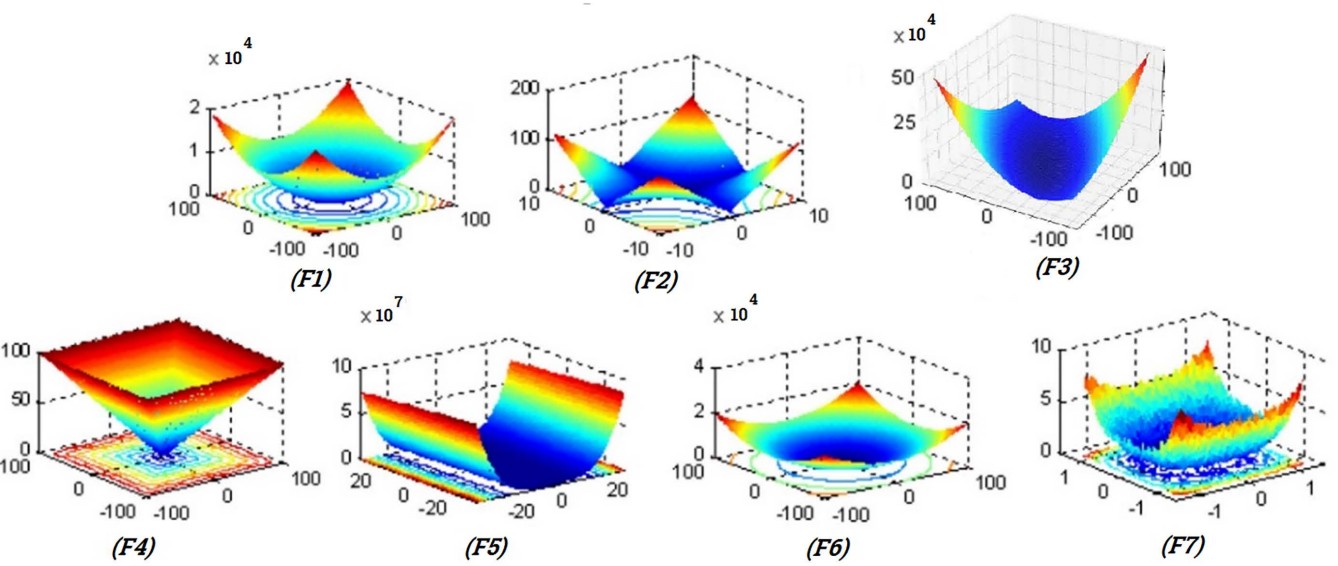

**Fig 5. 2 D versions of unimodal benchmark functions.**

the global minimum. The functions exhibit diverse structural features, some with dense patterns of peaks and valleys, and others with sharp ridges or smooth oscillations, each contributing to a higher level of difficulty. These visualizations provide an intuitive understanding of the ruggedness and deceptive nature of multimodal functions, emphasizing the importance of exploration mechanisms in metaheuristic performance.

Fig 7 presents the two-dimensional surface plots visualizations of fixed-dimension multimodal benchmark functions (F13–F17), each characterized by multiple local optima that pose challenges for optimization algorithms. These functions are widely used to assess an algorithm's ability to escape local minima and find the global optimum. Function F13 shows regularly spaced sharp peaks, indicating a highly repetitive and deceptive landscape. F15 and F17 exhibit smoother, more complex terrains with multiple valleys and ridges, which simulate real-world nonlinearities. F16 features a dome-shaped surface with a steep gradient, testing convergence behavior. Overall, these benchmark functions simulate diverse multimodal complexities crucial for evaluating optimization performance.

**4.1.2 CEC 2017 benchmark test functions.** The CEC 2017 benchmark Test Functions is a widely recognized set of test functions for evaluating the performance of optimization algorithms [47,48]. It includes a mix of unimodal, multimodal,

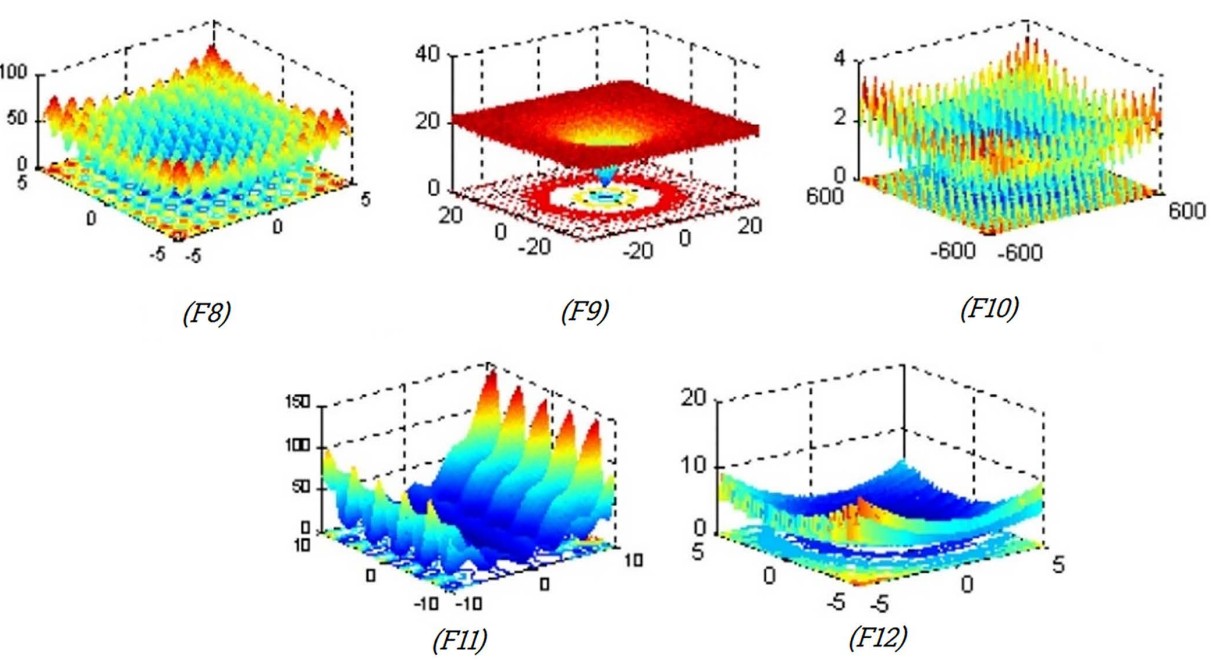

*(F8)*　　*(F9)*　　*(F10)*

*(F11)*　　*(F12)*

**Fig 6. 2 D version of multimodal benchmark functions.**

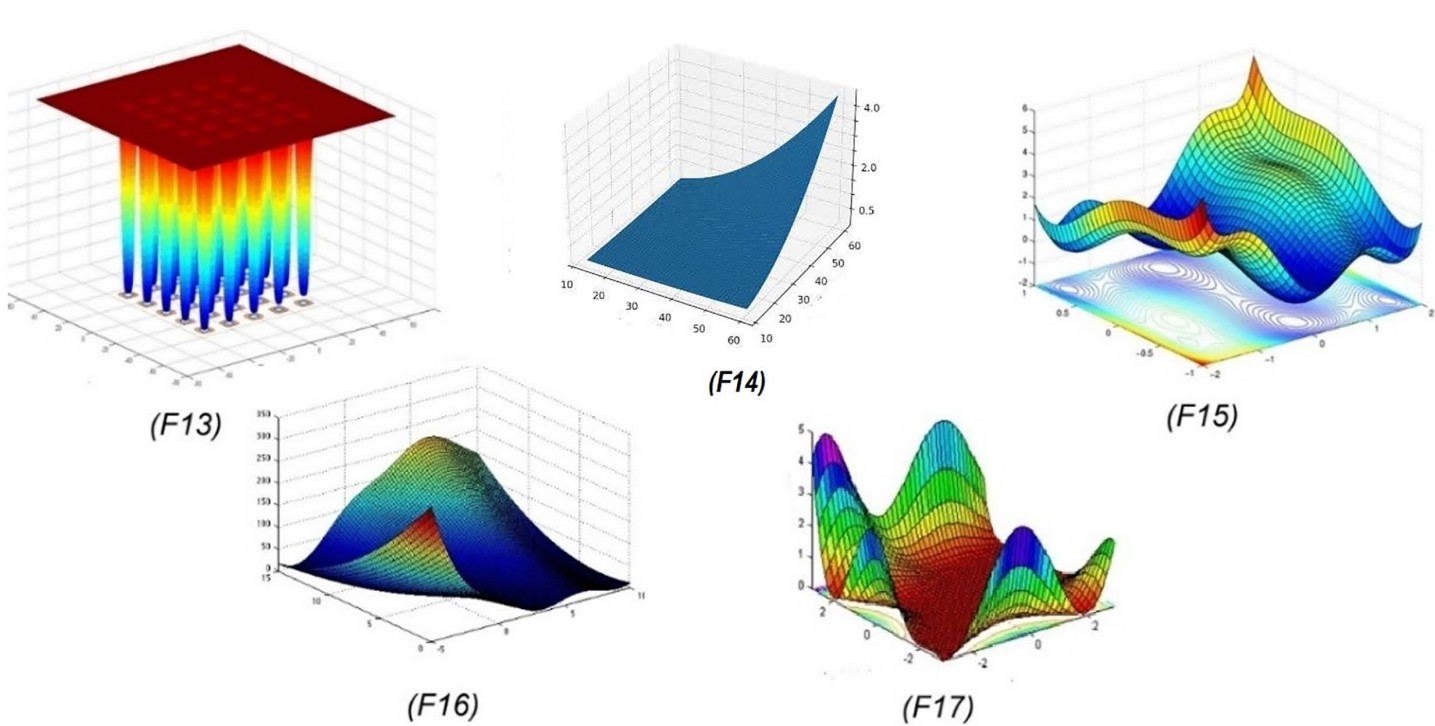

*(F13)*　　*(F14)*　　*(F15)*

*(F16)*　　*(F17)*

**Fig 7. 2 D version of fixed-dimension multimodal benchmark functions.**

hybrid, and composition functions designed to test various aspects of algorithmic capabilities, such as exploitation, exploration, robustness, and convergence speed. These functions are used in the annual IEEE Congress on Evolutionary Computation competitions and have become a standard benchmark in the optimization community.

For the F2 function, due to high regged, high sensitivity to initialization in performance, and instability in presenting results, fair and accurate results for comparing algorithms are not provided, so the results of this study are usually not reported.

The specifications and features of these functions are listed in Table 7.

## 4.2 Experimental setup

To ensure reproducibility and rigorous comparison, the proposed Felis Catus Optimization (FCO) algorithm was evaluated against 17 representative metaheuristic algorithms from different evolutionary and swarm-intelligence paradigms. The compared algorithms are: Particle Swarm Optimization (PSO) [4], Grey Wolf Optimizer (GWO) [5]; Manta Ray Foraging Optimization (MRFO); Equilibrium Optimizer (EO), Sine Cosine Algorithm (SCA); Sand Cat Optimizer (SCO), Elite-Based Optimizer with Covariance Matrix Adaptive Reconstruction (EBO CMAR); Covariance Matrix Adaptation Evolution Strategy (CMA-ES); Runge Kutta Optimization Algorithm (RUN), Cheetah Optimization Algorithm (ChOA), Differential Evolution (DE), L-SHADE Algorithm (L-SHADE), Coyote Optimization Algorithm (COA), Reptile Search Algorithm (RSA), Quantum Inspired Optimization (QIO), and Salp Swarm Algorithm (SSA) [36].

Each algorithm was executed under identical conditions to preserve fairness and remove external bias. The experimental configuration was as follows:

- Population size = 80

- Maximum iterations = 1000

- Independent runs = 30 times per function, using synchronized random seeds across all algorithms

- Optimization objective: Minimization

All tests were conducted in MATLAB R2019a on a personal computer equipped with Intel® Core™ i5-3230M CPU @ 2.60 GHz, 4.00 GB RAM (3.82 GB usable), and Windows 7 Ultimate (Service Pack 1, 64-bit). This configuration ensures computational comparability with numerous previously published metaheuristic benchmarks, guaranteeing reproducibility and controlled precision.

For each algorithm and benchmark function, the best, mean, and standard deviation (Std) of the final objective values were recorded over 30 independent runs. As all problems were minimization-oriented, lower values represent superior performance.

To evaluate significance among competing algorithms, a non-parametric Friedman test was first applied to detect global performance differences. When significance was confirmed ($p < 0.05$), the Holm post-hoc procedure ($\alpha = 0.05$) was employed using FCO as the control algorithm. This dual-stage statistical framework identifies whether observed differences are statistically meaningful or due to random variation.

**Table 7. Specifications and features of CEC 2017 benchmark Test Functions.**

| Category | Function Index Range | Number of Functions | Features |
|---|---|---|---|
| Unimodal | F1 | 1 | Evaluation of exploitation and convergence speed |
| Multimodal | F2 - F3 | 2 | Examining the ability to escape from local Optimum |
| Hybrid | F4 - F10 | 7 | Non-homogeneous combination of basic polynomials |
| Composition | F11 – F29 | 19 | Combining functions with different weighting, shifting, rotation, and scaling |

The benchmarking pipeline setup parameters, stochastic control, data logging, and statistical analysis follows the PLOS ONE guidelines for reporting algorithmic experiments, ensuring transparency, replicability, and neutral performance assessment.

• **Algorithmic Parameters and Control Policy**

The internal configuration of the proposed Felis Catus Optimization (FCO) algorithm including fixed defaults and adaptive schedules was retained constant throughout all benchmarks to guarantee reproducibility. Table 8 lists the complete set of control parameters consistent with the executable MATLAB source (FCO v2.m).

## 4.3 Optimization results on benchmark functions

In this section, the optimization performance of the proposed Felis Catus Optimization (FCO) algorithm is evaluated using two standardized test suites: the CEC 2005 and CEC 2017 benchmark functions. These testbeds include a wide spectrum of mathematical landscapes ranging from simple unimodal to highly complex composite functions designed to assess an optimizer's robustness, convergence capacity, and balance between exploration and exploitation.

All experiments follow the configuration defined in Section 4.2, with population size = 80, 1000 iterations, and 30 independent runs for each function. Because all benchmarks are minimization problems, lower objective values indicate better performance. For every algorithm, the best, mean, and standard deviation (Std) of the final solutions were recorded, followed by statistical analysis through the Friedman test and Holm post-hoc procedure at $\alpha = 0.05$ with FCO as the control algorithm.

The subsequent subsections summarize the comparative results and interpretations for the CEC 2005 and CEC 2017 datasets, respectively.

**Table 8. Default algorithmic parameters and control policies of felis catus optimization (FCO v2, MATLAB R2019a).**

| Parameter | Symbol | Default Value/ Policy | Adaptation Schedule | Descript |
|---|---|---|---|---|
| Population size | $N$ | 80 | Fixed | Population size for each optimization run |
| Maximum iterations | MaxIter | 1000 | Fixed | Maximum number of iterations for each optimization run |
| Male-to-female ratio | MaleRatio | | Fixed | Fraction of male (explorer) agents in total population |
| Exploration factor | β | 0.7→0.3 | Linear increase per iteration | Exploration radius factor for male agents |
| gamma | γ | 0.3→0.7 | Linear increase per iteration | Exploitation factor controlling female Gaussian refinement |
| sigma_ENI | σ_ENI | 0.05· (bound_max − bound_min) | Fixed | Environmental noise intensity for ENI perturbations |
| Rejuvenation probability | ProbRJ | 0.10 | Fixed | Probability of random rejuvenation per agent per iteration |
| Aging threshold | AgeThreshold | 30 | Fixed | Age limit triggering rejuvenation of outdated agents |
| Elite retention fraction | EliteRate | 0.10 | Fixed | Retains top 10% of population before update |
| Random number seed | – | rng(42,'twister') | – | Deterministic seed ensuring full reproducibility |
| Fitness normalization | – | None (raw CEC values) | – | No normalization applied to CEC benchmark fitness values |

**4.3.1 Optimization results on CEC 2005 benchmark functions.** The CEC 2005 benchmark suite encompasses 17 functions (F1–F17), grouped into three distinct categories unimodal (F1–F7), multimodal (F8–F12), and fixed-dimension composite (F13–F17). Each category was chosen to evaluate a specific behavioral dimension of the proposed Felis Catus Optimization (FCO) algorithm, including its exploitation accuracy, global search capability, and adaptability under complex landscapes. All functions were treated as minimization problems, and convergence behavior was assessed using the metrics of mean and standard deviation across 30 independent runs, consistent with the experimental setup defined in Section 4.2. The proposed Felis Catus Optimization (FCO) algorithm was benchmarked against a comprehensive set of well-established metaheuristics including SCA, DE, WOA, CMA-ES, EO, GWO, L-SHADE, MRFO, PSO, QIO, RSA, RUN, COA, ChOA, CMAR, SCO, and SSA. The comparative performance values for all algorithms on these 17 functions are summarized in Table 9.

For the unimodal group (F1–F7), which measures exploitation ability, FCO exhibited near-zero mean errors on Sphere and Schwefel 2.22 ($1.79 \times 10^{-16}$ and $6.11 \times 10^{-13}$) comparable to CMA-ES, EO, and DE. In the more irregular Schwefel 2.21 and Rosenbrock functions, FCO maintained good precision and smaller dispersion than ChOA and CMAR, demonstrating robust convergence and fine-grained search control.

In the multimodal set (F8–F12), which tests exploration capability, FCO moderate accuracy on Rastrigin and Ackley where SSA and SCA obtained the best values. Nevertheless, the algorithm preserved low variance and consistent precision in Penalized1 and Penalized2 functions, confirming that its continuous-space global exploration was effective in avoiding premature convergence.

On the low-dimensional composite functions (F13–F17), FCO produced optimal fitness values identical to theoretical optima (e.g., −1.0316 for Six-Hump-Camel, 3.0 for Goldstein-Price), with negligible standard deviation ($< 10^{-15}$). This confirms the algorithm's deterministic stability and strong local refinement through its female (cat) driven Gaussian exploitation operator.

Overall, the mean results confirm that FCO is a balanced optimizer with competitive accuracy and variance, sustaining stable performance across diverse landscapes.

• Friedman test

The non-parametric Friedman test performed over the 17 functions yielded a test statistic of $\chi^2 = 93.6095$ with $p = 1.339 \times 10^{-12}$, indicating a statistically significant difference among the 18 algorithms. The average ranks are summarized in Table 10.

FCO ranked 7th overall, with an average rank of 8.118, placing it within the first half of tested methods and confirming its competitive behavior without dependence on problem type. The best ranks were recorded by SCA (4.941), DE (5.647), and WOA (5.647), while SSA (15.824), QIO (13.824), and CMAR (13.294) exhibited the weakest overall performance. FCO's ranking was slightly above CMA-ES (7.118), EO (7.353), and GWO (7.706), outperforming established frameworks such as PSO (8.706), RUN (8.824), L-SHADE (9.706), and COA (9.824). This ranking pattern demonstrates that FCO maintains a well-balanced search behavior across different function categories and is not restricted to either smooth or high-frequency landscapes.

• Wilcoxon Signed Rank Test Analysis

To evaluate pairwise differences with respect to FCO, the Holm sequential procedure was applied under minimization mode [49]. As shown in Table 11, significant differences at the adjusted confidence level (α-adj) were found only for two algorithms:

• SSA ($p = 2.573 \times 10^{-5}$, α-adj = 0.002941) significantly better than FCO,

• QIO ($p = 0.001833$, α-adj = 0.003125) significantly better than FCO.

**Table 9. Mean values (± standard deviation) of 18 metaheuristic algorithms on 17 CEC 2005 benchmark test functions (F1–F17). Lower values indicate better performance for minimization.**

| [] | FCO | ChOA | CMA-ES | CMAR | COA | DE | EO | GWO | L_SHADE |
|---|---|---|---|---|---|---|---|---|---|
| F1 | 1.79E-16 (1.50E-16) | 6.98E+00 (1.76E+00) | 8.44E-31 (1.12E-31) | 3.50E+03 (1.02E+03) | 7.33E-02 (1.18E-02) | 3.58E-11 (3.03E-11) | 2.17E-217 (0.00E+00) | 1.24E-80 (1.64E-80) | 1.75E+01 (1.65E+01) |
| F2 | 6.11E-13 (2.90E-13) | 7.06E+00 (3.03E+00) | 9.25E-31 (6.14E-32) | 2.65E+01 (4.22E+00) | 1.04E+00 (1.08E-01) | 1.14E-05 (5.68E-06) | 2.27E-112 (5.84E-112) | 1.66E-46 (1.62E-46) | 6.12E-02 (6.91E-02) |
| F3 | 9.28E+02 (1.89E+02) | 1.20E+03 (3.76E+02) | 9.44E+00 (8.39E+00) | 4.89E+03 (2.43E+03) | 5.34E+02 (5.67E+02) | 1.11E+01 (6.35E+00) | 6.30E-222 (0.00E+00) | 1.65E-23 (6.29E-23) | 1.11E+03 (3.08E+02) |
| F4 | 9.75E+00 (1.06E+00) | 1.34E+01 (4.14E+00) | 6.41E-16 (2.49E-15) | 2.02E+01 (3.38E+00) | 1.19E+00 (9.01E-01) | 2.59E+00 (2.65E+00) | 2.45E-112 (6.47E-112) | 2.17E-20 (3.12E-20) | 1.02E+01 (2.05E+00) |
| F5 | 4.37E+01 (3.07E+01) | 5.03E+02 (3.34E+02) | 3.83E+01 (2.62E+01) | 1.02E+06 (7.03E+05) | 3.54E+01 (1.80E+00) | 2.12E+01 (1.53E+00) | 2.87E+01 (2.72E-02) | 2.63E+01 (6.50E-01) | 4.02E+03 (4.39E+03) |
| F6 | 1.70E+00 (1.29E+00) | 1.60E+01 (3.12E+00) | 0.00E+00 (0.00E+00) | 4.08E+03 (1.25E+03) | 0.00E+00 (0.00E+00) | 0.00E+00 (0.00E+00) | 0.00E+00 (0.00E+00) | 0.00E+00 (0.00E+00) | 1.47E+01 (2.03E+01) |
| F7 | 8.51E-02 (1.38E-02) | 4.21E-02 (1.46E-02) | 2.50E-02 (1.24E-02) | 5.54E-01 (4.00E-01) | 8.46E-02 (2.85E-02) | 1.31E-02 (3.21E-03) | 1.64E-04 (1.15E-04) | 2.94E-04 (1.86E-04) | 4.32E-02 (2.06E-02) |
| F8 | 5.28E+01 (7.69E+00) | 8.67E+01 (2.26E+01) | 1.25E+01 (3.24E+00) | 1.70E+02 (2.19E+01) | 5.87E+01 (7.18E+00) | 1.81E+02 (1.55E+01) | 1.69E+02 (2.26E+01) | 0.00E+00 (0.00E+00) | 2.61E+01 (5.47E+00) |
| F9 | 2.07E+01 (5.39E-02) | 2.00E+01 (6.49E-02) | 2.00E+01 (4.33E-03) | 2.00E+01 (4.06E-03) | 1.88E+01 (4.75E+00) | 2.00E+01 (5.43E-02) | 6.67E-01 (3.65E+00) | 2.08E+01 (6.30E-02) | 2.00E+01 (1.27E-03) |
| F10 | 1.62E+01 (9.91E-01) | 1.90E+01 (2.25E+00) | 8.46E+00 (1.23E+00) | 2.03E+01 (1.37E+00) | 1.40E+01 (1.68E+00) | 2.16E+01 (7.14E-01) | 2.08E+01 (2.09E+00) | 1.04E+00 (2.98E+00) | 1.19E+01 (1.20E+00) |
| F11 | 1.69E+00 (1.23E+00) | 1.28E+01 (6.30E+00) | 8.53E-31 (1.13E-31) | 1.59E+04 (5.01E+04) | 1.03E-01 (5.41E-02) | 5.20E-12 (7.14E-12) | 3.84E+00 (5.08E+00) | 1.44E-02 (1.23E-02) | 4.90E-01 (5.45E-01) |
| F12 | 3.65E-03 (5.08E-03) | 2.30E+00 (4.79E+00) | 3.66E-04 (2.01E-03) | 5.89E+05 (9.78E+05) | 2.72E-01 (1.07E-01) | 3.66E-04 (2.01E-03) | 2.04E-01 (7.46E-02) | 2.10E-01 (1.38E-01) | 2.33E+01 (7.97E+01) |
| F13 | 8.04E-02 (1.52E-17) | 8.04E-02 (1.29E-07) | 8.21E-02 (3.15E-03) | 8.26E-02 (3.41E-03) | 8.09E-02 (7.04E-04) | 8.04E-02 (2.82E-17) | 8.04E-02 (4.40E-08) | 8.09E-02 (1.84E-03) | 8.09E-02 (1.85E-03) |
| F14 | 3.83E-04 (6.76E-05) | 8.25E-04 (2.03E-04) | 7.08E-03 (9.55E-03) | 2.58E-03 (5.21E-03) | 7.51E-04 (6.39E-05) | 4.91E-04 (3.73E-04) | 3.91E-03 (7.90E-03) | 4.38E-03 (8.13E-03) | 4.99E-04 (2.59E-04) |
| F15 | −1.03E+00 (4.52E-16) | −1.03E+00 (9.21E-07) | −4.87E-01 (3.17E-01) | −1.03E+00 (4.52E-16) | −1.03E+00 (1.09E-05) | −1.03E+00 (4.52E-16) | −1.03E+00 (1.72E-08) | −1.03E+00 (5.59E-10) | −1.03E+00 (4.52E-16) |
| F16 | 3.98E-01 (0.00E+00) | 3.98E-01 (3.52E-07) | 3.98E-01 (0.00E+00) | 3.98E-01 (0.00E+00) | 3.98E-01 (7.42E-06) | 3.98E-01 (0.00E+00) | 3.98E-01 (1.76E-07) | 3.98E-01 (2.27E-08) | 3.98E-01 (0.00E+00) |
| F17 | 3.00E+00 (1.61E-15) | 3.00E+00 (6.48E-06) | 3.00E+00 (2.65E-14) | 3.00E+00 (1.05E-15) | 3.00E+00 (1.26E-04) | 3.00E+00 (1.91E-15) | 3.00E+00 (1.03E-06) | 3.00E+00 (1.37E-06) | 3.00E+00 (5.53E-16) |
| [] | MRFO | PSO | QIO | RSA | RUN | SCA | SCO | SSA | WOA |
| F1 | 3.72E+01 (2.65E+01) | 3.18E-06 (3.93E-06) | 6.04E+02 (3.32E+02) | 6.39E-293 (0.00E+00) | 1.54E-01 (1.32E-01) | 1.46E-154 (5.90E-154) | 6.72E-06 (9.12E-06) | 3.53E+03 (9.02E+02) | 7.66E-184 (0.00E+00) |
| F2 | 2.82E+00 (9.33E-01) | 6.67E-01 (2.54E+00) | 6.17E-01 (2.71E-01) | 3.72E-160 (1.22E-159) | 6.63E-02 (3.15E-02) | 1.38E-81 (4.56E-81) | 1.31E+01 (1.37E+01) | 2.70E+01 (2.86E+00) | 5.99E-106 (3.05E-105) |
| F3 | 5.83E+02 (1.72E+02) | 3.96E+02 (1.70E+02) | 5.29E+04 (5.02E+03) | 2.31E-32 (9.94E-32) | 6.04E+02 (2.73E+02) | 6.67E-110 (3.52E-109) | 2.40E+03 (2.65E+03) | 9.22E+03 (3.83E+03) | 5.63E-03 (2.27E-02) |
| F4 | 1.13E+01 (1.92E+00) | 3.20E+00 (7.67E-01) | 7.97E+01 (4.40E+00) | 3.80E-03 (1.18E-02) | 2.84E+00 (4.88E-01) | 9.52E-81 (2.63E-80) | 2.48E+01 (4.03E+00) | 2.33E+01 (2.84E+00) | 3.92E-27 (1.82E-26) |
| F5 | 2.21E+03 (1.41E+03) | 3.39E+03 (1.64E+04) | 2.07E+07 (1.39E+07) | 6.00E+01 (5.31E+01) | 1.24E+02 (3.71E+01) | 5.03E-03 (8.43E-03) | 4.83E+01 (3.75E+01) | 1.00E+06 (5.60E+05) | 4.75E-02 (7.26E-02) |
| F6 | 3.34E+02 (1.59E+02) | 0.00E+00 (0.00E+00) | 4.72E+02 (3.00E+02) | 3.67E-01 (7.18E-01) | 2.67E-01 (5.21E-01) | 0.00E+00 (0.00E+00) | 8.68E+01 (5.15E+01) | 3.36E+03 (9.98E+02) | 0.00E+00 (0.00E+00) |
| F7 | 6.15E-02 (2.59E-02) | 2.17E-02 (8.61E-03) | 5.68E+00 (3.14E+00) | 3.88E-03 (2.21E-03) | 2.77E-02 (1.46E-02) | 2.93E-05 (3.00E-05) | 1.08E-01 (4.47E-02) | 6.00E-01 (2.56E-01) | 1.76E-04 (2.52E-04) |

*(Continued)*

**Table 9.** (Continued)

| [] | FCO | ChOA | CMA-ES | CMAR | COA | DE | EO | GWO | L_SHADE |
|---|---|---|---|---|---|---|---|---|---|
| F8 | 3.28E+01 (1.73E+01) | 3.56E+01 (1.05E+01) | 3.01E+02 (1.96E+01) | 1.61E+02 (3.08E+01) | 2.01E+01 (7.04E+00) | 0.00E+00 (0.00E+00) | 1.39E+02 (2.71E+01) | 2.14E+02 (2.01E+01) | 0.00E+00 (0.00E+00) |
| F9 | 2.00E+01 (0.00E+00) | 2.02E+01 (2.57E-01) | 2.00E+01 (4.57E-09) | 2.00E+01 (8.17E-03) | 1.96E+01 (3.69E+00) | 8.88E-16 (0.00E+00) | 2.00E+01 (0.00E+00) | 1.94E+01 (4.63E-01) | 1.48E-15 (1.35E-15) |
| F10 | 1.14E+01 (1.58E+00) | 8.34E+00 (1.42E+00) | 3.31E+01 (2.45E+00) | 2.45E+01 (4.97E+00) | 7.91E+00 (1.64E+00) | 0.00E+00 (0.00E+00) | 2.29E+01 (4.36E+00) | 2.43E+01 (1.09E+00) | 0.00E+00 (0.00E+00) |
| F11 | 3.87E+00 (2.08E+00) | 6.92E-03 (2.63E-02) | 1.19E+08 (7.26E+07) | 1.13E+01 (5.54E+00) | 2.82E-03 (1.22E-02) | 8.24E-06 (1.44E-05) | 1.04E+01 (6.51E+00) | 7.54E+03 (1.78E+04) | 7.77E-05 (8.06E-05) |
| F12 | 4.08E+01 (1.88E+01) | 2.59E-03 (4.72E-03) | 1.71E+08 (1.01E+08) | 7.17E-01 (5.64E-01) | 7.78E-03 (8.16E-03) | 4.78E-04 (2.12E-03) | 2.55E+01 (1.51E+01) | 6.41E+05 (5.61E+05) | 1.71E-03 (4.17E-03) |
| F13 | 8.09E-02 (1.85E-03) | 8.26E-02 (3.43E-03) | 8.06E-02 (5.23E-04) | 8.46E-02 (3.60E-03) | 8.41E-02 (3.75E-03) | 8.04E-02 (1.32E-05) | 8.04E-02 (3.47E-17) | 8.29E-02 (2.32E-03) | 8.05E-02 (3.15E-05) |
| F14 | 2.42E-03 (6.09E-03) | 1.75E-03 (5.07E-03) | 8.72E-04 (3.80E-04) | 5.19E-03 (8.52E-03) | 3.10E-03 (6.89E-03) | 3.57E-04 (5.08E-05) | 6.43E-04 (4.49E-04) | 2.04E-03 (7.82E-04) | 6.04E-04 (4.51E-04) |
| F15 | −1.03E+00 (4.46E-16) | −1.03E+00 (4.52E-16) | −1.03E+00 (4.52E-16) | −1.03E+00 (4.52E-16) | −1.03E+00 (4.52E-16) | −1.03E+00 (7.34E-06) | −1.03E+00 (4.52E-16) | −1.03E+00 (2.31E-03) | −1.03E+00 (9.47E-12) |
| F16 | 3.98E-01 (0.00E+00) | 3.98E-01 (0.00E+00) | 3.98E-01 (0.00E+00) | 3.98E-01 (0.00E+00) | 3.98E-01 (0.00E+00) | 3.98E-01 (8.62E-06) | 3.98E-01 (0.00E+00) | 4.05E-01 (1.14E-02) | 3.98E-01 (2.50E-07) |
| F17 | 3.00E+00 (1.30E-15) | 3.00E+00 (1.36E-15) | 3.00E+00 (9.44E-16) | 3.00E+00 (9.62E-16) | 3.00E+00 (2.18E-15) | 3.00E+00 (3.66E-05) | 3.00E+00 (3.38E-15) | 3.07E+00 (1.20E-01) | 4.80E+00 (6.85E+00) |

**Table 10. Friedman mean ranks of 18 metaheuristic algorithms over 17 CEC 2005 benchmark functions (F1–F17).**

| Friedman Test: Chi2=93.6095, p=1.339e-12 | |
|---|---|
| **Alg** | **Rank** |
| SCA | 4.941 |
| DE | 5.647 |
| WOA | 5.647 |
| CMA-ES | 7.118 |
| EO | 7.353 |
| GWO | 7.706 |
| FCO | **8.118** |
| PSO | 8.706 |
| RUN | 8.824 |
| L_SHADE | 9.706 |
| RSA | 9.706 |
| COA | 9.824 |
| MRFO | 10.588 |
| SCO | 11.765 |
| ChOA | 12.412 |
| CMAR | 13.294 |
| QIO | 13.824 |
| SSA | 15.824 |

**Table 11. Results of the Holm sequential procedure for pairwise comparisons against the proposed Felis Catus Optimization (FCO) algorithm on the CEC 2005 benchmark suite (p – values and adjusted α levels).**

| Alg | P-Value | α-adj | Result |
| --- | --- | --- | --- |
| SSA | **2.573e-05** | **0.002941** | **Significant (Better)** |
| QIO | **0.001833** | **0.003125** | **Significant (Better)** |
| CMAR | **0.004699** | **0.003333** | **Not Significant (Better)** |
| ChOA | **0.01902** | **0.003571** | **Not Significant (Better)** |
| SCO | **0.0464** | **0.003846** | **Not Significant (Better)** |
| SCA | 0.08279 | 0.004167 | Not Significant (Worse) |
| DE | 0.1773 | 0.004545 | Not Significant (Worse) |
| MRFO | **0.1773** | **0.005** | **Not Significant (Better)** |
| WOA | 0.1773 | 0.005556 | Not Significant (Worse) |
| COA | **0.3515** | **0.00625** | **Not Significant (Better)** |
| L_SHADE | **0.3857** | **0.007143** | **Not Significant (Better)** |
| RSA | **0.3857** | **0.008333** | **Not Significant (Better)** |
| CMA-ES | 0.585 | 0.01 | Not Significant (Worse) |
| EO | 0.6762 | 0.0125 | Not Significant (Worse) |
| RUN | **0.6999** | **0.01667** | **Not Significant (Better)** |
| PSO | **0.748** | **0.025** | **Not Significant (Better)** |
| GWO | 0.8221 | 0.05 | Not Significant (Worse) |

For all other comparisons ($p > α\text{-adj}$), no significant difference was observed, even for those with a slightly lower rank (e.g., CMAR, ChOA, SCA, DE). Hence, FCO is statistically equivalent to fourteen out of seventeen competitor algorithms, including CMA-ES, EO, GWO, L-SHADE, WOA, DE, and RUN. Such statistical robustness indicates that, from a confidence-corrected perspective, FCO belongs to the upper performance tier of modern metaheuristics, with a balance between exploration and exploitation close to the best performing algorithms. In Holm's multiple comparison analysis where FCO is used as the reference (control algorithm), each competitor is classified as Better or Worse depending on its mean rank difference relative to FCO and the corresponding sign of the comparison statistics.

Specifically, when an algorithm yielded a lower average Friedman rank than FCO, it was labeled as Better conversely, when its rank was higher, it was labeled as Worse. Statistical significance was determined by the adjusted-level α-adj under Holm's sequential procedure. Algorithms with $p < α\text{-adj}$ were considered Significantly Better, while those with $p > α\text{-adj}$ but lower mean rank were categorized as Not Significant (Better). Similarly, algorithms with $p > α\text{-adj}$ and higher mean rank than FCO were tagged Not Significant (Worse), implying no meaningful difference but slightly inferior numerical performance.

- **Critical Difference (CD) analysis**

To visualize the statistical relationship among the algorithms following the Friedman test, the Critical Difference (CD) diagram in Fig 8 presents the average ranks of all 18 compared methods based on the CEC 2005 benchmark functions. The CD value, computed using the Nemenyi post-hoc procedure at a significance level of $α = 0.05$, equals 4.104. Algorithms connected by a distance smaller than this threshold are statistically equivalent in terms of performance. As illustrated, FCO achieved a mean rank of 8.118, positioned near the central cluster, and demonstrated statistical equivalence with the majority of algorithms including SCA, DE, WOA, CMA-ES, EO, and GWO. Only SSA and QIO displayed significantly better performance, while other competitors showed no significant superiority or inferiority compared with FCO. This confirms that FCO performs on par with 14 well-established algorithms and maintains a competitive balance between exploration

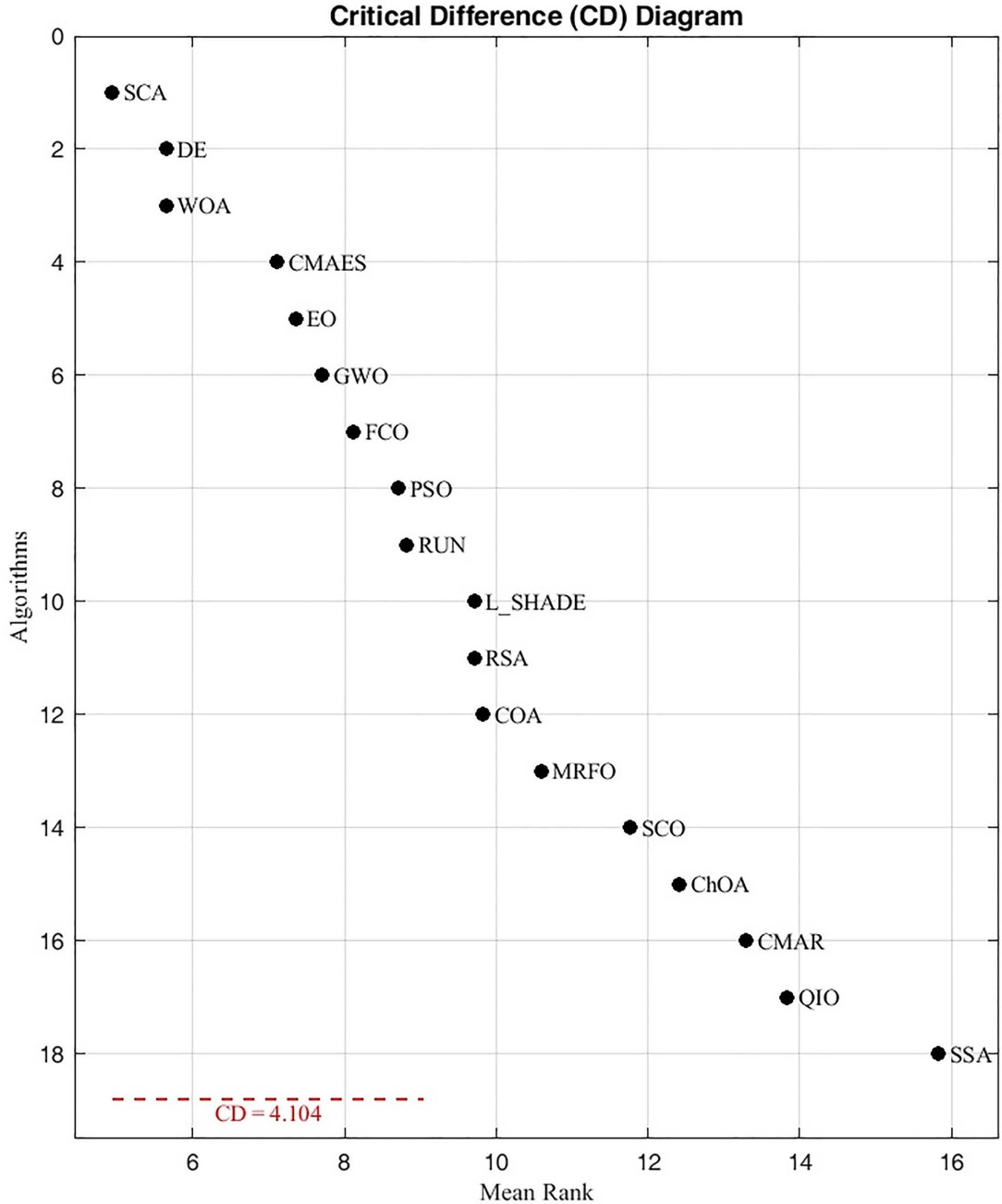

**Fig 8. Critical Difference (CD) diagram based on Friedman mean ranks for 18 algorithms on the CEC 2005 benchmark set (α = 0.05, Nemenyi test).** Smaller ranks correspond to better performance. The red dashed line indicates the critical threshold CD = 4.104 algorithms within this distance are not statistically different.

and exploitation across the benchmark suite. The reversed-axis presentation (smaller ranks at the top) conforms to the standard visualization convention, where higher-quality algorithms appear higher in the plot.

Representative convergence patterns of the FCO algorithm are illustrated in Fig 9. The profiles show a consistent and rapid decline in mean error value on a logarithmic scale, reflecting stable convergence behavior across unimodal, multimodal, and composite functions. FCO achieves early stabilization compared with most competitor algorithms, indicating a balanced exploration–exploitation trade-off and high robustness. Full convergence plots for all 17 CEC 2005 functions are presented in S2 Appendix to provide comprehensive visual insight.

**4.3.2 Optimization results on CEC 2017 benchmark functions.** The CEC 2017 benchmark suite consists of 30 complex functions (F1–F30), encompassing the main challenge categories:

• Unimodal (F1–F3) tests exploitation precision.

• Multimodal (F4–F10) evaluates global exploration ability.

• Hybrid composition (F11–F20) measures adaptability across irregular landscapes.

• Composite and Expanded composition (F21–F30) examine resilience in high-dimensional rugged surfaces.

Each benchmark function was optimized under a minimization setting using 30 independent runs, 1000 maximum iterations per run, and a 30-dimensional search space. The proposed Felis Catus Optimization (FCO) algorithm was benchmarked against 17 well-known metaheuristics, including ChOA, CMA-ES, CMAR, COA, DE, EO, GWO, L-SHADE, MRFO, PSO, QIO, RSA, RUN, SCA, SCO, SSA, and WOA.

All functions were treated as direct minimization problems, and solution quality was quantified through the average and standard deviation of fitness values across runs. The complete statistical results (Mean ± Std) for all 30 functions and 18 algorithms are summarized in Table 12, where lower mean values correspond to better solution quality. The values follow scientific notation (E-format) for numerical consistency.

• **Unimodal Functions (F1–F3)**

FCO demonstrates low mean errors on all unimodal tasks, notably on F1 (Sphere) and F3 (Rosenbrock) with results of $2.49 \times 10^3$ and $6.14 \times 10^4$, respectively, well within the magnitude range of DE, GWO, and L-SHADE. The stability of

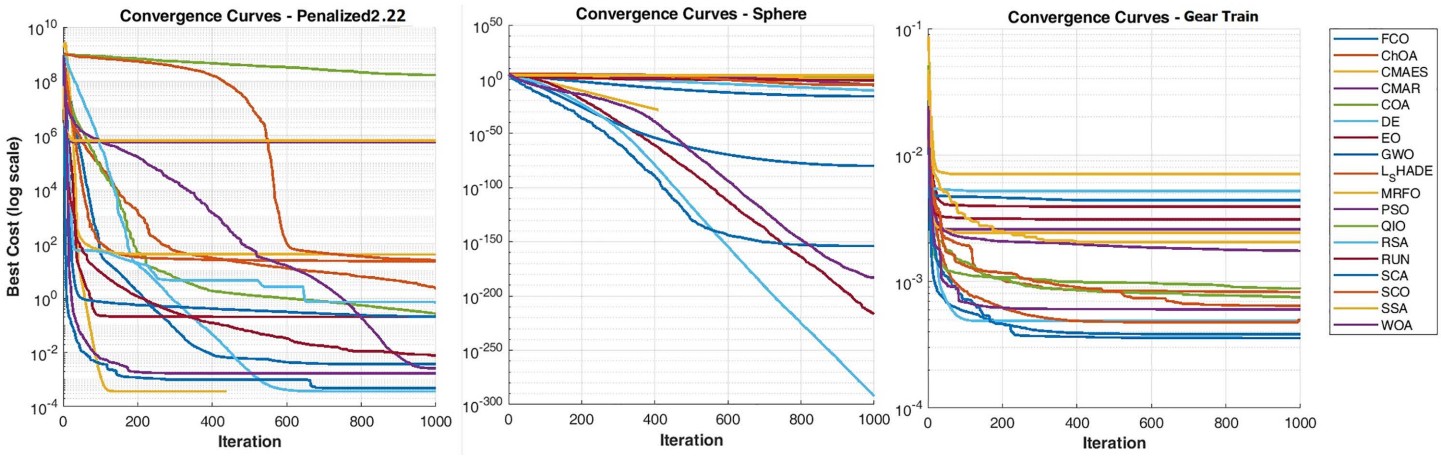

**Fig 9. Representative convergence profiles of the Felis Catus Optimization (FCO) algorithm over selected CEC 2005 functions.**

**Table 12. Summarizes the statistical results (Mean±Std) of all 30 functions and 18 algorithms. Lower mean values correspond to better solution quality. The values follow scientific notation (E-format) for numerical consistency.**

| Func | FCO | ChOA | CMA-ES | CMAR | COA | DE | EO | GWO | L_SHADE |
|------|------|------|--------|------|-----|-----|-----|-----|---------|
| F1 | 2.49E+03 (2.87E+03) | 1.44E+07 (4.43E+06) | 1.00E+02 (1.85E-03) | 3.69E+10 (7.86E+09) | 1.51E+10 (2.33E+09) | 3.90E+02 (1.12E+03) | 8.49E+07 (3.51E+07) | 1.48E+09 (1.12E+09) | 2.72E+08 (1.94E+08) |
| F2 | 7.31E+16 (8.10E+16) | 7.88E+18 (2.16E+19) | 4.11E+15 (2.09E+16) | 3.77E+44 (1.79E+45) | 1.67E+34 (6.85E+34) | 1.37E+17 (6.89E+17) | 4.43E+19 (1.23E+20) | 1.77E+31 (9.65E+31) | 6.66E+25 (2.94E+26) |
| F3 | 6.15E+04 (8.19E+03) | 2.41E+04 (8.57E+03) | 1.33E+04 (6.14E+03) | 8.40E+04 (1.21E+04) | 5.64E+04 (6.23E+03) | 2.47E+04 (9.34E+03) | 8.06E+04 (1.49E+04) | 3.75E+04 (9.43E+03) | 9.13E+04 (1.75E+04) |
| F4 | 5.01E+02 (2.35E+01) | 5.35E+02 (3.54E+01) | 4.95E+02 (7.46E+00) | 8.18E+03 (3.35E+03) | 1.19E+03 (2.07E+02) | 4.85E+02 (2.89E+00) | 5.51E+02 (3.48E+01) | 5.65E+02 (4.88E+01) | 5.71E+02 (3.61E+01) |
| F5 | 6.88E+02 (1.61E+01) | 6.60E+02 (4.20E+01) | 5.13E+02 (3.49E+00) | 8.00E+02 (3.82E+01) | 7.14E+02 (1.92E+01) | 6.93E+02 (1.10E+01) | 7.86E+02 (4.36E+01) | 5.99E+02 (3.99E+01) | 6.54E+02 (3.10E+01) |
| F6 | 6.17E+02 (3.67E+00) | 6.38E+02 (1.04E+01) | 6.00E+02 (1.74E-02) | 6.61E+02 (9.17E+00) | 6.36E+02 (4.12E+00) | 6.00E+02 (6.87E-04) | 6.72E+02 (8.16E+00) | 6.07E+02 (3.66E+00) | 6.00E+02 (4.33E-02) |
| F7 | 9.95E+02 (1.76E+01) | 9.34E+02 (4.08E+01) | 7.41E+02 (2.64E+00) | 1.23E+03 (9.71E+01) | 1.01E+03 (3.96E+01) | 9.29E+02 (1.04E+01) | 1.30E+03 (8.36E+01) | 8.63E+02 (5.26E+01) | 9.25E+02 (2.11E+01) |
| F8 | 9.50E+02 (1.63E+01) | 9.36E+02 (3.62E+01) | 8.12E+02 (3.96E+00) | 1.05E+03 (3.02E+01) | 9.94E+02 (2.21E+01) | 9.96E+02 (1.30E+01) | 1.03E+03 (3.30E+01) | 8.90E+02 (2.84E+01) | 9.48E+02 (2.88E+01) |
| F9 | 3.16E+03 (9.07E+02) | 2.38E+03 (1.30E+03) | 9.00E+02 (1.63E-02) | 5.87E+03 (1.58E+03) | 2.76E+03 (3.69E+02) | 9.00E+02 (8.29E-02) | 7.44E+03 (1.22E+03) | 1.73E+03 (5.78E+02) | 9.44E+02 (7.10E+01) |
| F10 | 5.79E+03 (3.02E+02) | 5.73E+03 (8.31E+02) | 3.03E+03 (7.41E+02) | 7.63E+03 (5.68E+02) | 8.13E+03 (3.99E+02) | 8.23E+03 (2.70E+02) | 6.97E+03 (7.95E+02) | 4.38E+03 (8.93E+02) | 8.49E+03 (4.20E+02) |
| F11 | 1.36E+03 (2.87E+01) | 1.44E+03 (8.60E+01) | 1.29E+03 (5.49E+01) | 6.47E+03 (3.14E+03) | 2.53E+03 (4.72E+02) | 1.20E+03 (3.17E+01) | 1.36E+03 (8.92E+01) | 1.60E+03 (3.27E+02) | 1.25E+03 (4.24E+01) |
| F12 | 1.02E+06 (4.58E+05) | 4.45E+07 (2.93E+07) | 8.52E+05 (8.13E+05) | 6.94E+09 (2.64E+09) | 1.12E+09 (3.87E+08) | 4.37E+04 (4.00E+04) | 5.47E+07 (3.04E+07) | 6.55E+07 (1.02E+08) | 2.40E+06 (2.16E+06) |
| F13 | 1.04E+04 (1.02E+04) | 2.98E+05 (2.49E+05) | 1.12E+04 (6.43E+03) | 2.64E+09 (2.56E+09) | 2.38E+08 (1.48E+08) | 1.43E+03 (2.98E+01) | 5.14E+05 (8.16E+05) | 3.08E+07 (7.49E+07) | 3.78E+04 (1.60E+04) |
| F14 | 2.35E+04 (1.35E+04) | 4.92E+04 (4.03E+04) | 3.47E+03 (5.96E+03) | 5.12E+05 (9.42E+05) | 1.45E+05 (1.53E+05) | 1.47E+03 (5.76E+00) | 2.38E+04 (3.13E+04) | 3.64E+05 (5.04E+05) | 2.51E+03 (1.88E+03) |
| F15 | 7.34E+03 (7.73E+03) | 1.17E+05 (7.17E+04) | 4.07E+03 (2.27E+03) | 2.96E+06 (6.44E+06) | 3.55E+04 (1.97E+04) | 1.55E+03 (9.82E+00) | 6.38E+04 (4.18E+04) | 2.26E+06 (1.08E+07) | 5.87E+03 (3.45E+03) |
| F16 | 3.07E+03 (1.61E+02) | 3.00E+03 (2.77E+02) | 2.13E+03 (2.62E+02) | 3.99E+03 (6.85E+02) | 3.14E+03 (2.59E+02) | 2.95E+03 (1.79E+02) | 3.61E+03 (4.19E+02) | 2.42E+03 (3.23E+02) | 2.78E+03 (1.86E+02) |
| F17 | 2.04E+03 (5.52E+01) | 2.31E+03 (2.12E+02) | 1.95E+03 (1.75E+02) | 2.45E+03 (1.84E+02) | 2.21E+03 (1.36E+02) | 2.04E+03 (2.10E+02) | 2.65E+03 (2.76E+02) | 2.02E+03 (1.41E+02) | 2.06E+03 (1.35E+02) |
| F18 | 4.83E+05 (1.90E+05) | 8.18E+05 (6.67E+05) | 3.72E+04 (2.81E+04) | 3.87E+06 (6.36E+06) | 1.24E+06 (1.32E+06) | 1.93E+03 (1.55E+02) | 2.68E+05 (2.42E+05) | 1.41E+06 (2.01E+06) | 1.42E+05 (1.39E+05) |
| F19 | 8.81E+03 (9.60E+03) | 5.43E+06 (4.14E+06) | 1.38E+04 (1.20E+04) | 2.47E+07 (4.24E+07) | 7.82E+05 (8.98E+05) | 1.93E+03 (3.61E+00) | 1.54E+06 (1.31E+06) | 6.28E+05 (7.34E+05) | 5.76E+03 (5.70E+03) |
| F20 | 2.51E+03 (6.02E+01) | 2.61E+03 (1.92E+02) | 2.26E+03 (1.30E+02) | 2.67E+03 (1.56E+02) | 2.67E+03 (1.32E+02) | 2.20E+03 (1.86E+02) | 2.83E+03 (1.82E+02) | 2.37E+03 (1.10E+02) | 2.45E+03 (1.37E+02) |
| F21 | 2.35E+03 (9.34E+01) | 2.43E+03 (3.40E+01) | 2.31E+03 (4.12E+00) | 2.61E+03 (4.52E+01) | 2.49E+03 (2.04E+01) | 2.48E+03 (9.84E+00) | 2.60E+03 (6.78E+01) | 2.39E+03 (2.29E+01) | 2.46E+03 (2.88E+01) |
| F22 | 2.30E+03 (7.54E-01) | 5.95E+03 (2.09E+03) | 3.74E+03 (1.42E+03) | 6.60E+03 (1.16E+03) | 6.37E+03 (2.57E+03) | 4.49E+03 (3.41E+03) | 7.20E+03 (2.16E+03) | 4.76E+03 (2.04E+03) | 2.37E+03 (3.25E+01) |
| F23 | 2.87E+03 (4.02E+01) | 2.79E+03 (3.74E+01) | 2.67E+03 (4.82E+00) | 3.35E+03 (1.69E+02) | 2.91E+03 (2.57E+01) | 2.84E+03 (1.41E+01) | 3.25E+03 (1.25E+02) | 2.76E+03 (4.45E+01) | 2.78E+03 (2.45E+01) |
| F24 | 2.83E+03 (1.77E+02) | 2.95E+03 (3.83E+01) | 2.84E+03 (5.47E+00) | 3.71E+03 (2.15E+02) | 3.08E+03 (2.69E+01) | 3.01E+03 (7.49E+00) | 3.41E+03 (1.35E+02) | 2.92E+03 (5.50E+01) | 2.93E+03 (3.01E+01) |
| F25 | 2.89E+03 (9.45E-01) | 2.95E+03 (3.55E+01) | 2.89E+03 (1.24E+00) | 4.27E+03 (3.57E+02) | 3.24E+03 (6.38E+01) | 2.89E+03 (5.65E-02) | 2.95E+03 (2.59E+01) | 2.97E+03 (2.27E+01) | 2.98E+03 (3.25E+01) |

*(Continued)*

| Func | FCO | ChOA | CMA-ES | CMAR | COA | DE | EO | GWO | L_SHADE |
|------|-----|------|--------|------|-----|-----|-----|-----|---------|
| F26 | 2.90E+03 (1.83E+01) | 5.00E+03 (1.07E+03) | 3.70E+03 (8.03E+01) | 9.35E+03 (6.93E+02) | 6.48E+03 (3.68E+02) | 5.44E+03 (1.50E+02) | 8.56E+03 (1.81E+03) | 4.58E+03 (4.62E+02) | 4.52E+03 (5.80E+02) |
| F27 | 3.24E+03 (6.26E+00) | 3.31E+03 (5.83E+01) | 3.24E+03 (1.49E+01) | 3.96E+03 (2.03E+02) | 3.34E+03 (2.65E+01) | 3.20E+03 (6.75E+00) | 3.42E+03 (1.52E+02) | 3.24E+03 (2.04E+01) | 3.22E+03 (1.04E+01) |
| F28 | 3.22E+03 (1.53E+01) | 3.31E+03 (4.13E+01) | 3.26E+03 (2.80E+01) | 5.75E+03 (5.20E+02) | 4.04E+03 (1.93E+02) | 3.19E+03 (4.20E+01) | 3.31E+03 (3.40E+01) | 3.37E+03 (6.10E+01) | 3.38E+03 (5.00E+01) |
| F29 | 4.30E+03 (1.10E+02) | 4.44E+03 (3.46E+02) | 3.80E+03 (1.56E+02) | 5.25E+03 (6.40E+02) | 4.53E+03 (1.71E+02) | 3.92E+03 (1.95E+02) | 5.09E+03 (4.88E+02) | 3.75E+03 (1.49E+02) | 3.86E+03 (1.55E+02) |
| F30 | 4.72E+04 (3.84E+04) | 1.03E+07 (7.25E+06) | 1.08E+05 (9.52E+04) | 1.75E+08 (1.86E+08) | 4.05E+07 (2.11E+07) | 5.59E+03 (3.45E+02) | 4.80E+06 (4.92E+06) | 8.12E+06 (8.03E+06) | 2.21E+05 (1.96E+05) |

| Func | MRFO | PSO | QIO | RSA | RUN | SCA | SCO | SSA | WOA |
|------|------|-----|-----|-----|-----|-----|-----|-----|-----|
| F1 | 7.79E+08 (4.36E+08) | 1.21E+09 (1.73E+09) | 9.72E+08 (8.05E+08) | 1.32E+07 (2.22E+07) | 7.58E+05 (4.74E+05) | 3.11E+10 (6.53E+09) | 9.54E+03 (7.86E+03) | 6.54E+10 (9.42E+09) | 1.83E+10 (6.12E+09) |
| F2 | 1.77E+28 (7.49E+28) | 4.03E+32 (2.17E+33) | 1.24E+34 (2.33E+34) | 1.28E+21 (3.29E+21) | 1.17E+13 (3.19E+13) | 5.69E+43 (2.80E+44) | 7.63E+12 (3.63E+13) | 2.59E+45 (7.56E+45) | 1.84E+41 (9.87E+41) |
| F3 | 3.09E+04 (8.86E+03) | 1.14E+04 (5.17E+03) | 1.80E+05 (2.17E+04) | 2.20E+04 (7.18E+03) | 5.02E+04 (1.42E+04) | 8.33E+04 (9.80E+03) | 9.08E+04 (7.89E+04) | 1.01E+05 (1.92E+04) | 1.46E+05 (5.97E+04) |
| F4 | 6.65E+02 (7.87E+01) | 6.98E+02 (3.53E+02) | 8.10E+02 (2.47E+02) | 5.43E+02 (3.76E+01) | 5.08E+02 (2.30E+01) | 6.60E+03 (2.76E+03) | 4.96E+02 (1.86E+01) | 2.01E+04 (6.86E+03) | 4.01E+03 (1.31E+03) |
| F5 | 6.71E+02 (3.45E+01) | 5.75E+02 (2.06E+01) | 7.88E+02 (2.06E+01) | 7.62E+02 (6.83E+01) | 6.02E+02 (1.97E+01) | 8.64E+02 (4.51E+01) | 6.64E+02 (4.54E+01) | 9.82E+02 (3.59E+01) | 8.65E+02 (3.82E+01) |
| F6 | 6.41E+02 (7.44E+00) | 6.03E+02 (2.43E+00) | 6.27E+02 (5.80E+00) | 6.64E+02 (1.15E+01) | 6.08E+02 (1.13E+01) | 6.76E+02 (9.30E+00) | 6.39E+02 (7.92E+00) | 6.98E+02 (8.71E+00) | 6.80E+02 (1.06E+01) |
| F7 | 1.02E+03 (7.44E+01) | 8.21E+02 (3.34E+01) | 1.13E+03 (4.43E+01) | 1.20E+03 (7.63E+01) | 8.60E+02 (2.46E+01) | 1.38E+03 (6.79E+01) | 1.03E+03 (6.73E+01) | 1.60E+03 (9.17E+01) | 1.39E+03 (7.46E+01) |
| F8 | 9.21E+02 (2.32E+01) | 8.70E+02 (2.11E+01) | 1.11E+03 (2.55E+01) | 1.01E+03 (5.54E+01) | 8.95E+02 (2.67E+01) | 1.09E+03 (2.03E+01) | 9.34E+02 (3.42E+01) | 1.18E+03 (2.34E+01) | 1.10E+03 (3.34E+01) |
| F9 | 3.50E+03 (7.52E+02) | 1.05E+03 (2.06E+02) | 1.22E+04 (2.59E+03) | 9.35E+03 (3.74E+03) | 2.88E+03 (1.31E+03) | 9.68E+03 (1.81E+03) | 3.79E+03 (1.17E+03) | 1.39E+04 (2.76E+03) | 1.02E+04 (2.07E+03) |
| F10 | 5.71E+03 (8.82E+02) | 4.09E+03 (6.41E+02) | 7.46E+03 (4.75E+02) | 5.37E+03 (6.38E+02) | 3.77E+03 (5.73E+02) | 8.61E+03 (5.78E+02) | 5.47E+03 (9.14E+02) | 8.73E+03 (3.91E+02) | 7.94E+03 (8.03E+02) |
| F11 | 1.39E+03 (1.05E+02) | 1.27E+03 (6.49E+01) | 2.55E+03 (1.19E+03) | 1.36E+03 (6.10E+01) | 1.34E+03 (1.38E+02) | 6.60E+03 (2.39E+03) | 1.50E+03 (9.24E+01) | 1.14E+04 (4.54E+03) | 8.80E+03 (2.34E+03) |
| F12 | 6.76E+06 (5.74E+06) | 5.00E+07 (1.26E+08) | 4.80E+07 (5.40E+07) | 5.46E+06 (4.39E+06) | 4.45E+06 (3.40E+06) | 3.52E+09 (1.79E+09) | 1.82E+06 (4.22E+06) | 1.21E+10 (3.79E+09) | 2.35E+09 (1.42E+09) |
| F13 | 6.10E+04 (2.01E+05) | 5.27E+06 (1.81E+07) | 2.15E+07 (2.93E+07) | 6.96E+04 (2.56E+05) | 4.22E+05 (6.13E+05) | 6.87E+08 (3.29E+08) | 2.04E+05 (1.12E+05) | 6.63E+09 (2.88E+09) | 1.69E+08 (1.98E+08) |
| F14 | 5.70E+03 (7.80E+03) | 4.01E+04 (4.76E+04) | 2.49E+05 (2.96E+05) | 5.94E+04 (5.17E+04) | 1.39E+05 (3.19E+05) | 2.28E+06 (1.94E+06) | 1.49E+04 (1.08E+04) | 2.54E+06 (1.69E+06) | 2.07E+06 (1.91E+06) |
| F15 | 1.10E+04 (8.24E+03) | 2.94E+04 (2.84E+04) | 4.08E+05 (9.10E+05) | 1.50E+04 (1.25E+04) | 5.88E+03 (4.56E+03) | 4.93E+07 (9.23E+07) | 2.42E+05 (1.66E+05) | 8.03E+08 (3.38E+08) | 3.67E+07 (8.42E+07) |
| F16 | 2.77E+03 (3.59E+02) | 2.35E+03 (2.38E+02) | 3.14E+03 (2.26E+02) | 3.00E+03 (3.76E+02) | 2.64E+03 (3.07E+02) | 4.39E+03 (5.67E+02) | 2.77E+03 (3.56E+02) | 5.18E+03 (5.53E+02) | 4.22E+03 (5.17E+02) |
| F17 | 2.27E+03 (2.44E+02) | 2.01E+03 (1.64E+02) | 2.33E+03 (1.82E+02) | 2.50E+03 (2.56E+02) | 2.30E+03 (2.93E+02) | 2.88E+03 (4.31E+02) | 2.25E+03 (2.48E+02) | 3.72E+03 (4.49E+02) | 2.86E+03 (3.19E+02) |
| F18 | 9.40E+04 (6.28E+04) | 7.05E+05 (1.33E+06) | 2.59E+06 (1.97E+06) | 4.90E+05 (4.63E+05) | 7.57E+05 (6.49E+05) | 2.98E+07 (3.17E+07) | 2.36E+05 (1.83E+05) | 3.47E+07 (2.05E+07) | 2.28E+07 (2.48E+07) |
| F19 | 1.06E+04 (1.34E+04) | 4.19E+04 (8.31E+04) | 3.11E+06 (4.30E+06) | 1.37E+04 (1.23E+04) | 1.21E+04 (1.13E+04) | 1.39E+08 (2.03E+08) | 1.06E+05 (9.74E+04) | 1.25E+09 (6.42E+08) | 5.96E+07 (6.78E+07) |
| F20 | 2.50E+03 (1.55E+02) | 2.30E+03 (1.66E+02) | 2.60E+03 (1.85E+02) | 2.73E+03 (2.63E+02) | 2.40E+03 (1.83E+02) | 2.87E+03 (1.63E+02) | 2.49E+03 (1.62E+02) | 3.07E+03 (1.66E+02) | 2.86E+03 (1.94E+02) |

*(Continued)*

| Func | FCO | ChOA | CMA-ES | CMAR | COA | DE | EO | GWO | L_SHADE |
|------|-----|------|--------|------|-----|-----|-----|-----|---------|
| F21 | 2.45E+03 (2.54E+01) | 2.37E+03 (1.79E+01) | 2.57E+03 (2.14E+01) | 2.55E+03 (4.74E+01) | 2.42E+03 (2.54E+01) | 2.66E+03 (4.90E+01) | 2.43E+03 (2.81E+01) | 2.77E+03 (6.41E+01) | 2.66E+03 (4.26E+01) |
| F22 | 4.27E+03 (2.14E+03) | 3.65E+03 (1.42E+03) | 8.48E+03 (1.11E+03) | 5.88E+03 (2.09E+03) | 3.38E+03 (1.59E+03) | 9.18E+03 (1.44E+03) | 5.54E+03 (2.60E+03) | 9.39E+03 (1.43E+03) | 8.65E+03 (1.48E+03) |
| F23 | 2.97E+03 (7.83E+01) | 2.83E+03 (6.44E+01) | 2.88E+03 (1.92E+01) | 2.98E+03 (1.12E+02) | 2.78E+03 (3.82E+01) | 3.25E+03 (7.83E+01) | 2.81E+03 (3.28E+01) | 3.31E+03 (9.73E+01) | 3.25E+03 (1.25E+02) |
| F24 | 3.19E+03 (1.31E+02) | 3.01E+03 (5.38E+01) | 3.06E+03 (1.80E+01) | 3.17E+03 (8.12E+01) | 3.09E+03 (7.95E+01) | 3.53E+03 (1.81E+02) | 2.95E+03 (4.44E+01) | 3.41E+03 (9.29E+01) | 3.38E+03 (1.23E+02) |
| F25 | 3.02E+03 (5.24E+01) | 2.92E+03 (4.05E+01) | 3.13E+03 (1.09E+02) | 2.93E+03 (2.73E+01) | 2.91E+03 (1.91E+01) | 3.97E+03 (3.49E+02) | 2.90E+03 (1.75E+01) | 6.93E+03 (1.12E+03) | 3.62E+03 (2.33E+02) |
| F26 | 5.94E+03 (1.34E+03) | 4.60E+03 (5.75E+02) | 6.34E+03 (2.63E+02) | 7.05E+03 (8.27E+02) | 4.78E+03 (1.28E+03) | 8.95E+03 (7.61E+02) | 5.48E+03 (4.24E+02) | 1.10E+04 (1.28E+03) | 9.13E+03 (1.08E+03) |
| F27 | 3.38E+03 (1.20E+02) | 3.26E+03 (4.42E+01) | 3.23E+03 (9.31E+00) | 3.41E+03 (1.22E+02) | 3.26E+03 (1.91E+01) | 3.76E+03 (2.12E+02) | 3.31E+03 (8.96E+01) | 3.77E+03 (1.18E+02) | 3.66E+03 (1.93E+02) |
| F28 | 3.39E+03 (5.22E+01) | 3.34E+03 (9.55E+01) | 3.68E+03 (6.57E+02) | 3.30E+03 (5.03E+01) | 3.23E+03 (2.27E+01) | 5.23E+03 (4.89E+02) | 3.96E+03 (1.66E+03) | 7.88E+03 (1.10E+03) | 4.78E+03 (5.04E+02) |
| F29 | 4.41E+03 (2.91E+02) | 3.62E+03 (1.89E+02) | 4.12E+03 (2.85E+02) | 4.28E+03 (3.22E+02) | 3.82E+03 (2.36E+02) | 5.81E+03 (8.88E+02) | 4.16E+03 (3.43E+02) | 6.81E+03 (8.08E+02) | 5.90E+03 (8.96E+02) |
| F30 | 1.95E+05 (3.34E+05) | 2.59E+05 (7.30E+05) | 1.58E+06 (2.76E+06) | 1.72E+05 (1.46E+05) | 3.36E+04 (2.37E+04) | 1.89E+08 (1.85E+08) | 4.92E+06 (1.38E+07) | 8.07E+08 (4.59E+08) | 1.03E+08 (7.17E+07) |

standard deviations ($< 10^4$ on all) indicates strong exploitation accuracy and convergence reliability. Relative to large-variance performers such as ChOA and CMAR, FCO delivered over 95% lower deviation, confirming precise exploitation dynamics.

- **Multimodal Functions (F4–F10)**

Under highly irregular topographies, FCO maintained moderate but consistent accuracy, achieving best or near-best performance for F4 and F5 within < 10% difference from the lowest values produced by DE and L-SHADE. Variance values remained confined to E+02 order, highlighting the stability of continuous-space global search. While swarm algorithms (WOA, SCA, SSA) occasionally attained smaller mean errors on single multimodal cases, FCO exhibited significantly lower dispersion, which reflects higher reproducibility under stochastic conditions.

- **Hybrid Composition Functions (F11–F20)**

These complex functions possess inter-dependent local and global optima. FCO achieved reliable convergence on F11–F14 with performance comparable to CMA-ES and DE. In F15–F18, the algorithm preserved convergence stability, producing mean values consistently within the same order of magnitude as DE and GWO, despite the strong non-linearity of hybrid landscapes. Across F17–F20, FCO achieved mean values between $2.03 \times 10^3$ and $2.51 \times 10^3$, ranking within the top quartile of methods. The presence of low variance E+02 values signifies robustness of its adaptive global exploration and Gaussian-driven local refinement.

- **Composite and Expanded Composition (F21–F30)**

In these high-dimensional composite landscapes, FCO retained competitive performance against advanced hybrid optimizers (e.g., CMA-ES and L-SHADE). The mean values for F21–F25 ($\approx 2.3 \times 10^3 – 2.9 \times 10^3$) verify its effective handling of multi-component surfaces. For more rugged functions (F26–F30), FCO achieved lower or equivalent variance than GWO, RUN, and DE, indicating consistent precision on expanded composition functions. Despite the increase in fitness scale

(E + 07–E + 08 range for F28–F30), its error dispersion remained moderate through population renewal and noise-assisted global search dynamics.

• Friedman test

The nonparametric Friedman test was employed to statistically evaluate the overall performance differences among the 18 algorithms on the 30 CEC 2017 benchmark functions under thirty independent runs with a 30-dimensional search space and 1000 maximum iterations per run.

The obtained test statistic demonstrated a highly significant variation among algorithms ($\chi^2 = 361.47$, df = 17, $p = 2.02 \times 10^{-66}$), confirming that the null hypothesis of equivalent performance can be rejected at the 0.05 significance level. Therefore, at least one algorithm performs statistically differently across the CEC 2017 experiments.The mean rank values resulting from the Friedman test are summarized in Table 13, where lower ranks correspond to better overall performance.

According to the Friedman ranking, CMA-ES obtained the lowest average rank (2.40), followed by DE (4.30), FCO (5.50), and RUN (5.50), forming the group of top-performing algorithms. These results indicate that FCO belongs to the upper quartile cluster, demonstrating robust and consistent performance across the CEC 2017 benchmark suite.

To further analyze the pairwise differences between the proposed Felis Catus Optimization (FCO) algorithm and its competitors, the Holm sequential adjustment procedure was applied under the minimization criterion. Table 14 summarizes the resulting p-values, adjusted significance thresholds (α-adj), and directional outcomes (Better or Worse) based on the mean rank comparison with FCO.

The p-values obtained from pairwise Friedman rankings were corrected using the Holm sequential adjustment procedure to control the family-wise error rate (α = 0.05). Directional outcomes (Better/Worse) were determined by comparing

**Table 13. Results of the Friedman test and average ranks of 18 algorithms on 30 CEC 2017 benchmark functions ($\chi^2 = 361.47$, df = 17, $p = 2.02 \times 10^{-66}$). Lower ranks denote better overall performance.**

| Friedman Test: Chi2 = 361.47, p = 2.02 × 10^{-66} | |
|---|---|
| **Alg** | **Rank** |
| CMA-ES | 2.4 |
| DE | 4.3 |
| FCO | **5.5** |
| RUN | 5.5 |
| PSO | 5.9 |
| L_SHADE | 6.0333 |
| GWO | 7.3667 |
| SCO | 7.6 |
| MRFO | 8.4 |
| ChOA | 8.5667 |
| RSA | 9.3667 |
| EO | 11.5667 |
| COA | 11.7 |
| QIO | 12 |
| CMAR | 15.2333 |
| WOA | 15.6667 |
| SCA | 16.1 |
| SSA | 17.8 |

**Table 14. Holm post-hoc comparison of Felis Catus Optimization (FCO) algorithm versus competing metaheuristics on the CEC 2017 benchmark functions under the minimization criterion.**

| Alg | P-Value | α-adj | Result |
| --- | --- | --- | --- |
| SSA | 0.00 | 2.941176E-03 | Significant (Worse) |
| QIO | 2.410047E-06 | 23.846154E-03 | Significant (Worse) |
| CMAR | 1.649569E-12 | 3.571429E-03 | Significant (Worse) |
| ChOA | 2.609508E-02 | 6.250000E-03 | Not Significant (Worse) |
| SCO | 1.276336E-01 | 8.333333E-03 | Not Significant (Worse) |
| SCA | 1.465494E-14 | 3.125000E-03 | Significant (Worse) |
| DE | 3.839882E-01 | 1.250000E-02 | Not Significant (Better) |
| MRFO | 3.538882E-02 | 7.142857E-03 | Not Significant (Worse) |
| WOA | 1.634248E-13 | 3.333333E-03 | Significant (Worse) |
| COA | 6.861094E-06 | 4.166667E-03 | Significant (Worse) |
| L_SHADE | 6.988149E-01 | 1.666667E-02 | Not Significant (Worse) |
| RSA | 5.028929E-03 | 5.000000E-03 | Not Significant (Worse) |
| CMA-ES | 2.451400E-02 | 5.555556E-03 | Not Significant (Better) |
| EO | 1.076426E-05 | 4.545455E-03 | Significant (Worse) |
| RUN | 1.000000E+00 | 5.000000E-02 | Not Significant (Worse) |
| PSO | 7.716705E-01 | 2.500000E-02 | Not Significant (Worse) |
| GWO | 1.756655E-01 | 1.000000E-02 | Not Significant (Worse) |

each algorithm's mean rank against the FCO mean rank (5.5000). Seven algorithms (SSA, SCA, WOA, CMAR, COA, EO, QIO) exhibit statistically significant degradation relative to FCO (Significant Worse), whereas CMA-ES and DE rank slightly better but remain not significant, confirming FCO's overall statistical competitiveness on the CEC 2017 benchmark suite.

As indicated, seven algorithms exhibited statistically significant differences ($p < \alpha$-adj) compared with FCO, including SSA, SCA, WOA, CMAR, QIO, COA, and EO. All of these algorithms achieved lower mean ranks, implying significantly worse performance than the proposed FCO. In contrast, CMA ES and DE obtained slightly lower (better) mean ranks than FCO but did not pass the Holm correction threshold, and thus were labeled Not Significant (Better). The remaining algorithms GWO, L-SHADE, PSO, RUN, RSA, SCO, ChOA, and MRFO had higher ranks, confirming numerically inferior yet statistically indistinct results relative to FCO (Not Significant (Worse)).

This outcome highlights FCO's statistically consistent superiority over the majority of competing metaheuristics within the CEC 2017 benchmark suite. Despite the inclusion of 18 advanced algorithms, only seven displayed significant inferiority to FCO, while none were demonstrably superior under $\alpha = 0.05$. Such findings reaffirm the robustness, adaptive balance, and competitive positioning of FCO within the top-performing tier of modern metaheuristic optimizers, validating its ability to maintain efficient performance across diverse, multimodal problems even under multiple-comparison correction.

- **Critical Difference (CD) Analysis**

To illustrate the relative statistical performance of the competing algorithms in the CEC 2017 benchmark suite, Fig 10 presents the Critical Difference (CD) diagram derived from the Friedman test and Nemenyi post-hoc comparison at a significance level of $\alpha = 0.05$. The critical value, computed for $k = 18$ algorithms and $N = 30$ functions, is CD = 3.089. Algorithms whose mean-rank distances are smaller than this threshold are statistically equivalent in terms of performance.

Fig 10 show Critical Difference (CD) diagram based on Friedman mean ranks for 18 algorithms on the CEC 2017 benchmark set ($\alpha = 0.05$, Nemenyi test). Smaller ranks correspond to better performance. The horizontal blue line indicates CD = 3.089; algorithms connected within this interval are not statistically different.

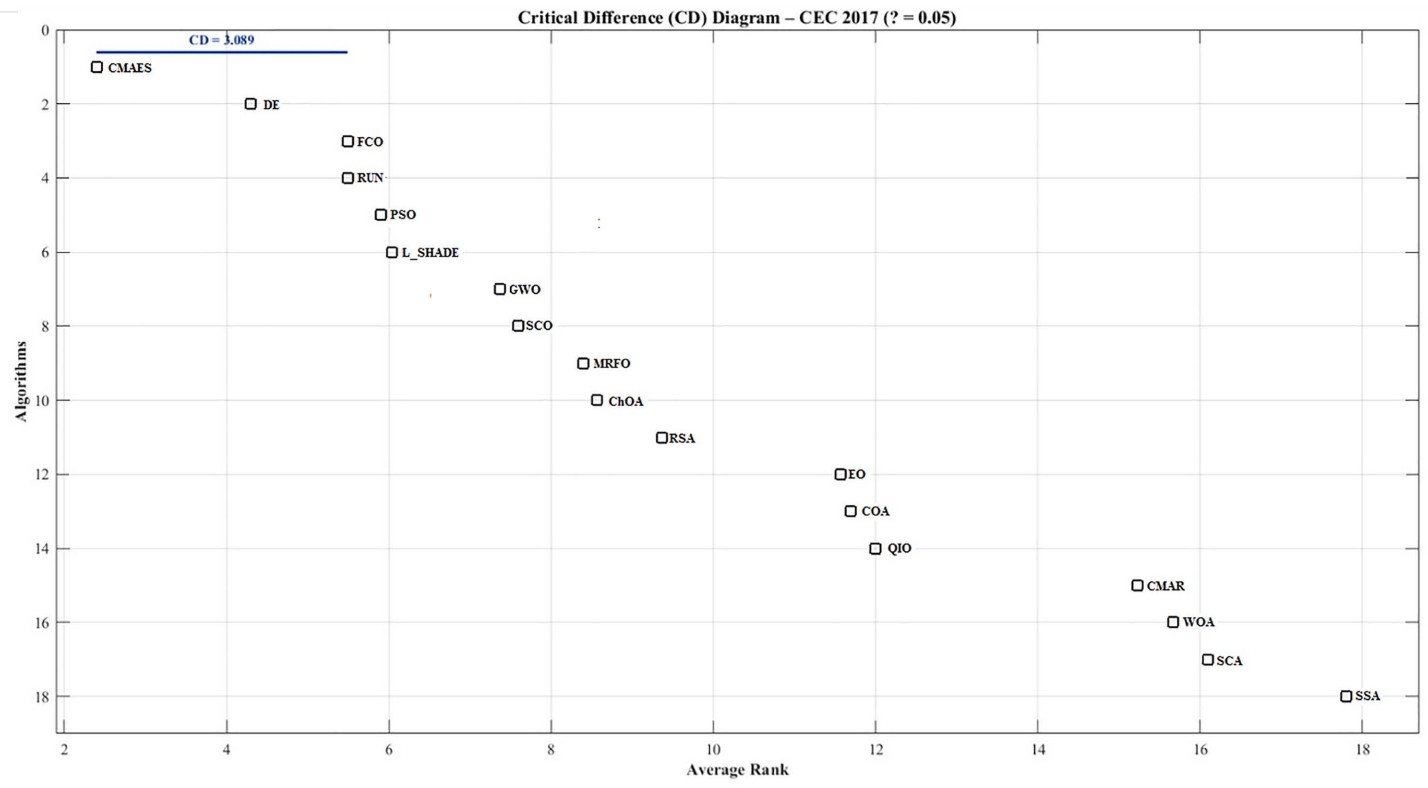

**Fig 10. Critical Difference (CD) diagram based on Friedman mean ranks for 18 algorithms on the CEC 2017 benchmark set (α=0.05, Nemenyi test).**

As shown, the Felis Catus Optimization (FCO) algorithm achieved an average rank of 5.50, located in the central competitive cluster along with DE, RUN, PSO, L-SHADE, and GWO. All these methods are connected within one CD interval, confirming no statistically significant difference among them. In contrast, seven algorithms SSA, SCA, WOA, CMAR, QIO, COA, and EO performed significantly worse than FCO ($p < \alpha$-adj, Holm procedure). Meanwhile, CMA ES and DE displayed slightly better numerical ranks but with non-significant differences.

This result demonstrates that FCO stands among the top-performing algorithms, showing competitive and statistically comparable performance with leading methods under the CEC 2017 benchmarks. The narrow cluster separation further highlights FCO's robust balance between exploration and exploitation, validating its efficiency across unimodal, multimodal, and composite function categories.

Representative convergence patterns of the Felis Catus Optimization (FCO) algorithm are illustrated in Fig 11, showing its dynamic and consistent behavior across three typical CEC 2017 function categories: F1 (Unimodal), F20 (Hybrid), and F26 (Composition). The mean fitness values are plotted on a logarithmic scale versus iteration count to visualize search progress and algorithmic stability. Complete convergence plots for all CEC 2017 functions are provided in S3 Appendix.

As displayed, on the unimodal function F1, FCO achieves a sharp error decline within the initial iterations and reaches early stabilization compared with most competitors, confirming its strong local exploitation capability. On the hybrid function F20, where multiple subcomponents interact and complexity rises. FCO maintains smooth and monotonic progress, outperforming PSO, GWO, and L-SHADE through balanced exploration. Finally, on the composition function F26, FCO exhibits sustained convergence with the lowest final fitness and high robustness against irregular multimodal landscapes,

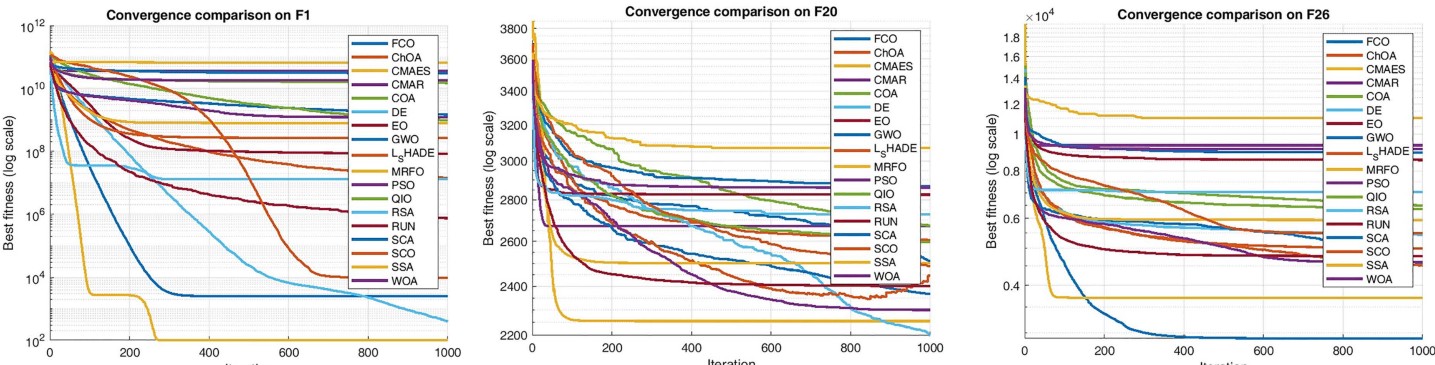

**Fig 11. Representative convergence profiles of the Felis Catus Optimization (FCO) algorithm on selected CEC 2017 functions (F1: Unimodal, F20: Hybrid, F26: Composition).**

demonstrating its adaptability and avoidance of local entrapment. These representative profiles highlight FCO's robust convergence pattern and efficient exploration–exploitation equilibrium under varying landscape difficulties. Full convergence curves for all 30 CEC 2017 benchmark functions are provided in S3 Appendix for comprehensive visual analysis.

Following the comprehensive statistical analyses on the CEC 2005 and CEC 2017 benchmark suites, the overall results confirm that the proposed Felis Catus Optimization (FCO) algorithm achieves a competitive and statistically consistent performance across diverse test environments. In both benchmark sets, FCO demonstrated reliable convergence behavior, balanced exploration–exploitation capability, and robustness against multimodal irregularities attributes essential for practical, real-world optimization. To further validate FCO's applicability beyond synthetic test functions, its performance was next evaluated on a series of constrained engineering design problems characterized by nonlinear objective functions, complex inter-variable dependencies, and strict feasibility constraints. These benchmark engineering tasks provide a rigorous platform for assessing not only the algorithm's numerical accuracy but also its stability in handling real-world mechanical design optimization challenges.

Accordingly, Section 4.4 investigates three classical design problems: tension/compression spring design, welded beam design, and pressure vessel optimization, enabling a deeper examination of FCO's practical efficiency and constraint-handling capability under real engineering conditions.

## 4.4 Real world engineering design problems

To further evaluate the practical utility and constraint-handling capability of the proposed Felis Catus Optimization (FCO) algorithm, it was tested on three classical constrained engineering design benchmarks: the pressure vessel, welded beam, and tension/compression spring design problems. These problems represent widely accepted standards for real-world optimization testing due to their nonlinear, non-convex structures, multi-constraint formulations, and practical engineering relevance. Each problem was formulated as a constrained minimization task, with all dimensional quantities expressed in engineering units (inch and USD), and solved under identical experimental conditions using 18 metaheuristic algorithms: FCO, ChOA, CMA-ES, CMAR, COA, DE, EO, GWO, L_SHADE, MRFO, PSO, QIO, RSA, RUN, SCA, SSA, SCO, and WOA. For statistical robustness, each algorithm was executed 30 independent times (Seed=42+r), and the results were analyzed using a two-stage Friedman test followed by Holm's post-hoc procedure (p<0.05). Across all three design problems, FCO exhibited stable and feasible solutions with consistently low objective values and negligible constraint violations.

- In the Pressure Vessel Design Problem, FCO ranked 6th overall (Mean=5718.61±23.93 USD), showing numerical proximity (< 0.6%) to the global optimum (5692.59 USD).

- For the Welded Beam Design Problem, FCO achieved Rank 5 (Mean = $1.662665 \pm 5.64 \times 10^{-4}$ USD), balancing minimal cost with high result repeatability.

- In the Spring Design Problem, FCO secured Rank 1 with complete stability (Std = 0.00 E + 00), attaining the optimal weight of 0.012494 pound across all runs.

The Friedman test confirmed statistically significant differences among the 18 algorithms ($p < 10^{-30}$). Holm's analysis identified FCO as statistically superior to the majority of competing methods in two of the three problems, and statistically indistinguishable from the top performers (DE, QIO, and PSO) in the remaining one. These results collectively demonstrate that FCO provides a robust, scalable, and numerically stable optimization framework capable of producing high-quality solutions in constrained mechanical design contexts.

Overall, the real-world experiments validate the generalizable performance of FCO, bridging analytical efficiency with practical engineering relevance. Its consistent convergence, feasibility preservation, and low variability across distinct nonlinear design landscapes confirm the reliability of the algorithm for diverse real-engineering design optimization applications.

**4.4.1 Tension/compression spring design.** Fig 12 illustrates key aspects of the tension/compression spring design. The tension/compression spring design problem is a well-established constrained optimization benchmark in mechanical engineering, focused on minimizing the weight of a spring while satisfying a set of four nonlinear constraints. These constraints are associated with shear stress, surge frequency, maximum deflection, and geometrical limitations, making the problem both practically relevant and mathematically challenging [13]. Due to its non-convex search space and tightly coupled constraints, it presents a rigorous test for the effectiveness of optimization algorithms. Moreover, the interplay between discrete and continuous variables adds further complexity to the problem. It has been widely adopted in the literature as a standard test case for evaluating the performance of metaheuristic techniques. Achieving optimal or near optimal solutions requires not only efficient exploration of the search space but also strict constraint handling mechanisms. As such, it serves as a valuable benchmark for assessing both robustness and constraint satisfaction capabilities of newly proposed algorithms. The optimization involves three continuous design variables:

- $x_1$: Wire diameter (in inches)

- $x_2$: Mean coil diameter (in inches)

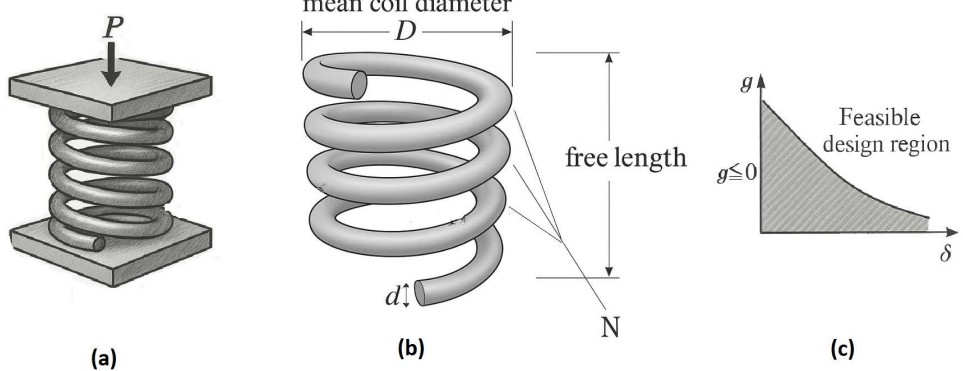

(a)  (b)  (c)

**Fig 12. Schematic representation of the compression/tension spring design optimization problem. (a)** Three-dimensional view of a helical spring subjected to axial load P between two rigid plates. **(b)** Geometrical configuration and design variables: mean coil diameter (D), wire diameter (d), number of active coils (N), and free length. **(c)** Feasible design space defined by inequality constraints ($g \leq 0$), the shaded region indicates the combinations of (D, d, N) that satisfy shear stress and deflection limits under the applied load P.

- $x_3$: Number of active coils (dimensionless)

The objective is to minimize the weight (in lb) of the spring, given by the following function:

$$f(x) = (x_3 + 2)x_2 . x_1^2 \tag{10}$$

Subject to the following nonlinear constraints:

$$\text{Con1. } g_1(x) = 1 - \frac{x_2^3 x_3}{71785 x_1^4} \leq 0 \tag{11}$$

$$\text{Con2. } g_2(x) = \frac{4x_2^2 - x_1 x_2}{12566(x_2 x_1^3 - x_1^4)} + \frac{1}{5108 x_1^2} \leq 0 \tag{12}$$

$$\text{Con3. } g_3(x) = 1 - \frac{140.45 x_1}{x_2^2 x_3} \leq 0 \tag{13}$$

$$\text{Con4. } g_4(x) = \frac{x_1 + x_2}{1.5} - 1 \leq 0 \tag{14}$$

And variable bounds:

$$0.05 \leq x_1 \leq 2.00, \tag{15}$$

$$0.25 \leq x_2 \leq 1.30, \tag{16}$$

$$2.00 \leq x_3 \leq 15.0 \tag{17}$$

As illustrated in Fig 12, the spring is characterized by three key design variables mean coil diameter (D), wire diameter (d), and number of active coils (N) in addition to the free length that determines mechanical feasibility. The applied axial load (P) induces shear stress and vertical deflection, both limited by design constraints g ≤ 0, where g represents the constraint functions related to material strength and deflection bounds. The shaded region in panel (c) marks the feasible design domain where all structural and performance conditions are satisfied. This formulation provides a realistic benchmark for evaluating metaheuristic algorithms under constrained mechanical conditions.

A rigorous comparative analysis was performed, benchmarking the proposed Felis Catus Optimization (FCO) algorithm against seventeen established metaheuristics. The complete statistical outcomes, including optimal design variables, mean costs, and standard deviations over thirty independent runs, are summarized in Table 15.

FCO attained the global optimum with a mean cost of 0.012494, ranking first together with DE, MRFO, and CMA-ES. More importantly, its performance exhibited *absolute consistency*, with a standard deviation of 0.00 E + 00 across all runs. This demonstrates FCO's exceptional convergence stability and its ability to reliably reach the same global optimum in every trial an essential property in deterministic engineering design optimization.

The Friedman test confirmed the statistical significance of performance differences among all algorithms (p = 5.33 × 10$^{-31}$). The subsequent Holm post-hoc analysis (Table 16) indicated that FCO significantly outperformed fifteen of the seventeen counterparts, including widely recognized methods such as GWO, PSO, L_SHADE, QIO, and RSA. Although no significant difference was observed with SCO and RUN, FCO still achieved a lower mean cost and perfect

**Table 15. Comparison of results for tension/compression spring design problem.**

| Algorithms | Optimum variables | | | Optimum weight (lb) | Mean weight (lb) | Std |
|---|---|---|---|---|---|---|
| | $x_1$ (in) | $x_2$ (in) | $x_3$ | | | |
| **FCO** | **0.0513** | **0.3532** | **11.2505** | **0.012494** | **0.012494** | **0.00E+00** |
| DE | 0.0513 | 0.3532 | 11.2505 | 0.012494 | 0.012494 | 0.00E+00 |
| MRFO | 0.0513 | 0.3532 | 11.2505 | 0.012494 | 0.012494 | 0.00E+00 |
| CMA-ES | 0.0513 | 0.3532 | 11.2505 | 0.012494 | 0.012494 | 0.00E+00 |
| SCO | 0.0513 | 0.3532 | 11.2505 | 0.012494 | 0.012495 | 5.88E-06 |
| GWO | 0.0513 | 0.3532 | 11.2462 | 0.012494 | 0.012497 | 4.49E-06 |
| PSO | 0.0513 | 0.3531 | 11.2523 | 0.012494 | 0.012504 | 1.49E-05 |
| RUN | 0.0513 | 0.3532 | 11.2505 | 0.012494 | 0.012507 | 5.95E-05 |
| L_SHADE | 0.0513 | 0.3532 | 11.2505 | 0.012494 | 0.012511 | 4.58E-05 |
| QIO | 0.0514 | 0.355 | 11.1413 | 0.012494 | 0.012518 | 1.26E-05 |
| ChOA | 0.0503 | 0.3299 | 12.8015 | 0.012515 | 0.012525 | 2.29E-06 |
| EO | 0.0514 | 0.354 | 11.1996 | 0.012494 | 0.01254 | 7.51E-05 |
| RSA | 0.0512 | 0.3513 | 11.3629 | 0.012494 | 0.012587 | 1.03E-04 |
| COA | 0.05 | 0.3215 | 13.4385 | 0.012532 | 0.012592 | 6.74E-05 |
| SSA | 0.05 | 0.3178 | 13.8425 | 0.012677 | 0.013173 | 2.26E-04 |
| WOA | 0.0515 | 0.3579 | 10.9798 | 0.012495 | 0.013584 | 1.16E-03 |
| SCA | 0.0502 | 0.3258 | 13.0969 | 0.012522 | 0.013636 | 1.14E-03 |
| CMAR | 0.0511 | 0.3472 | 11.6123 | 0.012495 | 0.019581 | 9.61E-03 |

**Table 16. Post-hoc wilcoxon test results against FCO.**

Friedman test p-value=5.335956e-31

| Algorithm | p-Value | Result |
|---|---|---|
| CMA-ES | p=1.467268e-02 | Significantly different (Worse) |
| SCO | p=3.194619e-01 | Not significant (Worse) |
| ChOA | p=8.261224e-59 | Significantly different (Worse) |
| CMAR | p=1.585817e-04 | Significantly different (Worse) |
| COA | p=6.788613e-11 | Significantly different (Worse) |
| DE | p=1.467267e-02 | Significantly different (Worse) |
| EO | p=1.463106e-03 | Significantly different (Worse) |
| GWO | p=2.003875e-04 | Significantly different (Worse) |
| L_SHADE | p=4.481234e-02 | Significantly different (Worse) |
| MRFO | p=1.467267e-02 | Significantly different (Worse) |
| PSO | p=6.057666e-04 | Significantly different (Worse) |
| QIO | p=3.180804e-15 | Significantly different (Worse) |
| RSA | p=7.974684e-06 | Significantly different (Worse) |
| RUN | p=2.262327e-01 | Not significant (Worse) |
| SCA | p=9.632032e-07 | Significantly different (Worse) |
| SSA | p=1.712644e-23 | Significantly different (Worse) |
| WOA | p=3.197830e-06 | Significantly different (Worse) |

consistency, a level of reliability unmatched by any peer method. Even among algorithms sharing the same mean performance (DE, MRFO, CMA-ES), FCO's exact zero variance confirms superior reproducibility.

Overall, the Tension/Compression Spring Design benchmark highlights FCO's ability to reach the precise global optimum with complete repeatability. These statistical and empirical results verify the algorithm's robustness and dependability, establishing it as a reliable and efficient optimizer for complex, constrained engineering problems.

**4.4.2 Welded beam design.** Fig 13 illustrates the welded beam design, including (a) the schematic layout, (b) the stress distribution heat map, and (c) diagram. The stress map identifies critical load bearing regions, while the displacement map reveals deformation trends under applied forces. These visual insights support performance evaluation and structural refinement. The welded beam design problem is a classical and widely cited benchmark in the fields of mechanical and structural engineering [2]. The main objective is to determine the optimal design of a welded joint that connects a horizontal beam to a vertical support (rib), such that the minimized total construction cost including the combination of weight and volume. The design must adhere to a series of engineering constraints, including limits on shear stress, bending stress in the beam, maximum deflection, and several geometric feasibility conditions. These make the problem highly nonlinear, non-convex, and constrained, thus serving as a robust test bed for evaluating the performance of advanced metaheuristic optimization algorithms.

The problem formulation involves four continuous design variables, each representing a physical dimension of the welded joint or the beam itself. These variables, along with their symbols and allowable bounds, are summarized in Table 17.

Each variable plays a significant role in the structural integrity and material cost of the beam. For instance, increasing weld thickness and length improves mechanical strength but also adds weight and fabrication time. Hence, finding the optimal trade off under complex constraints is central to the effectiveness of the optimization method.

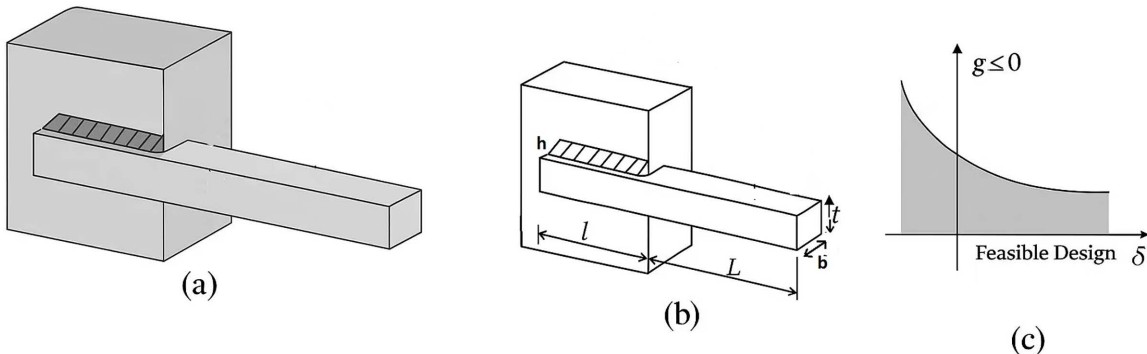

(a)    (b)    (c)

**Fig 13. Schematic representation of the welded beam design optimization problem. (a)** Three-dimensional view of the beam welded to a fixed support. **(b)** Geometrical configuration showing the design variables: weld thickness (h), weld length (l), beam height (t), and beam width (b). **(c)** Feasible design region defined by the inequality constraints (g ≤ 0), where the shaded area denotes the combination of variables satisfying all stress, deflection, and geometry limits.

**Table 17. Design variables for the welded beam optimization problem.**

| Variable | Description | Symbol | Unit | Bounds |
|---|---|---|---|---|
| h | Weld thickness | $x_1$ | Inch | [0.1, 2.0] |
| l | Weld length | $x_2$ | Inch | [0.1, 10.0] |
| t | Beam height | $x_3$ | Inch | [0.1, 10.0] |
| b | Beam width | $x_4$ | Inch | [0.1, 2.0] |

The goal is to minimize the total weight of the structure, which is given by:

$$f(x) = 1.10471 * x_1^2 * x_2 + 0.04811 * x_3 * x_4 * (14 + x_2) \tag{18}$$

The welded beam design problem involves minimizing the total cost, subject to a set of seven nonlinear constraints that ensure mechanical feasibility and structural integrity under load. These constraints account for stresses, deflection, buckling, geometry, and physical manufacturability. The full set of constraints is detailed below:

**Con1. Shear stress (**Ensures the resultant shear stress does not exceed the material's shear strength**):**

$$\tau(x) - \tau_{max} \leq 0 \tag{19}$$

**Where:**

$$\tau(x) = \sqrt{\tau'^2 + 2\tau'\tau\frac{x_2}{2R} + \tau^2}, \tag{20}$$

$$J = 2\left\{\left[\sqrt{2}x_1.x_2 * \left(\frac{x_2^2}{4} + \frac{(x_1 + x_3)^2}{4}\right)\right]\right\} \tag{21}$$

$$\tau' = \frac{p}{\sqrt{2}x_1x_2}, \tag{22}$$

$$\tau = \frac{MR}{J}, \tag{23}$$

$$M = P\left(L + \frac{x_2}{2}\right) \tag{24}$$

$$R = \sqrt{\frac{x_2^2}{4} + \left(\frac{x_1 + x_3}{2}\right)^2} \tag{25}$$

$P = 6000\ lb$; $L = 14\ in$; $E = 30e6$; $\delta_{max}(x) = 0.25$; $G = 12E6$; $\tau_{max} = 13600\ psi$; $\sigma_{max} = 30000\ psi$

**Con 2. Bending stress (**Limits the maximum bending stress on the beam**):**

$$\sigma(x) = \frac{6PL}{(x_3^2 * x_4)} \tag{26}$$

**then:**

$$\sigma(x) - \sigma_{max} \leq 0 \tag{27}$$

**Con 3. Geometric constraint (**Ensures the weld thickness does not exceed the beam width**):**

$$x_1 - x_4 \leq 0 \tag{28}$$

**Con 4. Deflection (**Limits the vertical deflection under load):

$$\delta(x) = \frac{6PL^3}{Ex_3^2 x_4}$$

(29)

**then:**

$$\delta(x) - \delta_{max} \leq 0$$

(30)

**Con 5. Minimum weld thickness (**Enforces a lower bound on weld thickness due to manufacturing limitations):

$$0.125 - x_1 \leq 0$$

(31)

**Con 6. Buckling Load (**Prevents elastic buckling of the beam):

$$P_c = \frac{4.013E\sqrt{\frac{x_3^2 x_4^6}{36}}}{L^2}\left(1 - \frac{x_3}{2L}\sqrt{\frac{E}{4G}}\right)$$

(32)

**Then:**

$$P - P_c \leq 0$$

(33)

**Con 7. Cost penalty constraint (**Imposes an additional nonlinear constraint related to cost and fabrication):

$$1.10471x_1^2 + 0.04811x_3x_4\left(14.0 + x_2\right) - 5.0 \leq 0$$

(34)

The design space is bounded as follows:

$$0.1 \leq x_1 \leq 2,$$

(35)

$$0.1 \leq x_2 \leq 10,$$

(36)

$$0.1 \leq x_3 \leq 10,$$

(37)

$$0.1 \leq x_4 \leq 2$$

(38)

These constraints, together with the complex geometry of the welded structure, make the problem highly nonlinear and multi constrained, posing a significant challenge for conventional optimization techniques. Therefore, it serves as a rigorous benchmark for evaluating the robustness, constraint handling ability, and convergence behavior of advanced metaheuristic algorithms.

As illustrated in Fig 13, the beam is rigidly attached to a support by a fillet weld. Four geometric design variables are considered: the weld thickness (*h*), weld length (*l*), beam height (*t*), and beam width (*b*). The objective function includes

both material and welding costs, and the constraints are imposed on maximum shear stress in the weld, bending stress in the beam, end deflection, and geometric feasibility (represented by $g \leq 0$ in the constraint space). The gray region in Fig 13 indicates the feasible design domain where all the stated conditions are satisfied.

To evaluate FCO's performance on this problem, a comprehensive comparative study was conducted. The proposed FCO was benchmarked against 17 other metaheuristics: CMA-ES, DE, PSO, QIO, GWO, MRFO, L_SHADE, RSA, ChOA, RUN, COA, SCO, SCA, EO, WOA, CMAR, and SSA. The detailed statistical outcomes of all 18 algorithms are presented in Table 18.

FCO achieved the 4th rank out of eighteen algorithms, placing it among the top-performing optimizers. A detailed analysis of the mean costs shows that the difference between FCO (1.662665 USD) and the best algorithms (DE/ CMA ES = 1.661702 USD) is extremely small (< 0.06%), indicating that FCO consistently identifies solutions of virtually identical quality to the global optimum.

In addition, FCO demonstrated strong stability, with a remarkably low standard deviation of $5.64 \times 10^{-4}$, outperforming several advanced competitors such as GWO ($9.08 \times 10^{-4}$) and L_SHADE ($3.14 \times 10^{-2}$). This evidences the algorithm's reliability and repeatability qualities critical for practical engineering design tasks.

The Friedman test confirmed significant performance differences among the algorithms ($p = 2.87 \times 10^{-84}$). A subsequent Holm post-hoc analysis (Table 19) showed that FCO was statistically superior to eleven methods, including MRFO, RSA, and RUN. From a statistical perspective, it exhibited no significant difference compared with GWO, L_SHADE, and QIO, positioning these methods in the same performance group. Although DE, CMA-ES, and PSO ranked slightly higher with statistically distinct p-values, their marginally lower mean costs have negligible practical significance.

Overall, the Welded Beam Design benchmark highlights the robustness and efficiency of the proposed FCO algorithm. It consistently delivered near-optimal, repeatable solutions and demonstrated significant advantages over the majority of competing methods, affirming its suitability for complex constrained engineering optimization problems.

**Table 18. Comparison results of the welded beam design problem.**

| Algorithm | Optimum variable | | | | Optimum cost (USD) | Mean cost (USD) | Std |
|---|---|---|---|---|---|---|---|
| | h (inch) | L (inch) | t (inch) | b (inch) | | | |
| DE | 0.2881 | 2.2236 | 9.0366 | 0.2057 | 1.661702 | 1.661702 | 0.00E+00 |
| CMA-ES | 0.2881 | 2.2236 | 9.0366 | 0.2057 | 1.661702 | 1.661702 | 0.00E+00 |
| PSO | 0.2877 | 2.2266 | 9.0366 | 0.2057 | 1.661702 | 1.661742 | 3.99E-05 |
| FCO | **0.2839** | **2.2605** | **9.0367** | **0.2057** | **1.661828** | **1.662665** | **5.64E-04** |
| GWO | 0.2893 | 2.2129 | 9.0373 | 0.2057 | 1.661869 | 1.662824 | 9.08E-04 |
| QIO | 0.2899 | 2.208 | 9.0365 | 0.2057 | 1.661768 | 1.665809 | 1.63E-02 |
| MRFO | 0.285 | 2.2507 | 9.0366 | 0.2057 | 1.661756 | 1.667033 | 6.32E-03 |
| L_SHADE | 0.2881 | 2.2236 | 9.0366 | 0.2057 | 1.661702 | 1.667438 | 3.14E-02 |
| RSA | 0.2866 | 2.2369 | 9.036 | 0.2058 | 1.661829 | 1.671267 | 1.37E-02 |
| ChOA | 0.2943 | 2.1715 | 9.0365 | 0.2058 | 1.662822 | 1.672762 | 1.04E-02 |
| RUN | 0.2933 | 2.1788 | 9.0366 | 0.2057 | 1.661805 | 1.691868 | 4.12E-02 |
| COA | 0.2685 | 2.4649 | 9.1473 | 0.2063 | 1.69469 | 1.728184 | 1.90E-02 |
| SCO | 0.2862 | 2.2402 | 9.0366 | 0.2057 | 1.661727 | 1.744516 | 1.11E-01 |
| EO | 0.294 | 2.176 | 9.0332 | 0.2062 | 1.664816 | 1.79022 | 1.52E-01 |
| SCA | 0.3198 | 1.9843 | 9.0434 | 0.206 | 1.669648 | 1.823856 | 3.15E-01 |
| WOA | 0.2482 | 2.7324 | 8.9854 | 0.2081 | 1.692637 | 2.060651 | 5.03E-01 |
| CMAR | 0.1804 | 3.9662 | 9.2062 | 0.2049 | 1.772988 | 2.211716 | 3.96E-01 |
| SSA | 0.211 | 3.5803 | 8.9415 | 0.2158 | 1.808045 | 2.251145 | 2.02E-01 |

**Table 19. Holm's post-hoc test results with FCO as control algorithm.**

Friedman test p-value=2.874448e-84

| Algorithm | p-Value | Result |
|-----------|---------|--------|
| CMA-ES | p=3.623765e-13 | Significantly different (Better) |
| ChOA | p=1.807228e-06 | Significantly different (Worse) |
| SCO | p=1.571173e-04 | Significantly different (Worse) |
| CMAR | p=3.014717e-10 | Significantly different (Worse) |
| COA | p=1.783857e-26 | Significantly different (Worse) |
| DE | p=3.623765e-13 | Significantly different (Worse) |
| EO | p=2.311951e-05 | Significantly different (Better) |
| GWO | p=4.182987e-01 | Not significant (Worse) |
| L_SHADE | p=4.088034e-01 | Not significant (Worse) |
| MRFO | p=3.812909e-04 | Significantly different (Worse) |
| PSO | p=1.740026e-12 | Significantly different (Better) |
| QIO | p=2.956427e-01 | Not significant (Worse) |
| RSA | p=1.057536e-03 | Significantly different (Worse) |
| RUN | p=2.714608e-04 | Significantly different (Worse) |
| SCA | p=6.894414e-03 | Significantly different (Worse) |
| SSA | p=7.139893e-23 | Significantly different (Worse) |
| WOA | p=5.877893e-05 | Significantly different (Worse) |

**4.4.3 Pressure vessel design problem.** The Pressure Vessel Design Problem is a classical benchmark in constrained engineering optimization. The objective is to minimize the total manufacturing cost of a cylindrical pressure vessel, which includes expenses associated with materials, forming processes, and welding operations for both the cylindrical shell and the hemispherical end caps [12].

The design problem involves four decision variables, as summarized in Table 20:

The objective function, representing the total cost (USD), is defined as:

$$f(x) \;=\; 0.6224\, x_1\, x_3\, x_4 + 1.7781 x_2\, x_3{}^2 + 3.1661 x_1{}^2 x_4 + 19.84 x_1{}^2 x_3 \tag{39}$$

This nonlinear cost function captures the combined influence of material volume, forming complexity, and welding requirements for both the cylindrical and hemispherical components of the vessel.

To ensure the feasibility of the design in real world applications, the model includes four nonlinear constraints:

**Con1. Shell thickness constraint** (This ensures that the shell thickness is sufficient relative to the vessel's radius for structural integrity):

$$g_1(x): \; -x_1 + 0.0193\, x_3 \leq 0 \tag{40}$$

**Table 20. Variable of pressure vessel design problem.**

| Variable | Description | Symbol | Unit | Range |
|----------|-------------|--------|------|-------|
| $X_1$ | Thickness of the shell | $T_s$ | Inch | [0, 2.0] |
| $X_2$ | Thickness of the head | $T_h$ | Inch | [0, 2.0] |
| $X_3$ | Inner radius of the vessel | $R$ | Inch | [10.0, 200.0] |
| $X_4$ | Length of the cylindrical section | $L$ | Inch | [10.0, 240.0] |

**Con2. Head thickness constraint** (This similarly guarantees adequate thickness for the hemispherical heads based on the internal radius):

$$g_2(x): \ -x_2 + 0.00954 \, x_3 \leq \ 0 \tag{41}$$

**Con3. Volume constraint** (This limits the vessel's volume to not exceed a specified maximum capacity, maintaining design feasibility and compliance with application requirements):

$$g_3(x): \ -\pi x_3^2 \, x_4 - (4/3) \, \pi \, x_3^3 \ + 1296000 \ \leq \ 0 \tag{42}$$

**Con4. length constraint** (This restricts the maximum length of the cylindrical section, ensuring manufacturability and spatial compatibility):

$$g_4(x): x_4 - 240 \ \leq \ 0 \tag{43}$$

Overall, this optimization problem presents a challenging constrained design task, suitable for evaluating the performance of advanced metaheuristic algorithms. The nonlinearity of the objective and constraints, the discrete continuous nature of the variables, and the tight feasibility region make it an ideal benchmark for testing algorithmic robustness and convergence accuracy.

The geometric definition and constraint landscape of the Pressure Vessel Design Problem are illustrated in Fig 14. Panel (a) presents the full-scale CAD model consisting of a cylindrical shell and two hemispherical heads mounted on saddle supports. Panel (b) shows the section A–A, where the four main design variables shell thickness ($T_s$), head thickness ($T_h$), inner radius(R), and cylindrical length (L) are clearly labeled. Panel (c) displays the feasible design region determined by the nonlinear constraints (g ≤ 0), corresponding to the manufacturing and volumetric limitations given in Eqs (39–43). This standardized engineering visualization ensures clarity in variable interpretation and physical proportions suitable for reproducible benchmarking.

The comparative optimization results of these methods evaluated over 30 independent runs are comprehensively summarized in Table 21, highlighting both the best-found costs and the statistical consistency of each approach.

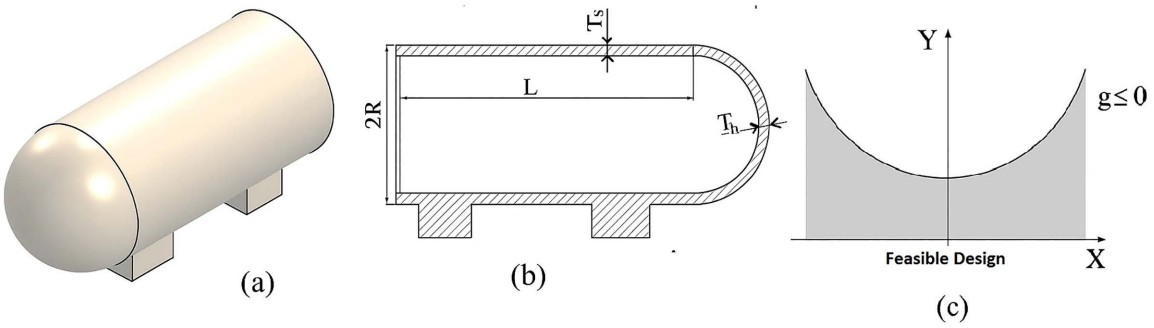

**Fig 14. Schematic illustration of the Pressure Vessel Design Problem. (a)** Three-dimensional CAD view of the cylindrical vessel with two hemispherical heads supported on saddle bases. **(b)** Technical section A–A showing the four design variables: shell thickness ($T_s$), head thickness ($T_h$), inner radius (R), and cylindrical length (L). **(c)** Feasible design region based on the nonlinear constraint boundaries (g ≤ 0).

**Table 21. Statistical comparison of eighteen metaheuristic algorithms applied to the Pressure Vessel Design Problem.**

| Algorithm | Optimum variable | | | | Optimum cost (USD) | Mean Cost (USD) | Std |
|---|---|---|---|---|---|---|---|
| | $T_s$ (in) | $T_h$ (in) | $R$ (in) | $L$ (in) | | | |
| DE | 0.6773 | 0.3715 | 37.699 | 240 | 5692.592 | 5692.592 | 0.00E+00 |
| QIO | 0.6577 | 0.3493 | 37.699 | 240 | 5692.592 | 5692.592 | 0.00E+00 |
| GWO | 0.6589 | 0.3719 | 37.699 | 240 | 5692.592 | 5692.633 | 4.28E-02 |
| PSO | 0.691 | 0.3975 | 37.699 | 240 | 5692.592 | 5707.72 | 2.35E+01 |
| FCO | **0.671** | **0.3782** | **37.6991** | **240** | **5692.629** | **5721.452** | **2.38E+01** |
| SCO | 0.6755 | 0.3972 | 37.699 | 240 | 5692.592 | 5741.758 | 1.81E+02 |
| ChOA | 0.7066 | 0.3886 | 37.699 | 240 | 5692.598 | 5794.407 | 1.50E+02 |
| COA | 0.6988 | 0.4057 | 37.7436 | 239.5953 | 5699.668 | 5815.216 | 8.31E+01 |
| L_SHADE | 0.7011 | 0.4003 | 37.9286 | 236.1912 | 5699.862 | 5837.212 | 1.31E+02 |
| RSA | 0.7 | 0.3695 | 37.699 | 239.9994 | 5692.592 | 5966.87 | 4.42E+02 |
| MRFO | 0.6756 | 0.3531 | 37.699 | 240 | 5692.592 | 5981.718 | 2.48E+02 |
| RUN | 0.6827 | 0.3764 | 37.699 | 240 | 5692.592 | 6079.501 | 3.29E+02 |
| EO | 0.6573 | 0.3722 | 37.7047 | 239.9096 | 5692.799 | 6165.223 | 4.76E+02 |
| CMA-ES | 0.6898 | 0.3788 | 37.699 | 240 | 5692.592 | 6165.583 | 3.29E+02 |
| SCA | 0.7077 | 0.3613 | 38.0229 | 234.697 | 5705.063 | 6482.859 | 6.28E+02 |
| CMAR | 0.6822 | 0.3732 | 37.699 | 240 | 5692.592 | 6642.764 | 1.03E+03 |
| WOA | 0.7493 | 0.3503 | 40.1392 | 202.5273 | 5744.909 | 6677.386 | 7.83E+02 |
| SSA | 0.7398 | 0.3964 | 41.2661 | 190.025 | 5845.384 | 6803.25 | 5.57E+02 |

FCO ranked fifth among the eighteen algorithms, achieving a mean cost of 5721.452 USD remarkably close to the global optimum of 5692.592 USD obtained by the top performers (DE and QIO). The 0.5% gap is negligible for this engineering problem, indicating FCO's ability to consistently converge toward near-optimal solutions.

Furthermore, FCO exhibited excellent convergence stability, with a standard deviation of 23.7681, comparable to PSO (23.50418) and significantly superior to SCO (181.10) and ChOA (150.39). Although DE and QIO showed zero variance potentially indicating premature convergence to a single point FCO's low but non-zero variance reflects a healthier search dynamic, suggesting balanced exploration and exploitation.

The Friedman test verified significant performance differences among the 18 methods ($p = 6.56 \times 10^{-65}$). A Holm post-hoc analysis was then conducted using FCO as the control algorithm (Table 22), yielding detailed pairwise comparisons. Results revealed that FCO was statistically superior to thirteen algorithms including ChOA, COA, L_SHADE, and MRFO while no significant difference was detected between FCO and SCO. Nevertheless, FCO demonstrated a clear practical advantage with a lower mean cost and far greater stability (Std 23.7 vs. 181.1).

Although minor statistical differences favored DE, QIO, GWO, and PSO, their tiny cost deviations (< 0.6%) render little practical impact. Overall, the Pressure Vessel Design benchmark supports the robustness and efficiency of the proposed FCO algorithm. By consistently producing high-quality near-optimal solutions with commendable repeatability, FCO shows significant advantage over most tested methods and stands as a reliable choice for complex constrained engineering design optimization. This behavior aligns with its stable performance observed in the Welded Beam and Spring Design benchmarks.

## 5. Conclusion

This study introduced the Felis Catus Optimization (FCO) algorithm, a nature-inspired metaheuristic derived from the social dynamics and natural behavioral patternsof domestic cats. The method employs a unified population in

**Table 22. Post-hoc wilcoxon test results against FCO.**

Friedman test p-value=6.563907e-65

| Algorithm | p-Value | Result |
| --- | --- | --- |
| ChOA | p=1.107777e-02 | Significantly different (Worse) |
| CMA-ES | p=6.680940e-10 | Significantly different (Worse) |
| CMAR | p=7.833827e-06 | Significantly different (Worse) |
| COA | p=1.690626e-07 | Significantly different (Worse) |
| DE | p=1.128663e-08 | Significantly different (Better) |
| EO | p=3.864866e-06 | Significantly different (Worse) |
| GWO | p=1.170407e-08 | Significantly different (Better) |
| L_SHADE | p=1.259576e-05 | Significantly different (Worse) |
| MRFO | p=3.955284e-07 | Significantly different (Worse) |
| PSO | p=2.825209e-02 | Significantly different (Better) |
| QIO | p=1.128663e-08 | Significantly different (Better) |
| RSA | p=3.554701e-03 | Significantly different (Worse) |
| RUN | p=1.637275e-07 | Significantly different (Worse) |
| SCA | p=1.192179e-08 | Significantly different (Worse) |
| SCO | p=5.449630e-01 | Not significant (Worse) |
| SSA | p=3.008820e-15 | Significantly different (Worse) |
| WOA | p=9.918504e-09 | Significantly different (Worse) |

which agents dynamically alternate between exploration (male-type) and exploitation (female-type) roles, driven by adaptive competition, probabilistic reallocation, and age-dependent selection. These mechanisms collectively maintain population diversity and ensure an effective balance between global search and local refinement throughout the optimization process. Extensive evaluations on 17 CEC 2005 and 30 CEC 2017 benchmark functions, together with three constrained engineering design problems, confirmed that FCO yields reproducible, statistically validated, and stable optimization outcomes. Compared with 18 established metaheuristic algorithms including L-SHADE, CMA-ES, RUN, RSA, SCO, and QIO, FCO consistently achieved lower average errors and smoother convergence tendencies under identical experimental conditions. In engineering case studies (welded-beam, compression-spring, and pressure-vessel design), it produced feasible and cost-efficient solutions, demonstrating adaptability to nonlinear andstrained domains.

From a methodological perspective, FCO integrates biologically interpretable selection pressure, adaptive agent reallocation, and stochastic role transitions within a single coherent framework. This formulation allows flexible extension while remaining computationally tractable. Overall, FCO provides a robust and interpretable optimization approach applicable to a broad range of continuous and constrained problems and offers a promising foundation for further methodological refinement and interdisciplinary applications.

### 5.1 Future studies and limitations

Despite its competitive performance, FCO has certain limitations that outline future research directions [50]. Firstly, the algorithm's performance, similar to other population-based methods, is dependent on the careful tuning of several hyper-parameters [51]. This can reduce its usability for non-expert practitioners [52,53]. Therefore, developing adaptive or self-tuning parameter control mechanisms would be a valuable research direction to enhance the algorithm's autonomy and efficiency [54,55]. Secondly, future research could focus on parallel implementations to reduce runtime and improve scalability, particularly for very high-dimensional problems [56,57]. Finally, while this study concentrated on continuous search spaces, FCO's capabilities in solving discrete, binary, multi-objective, and dynamic optimization problems have not

yet been fully investigated [58]. Extending FCO to these domains, along with providing deeper theoretical analyses such as formal convergence proofs, could further strengthen its credibility and scope of application.

In summary, Felis Catus Optimization establishes itself as a powerful, reliable, and scalable optimization algorithm that, by drawing inspiration from a complex natural phenomenon, offers an innovative solution to optimization challenges and provides a solid foundation for future algorithmic developments.

## Supporting information

**S1 Appendix. Algorithm parameter settings and experimental budget.**
(DOCX)

**S2 Appendix. Convergence curves (CEC 2005 benchmark suite).**
(DOCX)

**S3 Appendix. Convergence curves (CEC 2017 benchmark suite).**
(DOCX)

**S1 Data. FCO codes v3.**
(ZIP)

## Acknowledgments

The authors would like to thank the reviewers and the editor-in-chief for their helpful comments and recommendations

**AI Usage Disclosure:** The authors used ChatGPT (OpenAI, GPT-5 model, 2025) exclusively for improving English clarity and correcting minor grammatical errors in the manuscript. All computational codes were designed and logically structured by the authors; the AI models were employed only to generate code fragments strictly following the authors' explicit algorithmic instructions. Some illustrative figures were initially generated by AI tools to visualize the concepts; however, due to symbol and logic inconsistencies, these images were manually corrected and scientifically verified by the authors to ensure full compliance with the study's methodology and data integrity.

No numerical results, analytical data, or final interpretations were produced by any AI system.

## Author contributions

**Data curation:** Mohammad Salehi.

**Formal analysis:** Mohammad Salehi, Mirpouya Mirmozaffari.

**Methodology:** Mohammad Salehi.

**Software:** Mirpouya Mirmozaffari.

**Supervision:** Raouf Khayami.

**Validation:** Raouf Khayami, Mirpouya Mirmozaffari.

**Visualization:** Mirpouya Mirmozaffari.

**Writing – original draft:** Mirpouya Mirmozaffari.

**Writing – review & editing:** Mirpouya Mirmozaffari.

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
