## [Decision Letter · Decision Letter 0]

2 Oct 2025

Thank you for submitting your manuscript to PLOS ONE. After careful consideration, we feel that it has merit but does not fully meet PLOS ONE’s publication criteria as it currently stands. Therefore, we invite you to submit a revised version of the manuscript that addresses the points raised during the review process.

We look forward to receiving your revised manuscript.

Kind regards,

Aykut Fatih Güven, Ph.D.

Academic Editor

PLOS ONE

Journal Requirements:

3. Please include captions for your Supporting Information files at the end of your manuscript, and update any in-text citations to match accordingly. Please see our Supporting Information guidelines for more information: http://journals.plos.org/plosone/s/supporting-information ..

Additional Editor Comments:

Following peer review, your manuscript requires major revision. In particular, clarifications are needed regarding the algorithmic description and its implementation, the inclusion of additional comparative analyses, and more rigorous reporting of complexity, convergence, and statistical results. Please revise the manuscript accordingly and provide a detailed response to the reviewers’ comments.

Reviewer's Responses to Questions

**Comments to the Author**

1. Is the manuscript technically sound, and do the data support the conclusions?

Reviewer #1: Yes

Reviewer #2: Yes

2. Has the statistical analysis been performed appropriately and rigorously?

Reviewer #1: Yes

Reviewer #2: Yes

3. Have the authors made all data underlying the findings in their manuscript fully available?

Reviewer #1: Yes

Reviewer #2: Yes

4. Is the manuscript presented in an intelligible fashion and written in standard English?

Reviewer #1: Yes

Reviewer #2: Yes

Reviewer #1: Dear authors.

Below is a concise, author-facing summary of my review (the full version with mathematical formulas is in the attached file). It highlights what works well, the main concerns we found, and the concrete actions we recommend before further consideration:

Summary of the manuscript

You propose a new nature-inspired metaheuristic, Felis Catus Optimization (FCO). The intended design divides the population into “male” explorers and “female” exploiters that interact on a 2-D grid of territories, with one dominant male per cell. Additional mechanisms include aging and elimination, as well as a tournament-style migration. You evaluate FCO on the CEC 2005 and CEC 2017 benchmark suites and on three engineering problems (spring, welded beam, pressure vessel), report nonparametric statistics, and present an ablation study. Code availability is announced via a DOI in Zenodo.

Strengths

• The core idea is original and engaging: territorial dominance plus an explicit split of exploration vs. exploitation.

• The experimental scope is broad (two CEC suites + three engineering case studies).

• You aim for robustness via multiple runs and statistical tests; ablation is a valuable addition.

• Code availability is claimed, which is essential for reproducibility.

High-level weaknesses

• The mathematical/model description contains several ambiguities (grid definition, cell membership, migration/redistribution rules, capacity limits, etc.).

• The provided MATLAB code is inconsistent with the model as written (no 2-D grid/territories; update rules differ materially from the described ones).

• The comparative baseline is too small and not representative of the current state-of-the-art.

• Reproducibility details (defaults, seeding, exact statistical procedures) are incomplete.

• Parts of the engineering section have figure/content issues and include visualizations that are not outputs of FCO.

• Language/terminology inconsistencies reduce clarity.

Specific issues to address in the model description

• Grid notation and meaning: separate clearly the grid size from the set of cells; avoid self-referential notation and mixed one-index/two-index symbols.

• Cell membership: explicitly include each agent’s cell as part of its state, or define a precise mapping from agents to cells.

• Initialization: specify how agents are assigned to cells at the start (random, round-robin, one male per cell, etc.).

• Adjacency and challenges: reconcile “current or adjacent cell” with “from another cell” and define adjacency (4- or 8-neighborhood, wraparound, etc.). State whether intra-cell challenges are allowed.

• Capacity and assignment: define how females are assigned “based on proximity and dominance” (which metric, in which space), in what order, how capacity limits are enforced, and how ties are broken.

• “Less competitive cell”: define the criterion used in tournament migration (e.g., fitness of the dominant male, number of females, or a composite score).

• Aging/elimination: beyond an age threshold, specify all triggers (e.g., prolonged loss of territory, stagnation) and the exact action taken.

• Boundary handling: state the policy when a move leaves the domain (clamp, reflect, wrap).

• Defaults/schedules: provide recommended values/ranges and any adaptivity for key hyperparameters (movement coefficients, harém capacity, age threshold, grid size).

• After a dominant male dies: give a concrete algorithm for redistributing females, for inserting a new male (including where he is placed), and, if multiple males can belong to one cell, for succession (who becomes the new dominant). If multiple males per cell are disallowed, state that explicitly.

Model–code inconsistency (critical)

The MATLAB implementation we received does not realize the grid/territory model described in the text. In particular:

• There is no 2-D grid of cells at runtime. The initialization of “grid-based” logic stratifies roughly 20% of agents along the first coordinate and places the rest near selected “males”; however, thereafter, the algorithm operates without cells, dominants, or neighborhood structure.

• Movement rules diverge from the manuscript’s description: all agents share the same move operator toward either the global best or a tournament-selected peer; there is no distinct “female-only” local Gaussian step.

• Tournament migration is implemented as a global relocation of aged agents near the best of several random candidates—not as a migration between two selected cells or toward a “less competitive” cell.

• Aging/elimination logic performs relocations with age reset rather than elimination and replacement as implied in the text.

• Sex/role is not used after initialization; behavior switches simply between global-best and peer-targeted moves.

• There is a boundary inconsistency: one routine clamps positions to the problem bounds, while another clamps to [0,1], which breaks general-domain runs.

Action: either (a) revise the manuscript to describe exactly the implemented algorithm (and remove grid/territory constructs you do not implement), or (b) revise the code to implement the grid/territory model as written (cell membership, one dominant male per cell, defined adjacency, female-specific local search, proper tournament migration between cells, consistent boundary handling). Please also supply a minimal script that reproduces the main tables and figures from a clean run.

Comparative baseline (major)

The current baseline set (~10 methods; mostly GWO, PSO, GSA, LFD, TSA, MVO, SSA, WSO, AVOA, RSA, WOA) is too narrow and not representative. For a fair and convincing comparison, please raise the number of baselines to at least 15 and include established CEC winners/top performers and highly cited recent methods. Specifically, we recommend adding: L-SHADE (and variants), CMA-ES, EBO with CMAR, plus recent high-impact algorithms such as RUN (Beyond the Metaphor), Coati Optimization Algorithm, and Quadratic Interpolation Optimization (QIO). Ensure diversity across families (DE/ES/metaphor-free/swarm), and report consistent budgets and parameter settings.

Statistical analysis and reproducibility

• Specify exactly the statistical pipeline: which tests on which metrics (best/mean/median), whether and how multiple comparisons are controlled (e.g., Friedman with Nemenyi), and how the critical difference is computed.

• Report seeds or seeding policy, software/version details, and any normalization used.

• Provide default hyperparameters for FCO (including schedules) in a single table.

• Ensure the Zenodo package contains all scripts needed to regenerate tables/figures from scratch.

Engineering case studies: quality and relevance

• For each reported optimum, print the values of all constraint functions with units (stress limits, deflection, geometry, etc.) so feasibility can be verified at a glance.

• Fix the figure set: for example, in one multi-panel figure, the content of panels a–c does not match the captions, and panel d is blurred.

• Remove or relocate “stress/displacement heat maps”; they are not outputs of FCO and should only appear, if at all, in a separate CAE validation appendix with a clear explanation of their source.

Language and presentation

• Unify terminology (grid/zone/cell; male/explorer; female/exploiter) and correct technical names (e.g., Davidon–Fletcher–Powell).

• Provide a concise symbol glossary and ensure consistent units and variable names in tables and captions.

• Remove residual submission-system artifacts and ensure figure/table references are consistent.

Reviewer #2: The manuscript proposes Felis Catus Optimization (FCO), a nature-inspired metaheuristic that partitions the population into explorer (male) and exploiter (female) agents, uses a grid-based exploration for males and Gaussian local search for females, and includes aging/migration mechanisms. The authors evaluate FCO on 17 CEC-2005 and 28 CEC-2017 benchmark functions and three engineering design problems, and report statistical tests (Friedman and Wilcoxon).

Positive points

Novel idea: The dual male/female agent design with grid territories and tournament migration is an interesting, biologically-motivated hybridization that is clearly described.

Extensive empirical evaluation: The manuscript uses large standard benchmark sets (CEC 2005 and CEC 2017) and three classical constrained engineering problems (welded beam, compression spring, pressure vessel). This breadth of tests is commendable.

Statistical testing: The authors applied non-parametric tests (Friedman and Wilcoxon) to support claims of significance.

Ablation study: There is at least a component analysis that identifies the importance of the grid-based exploration.

Major concerns (must be addressed before acceptance)

Missing comparison with the specified algorithm ("Sand Cat swarm optimization")

The reviewer’s instruction requested a theoretical and experimental comparison with Sand Cat swarm optimization: A nature-inspired algorithm to solve global optimization problems. The manuscript does not include this comparison (no mention found). Add a focused subsection that (a) summarizes Sand Cat's mechanisms and complexity, (b) discusses conceptual similarities/differences to FCO, and (c) runs head-to-head experiments on the same benchmarks (same parameter settings, same seeds/runs) so comparisons are fair. (If the authors cannot obtain Sand Cat code, they must re-implement it using the original paper and make the code/public data available.)

Time / computational complexity analysis is missing

The manuscript itself notes that "formal computational complexity has not yet been rigorously analyzed." Provide a clear complexity analysis (time complexity per iteration and per run as a function of population size (N), number of dimensions (D), grid size (G), and number of iterations (T)). Also report wall-clock runtime comparisons (mean ± std) under a fixed hardware/software setup to quantify the overhead of grid management, migration, and aging.

Convergence curves and exploration vs exploitation metrics are insufficient/absent

The paper reports mean improvements and convergence time improvements, but I could not find explicit convergence plots across iterations for the main benchmarks nor plots quantifying exploration vs exploitation dynamics. Add:

a) representative convergence curves (fitness vs iteration) averaged across runs (with shaded ±1 std) for several benchmark problems (unimodal, multimodal, composite).

b) an exploration/exploitation diagnostic (e.g., population diversity measure, step-size distribution, or a specific "exploration vs exploitation score" over iterations) and a plot showing how FCO trades off exploration/exploitation over time. If possible include these for both FCO and key competitors.

Clarity on statistics, reporting and variability

Some tables already show Avg and Std (e.g., Table 10 entries include Avg and Std), which is good, but the discussion must systematically interpret both mean and variance. For every benchmark / engineering problem: report (i) mean ± std across all runs, (ii) median, (iii) number of successful runs (if relevant), and (iv) effect sizes. Also, when presenting Wilcoxon results mention sample size and exact p-values, not only “p<0.05”.

Specific required experiments / analyses

Add Sand Cat comparison (theoretical paragraph + experimental table + convergence curves).

Complexity and runtime: include Big-O analysis and measured runtimes (mean ± std) per benchmark. Provide algorithm pseudo-code complexity annotation or a small table summarizing costs of each operator.

Convergence and exploration/exploitation plots: averaged curves (with ±std shading) across runs for representative functions (at least one unimodal, one multimodal, and one composition). Also add a diversity/time plot (e.g., population standard deviation of positions or average inter-agent distance).

Report reproducibility artifacts: parameter settings table, random seeds or seed generation procedure, number of independent runs (≥30 recommended), and a link to code/data (Zenodo link present — ensure it contains all scripts and seed settings).

Minor / editorial suggestions

Related work — expand taxonomy: the related work should be reorganized to explicitly include different inspiration sources the reviewer requested (e.g., physics-based, human-inspired, swarm-based, predator-prey, and social/territorial). Cite representative algorithms in each category and explicitly state how FCO differs (mechanisms and objectives). This will strengthen claims of novelty.

Tables and Figures: ensure every table shows units, algorithm hyperparameters used, and the #runs. Make convergence plots with consistent axis scales so comparisons are fair. If Table 10 is used as example, keep the Avg/Std columns and add a "best found" column.

Statistical reporting: present exact p-values, and where appropriate include post-hoc tests with correction for multiple comparisons (e.g., Holm or Bonferroni) when many pairwise Wilcoxon tests are reported.

Ablation clarity: expand ablation study details (which components were removed, how many runs, and statistical tests for each ablation). The current note that grid-based exploration contributes most is useful but needs more quantitative support.

**Do you want your identity to be public for this peer review?** For information about this choice, including consent withdrawal, please see our For information about this choice, including consent withdrawal, please see our Privacy Policy .

Reviewer #1: No

Reviewer #2: No

While revising your submission, please upload your figure files to the Preflight Analysis and Conversion Engine (PACE) digital diagnostic tool, https://pacev2.apexcovantage.com/ . PACE helps ensure that figures meet PLOS requirements. To use PACE, you must first register as a user. Registration is free. Then, login and navigate to the UPLOAD tab, where you will find detailed instructions on how to use the tool. If you encounter any issues or have any questions when using PACE, please email PLOS at . PACE helps ensure that figures meet PLOS requirements. To use PACE, you must first register as a user. Registration is free. Then, login and navigate to the UPLOAD tab, where you will find detailed instructions on how to use the tool. If you encounter any issues or have any questions when using PACE, please email PLOS at figures@plos.org . Please note that Supporting Information files do not need this step.. Please note that Supporting Information files do not need this step.

---

## [Author Response · Author response to Decision Letter 1]

7 Nov 2025

A detailed, point‑by‑point response to all reviewer and editor comments has been provided in the uploaded “Response to Reviewers” document. All requested revisions have been made accordingly in the revised manuscript and supporting files.

---

## [Decision Letter · Decision Letter 1]

26 Nov 2025

Dear Dr. Salehi,

Thank you for submitting your manuscript to PLOS ONE. After careful consideration, we feel that it has merit but does not fully meet PLOS ONE’s publication criteria as it currently stands. Therefore, we invite you to submit a revised version of the manuscript that addresses the points raised during the review process.

The manuscript presents a promising idea but requires substantial improvement before further consideration. The described model and the provided implementation are not yet aligned, and further clarification and reproducibility updates are needed. A comprehensive revision is requested, including corrections to the model description, implementation details, and outputs.

Please submit your revised manuscript by Jan 10 2026 11:59PM. If you will need more time than this to complete your revisions, please reply to this message or contact the journal office at plosone@plos.org . . A rebuttal letter that responds to each point raised by the academic editor and reviewer(s). You should upload this letter as a separate file labeled 'Response to Reviewers'.A marked-up copy of your manuscript that highlights changes made to the original version. You should upload this as a separate file labeled 'Revised Manuscript with Track Changes'.An unmarked version of your revised paper without tracked changes. You should upload this as a separate file labeled 'Manuscript'.

We look forward to receiving your revised manuscript.

Kind regards,

Aykut Fatih Güven, Ph.D.

Academic Editor

PLOS ONE

Journal Requirements:

Additional Editor Comments (if provided):

The manuscript presents a promising idea but requires substantial improvement before further consideration. The described model and the provided implementation are not yet aligned, and further clarification and reproducibility updates are needed. A comprehensive revision is requested, including corrections to the model description, implementation details, and outputs.

Reviewers' comments:

Reviewer's Responses to Questions

**Comments to the Author**

Reviewer #1: (No Response)

Reviewer #2: All comments have been addressed

2. Is the manuscript technically sound, and do the data support the conclusions?

Reviewer #1: Partly

Reviewer #2: Yes

3. Has the statistical analysis been performed appropriately and rigorously?

Reviewer #1: Yes

Reviewer #2: Yes

4. Have the authors made all data underlying the findings in their manuscript fully available?

Reviewer #1: Yes

Reviewer #2: Yes

5. Is the manuscript presented in an intelligible fashion and written in standard English?

Reviewer #1: Yes

Reviewer #2: Yes

Reviewer #1: Overall, the idea is promising, but it needs significant revision. The MATLAB code does not implement the described grid or territory model. Either create a formal grid (including cells, dominance, adjacency, and redistribution), or revise the text to reflect the actual ecological dynamics. Improve or correct the baselines, fix code issues, ensure full reproducibility, and update figures and tables (including Table 5). Please submit the updated code and regeneration scripts along with the revision.

Reviewer #2: The revised report has been thoroughly reviewed, and I can confirm that all previous comments have been successfully addressed.

The document is now excellent and approved for finalization. Thank you for your diligent work and high-quality revisions.

**Do you want your identity to be public for this peer review?** For information about this choice, including consent withdrawal, please see our For information about this choice, including consent withdrawal, please see our Privacy Policy .

Reviewer #1: No

Reviewer #2: No

---

## [Author Response · Author response to Decision Letter 2]

29 Nov 2025

We respectfully confirm that all reviewer and editor comments in the decision letter have been fully addressed in the revised manuscript.

All requested corrections are incorporated in the Track Changes version, and a detailed point‑by‑point rebuttal has been provided.

We appreciate the opportunity to submit Revision 2 and thank the reviewers and the Academic Editor for their constructive guidance.

---

## [Decision Letter · Decision Letter 2]

6 Jan 2026

Felis Catus Optimization (FCO): A Novel Nature Inspired Metaheuristic Algorithm

PONE-D-25-48643R2

Dear Dr. SALEHI,

We’re pleased to inform you that your manuscript has been judged scientifically suitable for publication and will be formally accepted for publication once it meets all outstanding technical requirements.

Kind regards,

Aykut Fatih Güven, Ph.D.

Academic Editor

PLOS One

Additional Editor Comments (optional):

Dear Author,

I am pleased to inform you that your revised manuscript entitled “Felis Catus Optimization (FCO): A Novel Nature-Inspired Metaheuristic Algorithm” has been accepted for publication in PLOS ONE.

The reviewers’ comments have been satisfactorily addressed, and the manuscript is now suitable for publication.

Reviewers' comments:

Reviewer's Responses to Questions

**Comments to the Author**

Reviewer #1: All comments have been addressed

2. Is the manuscript technically sound, and do the data support the conclusions?

Reviewer #1: Yes

3. Has the statistical analysis been performed appropriately and rigorously?

Reviewer #1: Yes

4. Have the authors made all data underlying the findings in their manuscript fully available?

Reviewer #1: Yes

5. Is the manuscript presented in an intelligible fashion and written in standard English?

Reviewer #1: Yes

Reviewer #1: I am satisfied with the changes made (the most important ones are complete) and consider them finished. The article is now ready for publication—only a final technical review by the authors remains (e.g., some figures are wider than the page width).

**Do you want your identity to be public for this peer review?** For information about this choice, including consent withdrawal, please see our For information about this choice, including consent withdrawal, please see our Privacy Policy .

Reviewer #1: No

---

## [Editor Report · Acceptance letter]

PONE-D-25-48643R2

PLOS One

Dear Dr. Salehi,

I'm pleased to inform you that your manuscript has been deemed suitable for publication in PLOS One. Congratulations! Your manuscript is now being handed over to our production team.

Kind regards,

on behalf of

Assoc. Prof. Dr. Aykut Fatih Güven

Academic Editor

PLOS One